# Systematic mapping of antibiotic cross-resistance and collateral sensitivity with chemical genetics

Nazgul Sakenova[1,2,3], Elisabetta Cacace [1,9], Askarbek Orakov [4], Florian Huber[1], Vallo Varik [1], George Kritikos [1,10], Jan Michiels [2,3], Peer Bork[4,5,6], Pascale Cossart[1,7], Camille V. Goemans [1,8] & Athanasios Typas [1,4]

By acquiring or evolving resistance to one antibiotic, bacteria can become cross-resistant to a second antibiotic, which further limits therapeutic choices. In the opposite scenario, initial resistance leads to collateral sensitivity to a second antibiotic, which can inform cycling or combinatorial treatments. Despite their clinical relevance, our knowledge of both interactions is limited. We used published chemical genetics data of the *Escherichia coli* single-gene deletion library in 40 antibiotics and devised a metric that discriminates between known cross-resistance and collateral-sensitivity antibiotic interactions. Thereby we inferred 404 cases of cross-resistance and 267 of collateral-sensitivity, expanding the number of known interactions by over threefold. We further validated 64/70 inferred interactions using experimental evolution. By identifying mutants driving these interactions in chemical genetics, we demonstrated that a drug pair can exhibit both interactions depending on the resistance mechanism. Finally, we applied collateral-sensitive drug pairs in combination to reduce antibiotic-resistance development in vitro.

Although antibiotic resistance is increasing at alarming rates[1], fewer and fewer novel antibiotics are being approved for clinical use[2,3]. Importantly, the development or acquisition of resistance to one drug can lead to cross-resistance (XR)[4] to other drugs, limiting treatment options. The same processes can also give rise to collateral sensitivity (CS)[5] to other drugs due to trade-offs or fitness costs of resistance mechanisms[6,7] (Fig. 1a). The principle of CS has been successfully used to reduce the rates of resistance emergence[8–15], or even re-sensitize microorganisms to antibiotics[16], by combining or cycling CS drug pairs. In an era

of diminishing therapeutic options, knowledge of XR and CS is more important than ever.

The most common approach to measure XR and CS is to experimentally evolve resistance to one drug for several lineages and then measure their susceptibility to another drug (Fig. 1a). Our understanding of the underlying mechanism(s) relies on sequencing the genomes of the evolved strains[8,17–19]. Although powerful, this approach is effort, scale and cost heavy. Hence, current knowledge of XR and CS interactions is limited to a few bacteria and a relatively small number of

[1]Genome Biology Unit, European Molecular Biology Laboratory, Heidelberg, Germany. [2]Center for Microbiology, VIB–KU Leuven, Leuven, Belgium. [3]Center of Microbial and Plant Genetics, KU Leuven, Leuven, Belgium. [4]Molecular Systems Biology Unit, European Molecular Biology Laboratory, Heidelberg, Germany. [5]Department of Bioinformatics, University of Würzburg, Würzburg, Germany. [6]Max Delbrück Centre for Molecular Medicine, Berlin, Germany. [7]Department of Cell Biology and Infection, Institut Pasteur, Paris, France. [8]Global Health Institute, School of Life Sciences, École Polytechnique Federale de Lausanne, Lausanne, Switzerland. [9]Present address: Institute of Microbiology and Swiss Institute of Bioinformatics, ETH Zürich, Zürich, Switzerland. [10]Present address: European Food Safety Authority, Parma, Italy. ✉e-mail: camille.goemans@epfl.ch; typas@embl.de

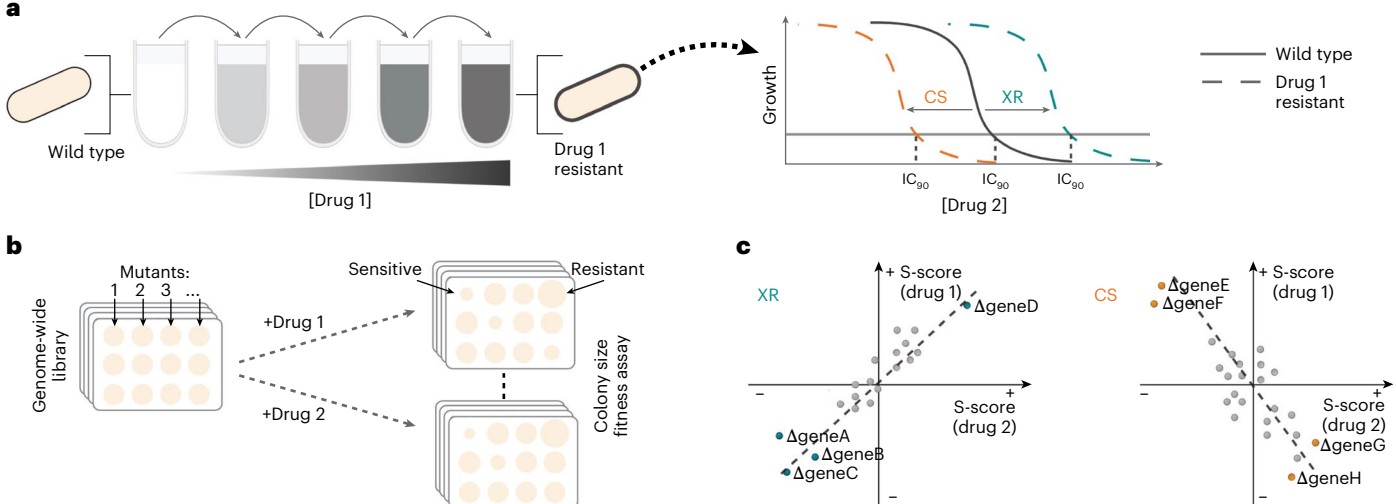

**Fig. 1 | Chemical genetics allow for systematic XR and CS assessment.**
**a**, Schematic illustration of the conventional way XR and CS drug interactions are assessed by experimental evolution. Resistant mutants selected by drug 1 are tested for susceptibility to drug 2. The MIC, or 90% inhibitory concentration (IC$_{90}$), of drug 2 is compared with that of the ancestral strain. **b**, Schematic illustration of chemical genetic screens with arrayed libraries. Several drugs (drug 1, 2 and so on) are profiled across genome-wide gain-of-function or loss-of-function mutant libraries. The fitness of each mutant is

evaluated independently—for example, by measuring colony size. **c**, XR and CS are associated with chemical genetics profile similarity and dissimilarity, respectively. The s-score (used as a proxy for fitness) of each deletion mutant is plotted for two drugs involved in either XR or CS. If the same mutations make cells more resistant or sensitive to two drugs, cells are more likely to evolve mechanisms that inhibit or promote these exact processes during evolution and become XR to both drugs, whereas the opposite is true for CS.

antibiotics[8,9,12,16–26]. Importantly, experimental evolution probes a limited number of lineages and a small part of the solution space in terms of possible resistance mutations, which strongly depends on the selection pressure applied. This may lead to inconsistencies when assessing drug-pair interactions. Furthermore, experimental evolution leads to the acquisition of numerous mutations that make the identification of causal resistance mechanisms difficult without additional experiments. To facilitate drug susceptibility testing of experimentally evolved strains or to dissect the evolved resistance mechanism(s), adaptations to the original method have been proposed—for example, automation of minimum inhibitory concentration (MIC) measurements[26] and phenotypic stratification of evolved strains[27,28]. Although these adaptations allow for an increase in the number of lineages, chemicals and interactions probed, the genetic space explored for resistance is limited and extensive sequencing, as well as previous knowledge are required to identify the causal resistance mechanisms. Here we set out to overcome these limitations by developing a predictive framework based on the systematic nature of chemical genetics screens.

Chemical genetics involve the systematic assessment of drug effects on genome-wide mutant libraries[29,30]. Such data have been previously shown to capture information on drug mode of action, resistance and interactions in *Escherichia coli*[31–36]. Importantly, chemical genetics systematically quantify how each gene in the genome contributes to resistance or susceptibility to a large set of drugs (Fig. 1b). The similarity between chemical genetic profiles for different drugs has been reported to correlate with XR frequency[18] and has been used to minimize XR between antimicrobial peptides and antibiotics[37]. Several years ago we proposed that such chemical genetics data could be used to identify both XR and CS interactions by comparing drug profiles[30] (Fig. 1c), expediting the systematic identification of XR/CS interactions and mapping of their underlying mechanisms.

In this study we used available *E. coli* chemical genetics data[31] for 40 antibiotics (Methods) and explored different similarity metrics to identify the one that best discerns between known XR and CS interactions. We applied this metric to all antibiotic drug pairs therein and discovered three times more XR and six times more CS interactions than previously identified, including the reclassification of 116 previously tested

drug-pair relationships. We independently validated 8.3% (70/840) of these interactions by experimental evolution with 91% precision (64/70). By integrating all data into a drug-interaction network, we examined the monochromaticity (that is, if a given interaction is exclusively XR or CS) and conservation within antibiotic classes, identifying antibiotic (classes) with extensive XR or CS interactions. All data are available at https://shiny-portal.embl.de/shinyapps/app/21_xrcs. We also used the available chemical genetics data to identify the mutations driving specific interactions, thereby confirming known and resolving new mechanisms. Finally, we showed that newly identified CS pairs used in combination could reduce resistance evolution compared with single drugs. Overall, we present a systematic framework to accelerate XR and CS discovery and mechanism deconvolution, paving the way for the development of rationally designed antibiotic combination treatments.

## Results

### Building a training set of known XR and CS interactions
To build a training set of known XR and CS interactions, we collected data from four studies that performed experimental evolution in *E. coli*[8,17–19]. The majority of interactions (78%; 338/429) were only tested in one study. From the 91 antibiotic pairs tested in at least two studies, only one-third (*n* = 30; 20 neutral, nine XR and one CS) was called uniformly across studies, whereas 56 were called XR or CS interactions in one study but neutral in the other (Fig. 2a). The discrepancy between experimental evolution results could be due to several reasons: selection biases (for example, different selection pressure and number of generations used), slightly different criteria used to define XR and CS (for example, methods and cutoffs used for fitness-effect measurements), low power to call interactions (limited number of lineages tested) and population complexity (resistance or sensitivity assessment is typically done at the population level). We reasoned that most discrepancies were probably due to false negatives (interaction missed and reported neutral in one study), as studies were undersampling the antibiotic-resistance solution space and used different metrics to call interactions. For this reason, we designated drug pairs as XR or CS if they exhibited an interaction in at least one study, even if they were neutral in other(s). In contrast, drug pairs displaying conflicting

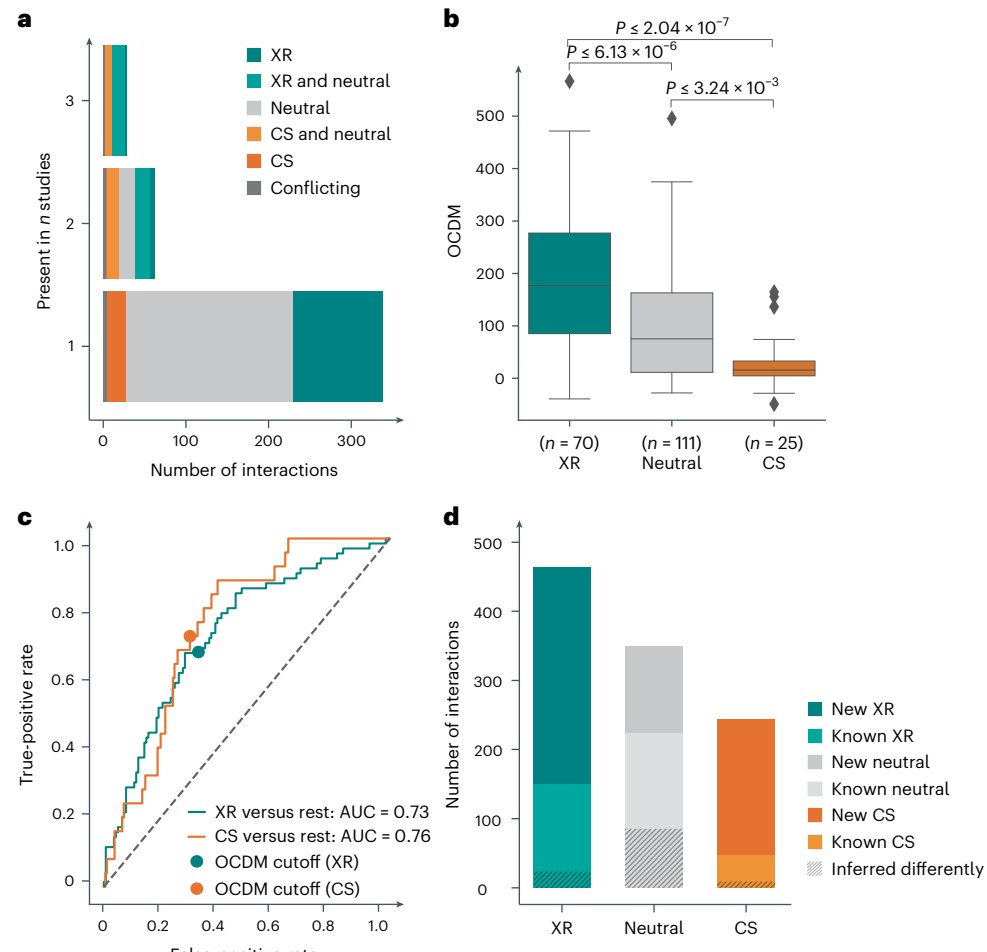

**Fig. 2 | Chemical genetics-derived metric separates well-known XR and CS interactions and infers new ones. a**, The overlap between published XR and CS interactions from four existing datasets[8,17–19] is low, even when directionality is not taken into account. **b**, A devised metric derived from chemical genetics profile similarity, OCDM, can robustly discern between known XR, CS and neutral interactions. False detection rate-adjusted *P* values were obtained from a two-sided Mann–Whitney *U*-test. The box boundaries represent the first and third quartiles, with the median indicated. The whiskers extend to the furthest data points within 1.5× the interquartile range. **c**, ROC curves for the classification of XR (positive class) versus non-XR (negative class) and CS (positive class) versus

non-CS (negative class). Each OCDM cutoff represents a point on the curve and is associated with a true-positive rate and a false-positive rate. The OCDM cutoffs chosen for XR and CS interactions are depicted with a closed circle. **d**, New XR, neutral and CS pairs inferred by chemical genetics using the OCDM cutoff expand the currently known XR and CS interactions in *E. coli* by two- and fourfold, respectively. This difference further increases if we take into account drug pairs for which the interaction is inferred differently from previous studies (Extended Data Fig. 2). Note that known interactions (*n* = 420 total) include drug pairs for which there is no available chemical genetics data.

responses (that is, XR in one study and CS in another) were excluded (*n* = 5). After comparing drugs for which chemical genetics data are available[31], we came up with 206 drug pairs (111 neutral, 70 XR and 25 CS) involving 24 different antibiotics (Source Data Fig. 2), which we used as the training set and ground truth for devising a chemical genetics-based metric to infer XR and CS relationships. The power of chemical genetics is that they probe the impact of loss-of-function mutations of each non-essential gene on the resistance or sensitivity to many drugs. In the chemical genetics data we used, the drug effects on each mutant are represented by s-scores, assessing the fitness of a mutant in one condition compared with its fitness across conditions[31,38] (Methods and Supplementary Table 1).

**Chemical genetics profile concordance identifies XR and CS**
Using our training set, we hypothesized that XR drugs should share resistance mechanisms (XR) and thus have concordant chemical genetic profiles, as previously suggested for a subset of XR pairs (*n* = 36)[18]. The opposite should be true for CS pairs, as mutations causing resistance to one drug would sensitize cells to another, leading to discordant

chemical genetics profiles for the two drugs (Fig. 1c). We first tested whether different correlation-based metrics from chemical genetics data could discriminate between known XR, CS or neutrality (Methods) but all performed poorly (area under the curve (AUC) for the receiver operating curve (ROC), 0.52–0.67; Extended Data Fig. 1a). We reasoned that the noise generated by the high proportion of neutral phenotypes in the chemical genetics data[31] was compromising performance. To overcome this, we used six features based only on extreme s-scores per condition: the sum and count of positive concordant s-scores, negative concordant s-scores and total discordant s-scores (Methods). We then trained decision tree models with these features for each drug pair. The trained classifier performed well, with the F1 score, recall, precision and ROC AUC consistently exceeding 0.7 (Extended Data Fig. 1b). To avoid overfitting of a model based on a suboptimal training dataset of XR/CS (Fig. 2a), we aimed to interpret the model instead of applying it directly on our test dataset. We learned from decision tree attributes (Extended Data Fig. 1c) that the sum and count of concordant negative s-scores are the most informative features, followed by the sum of discordant s-scores. In addition, if the count of concordant

negative s-scores was higher than the median count of concordant hits across all drug pairs ($n = 7$), the level of discordance would lose importance for classifying interactions. Based on this information we came up with the outlier concordance–discordance metric (OCDM), which discriminated previously reported CS and XR interactions from neutral ones (ROC AUC = 0.76 and 0.73, respectively; Fig. 2b,c, Source Data Fig. 2 and Methods), and selected the cutoff for extreme s-scores based on the OCDM performance (Extended Data Fig. 1d). We then used the OCDM cutoffs (Fig. 2c and Methods) to classify all possible interactions between the 40 antibiotics within the chemical genetics data[31] (the confusion matrix is shown in Extended Data Fig. 1e). This yielded 634 new drug-pair relationships (313 XR, 196 CS and 125 neutral), expanding the number of known XR and CS interactions by two and four times, respectively (Fig. 2d and Supplementary Table 2).

Based on the OCDM, drug pairs were inferred as XR if there was high concordance in the mutant profiles despite any discordance signal. In contrast, CS relationships required not only high discordance, but also no concordance signal in the chemical genetics profiles. The priority in concordance when defining interactions reflects the fact that XR-conferring mutations will dominate over CS mutations when a heterogeneous population, evolved in the first drug, is treated with a second drug. Overall, our metric does not classify interactions as exclusively XR or CS but rather reflects the frequency/strength of concordance or discordance of chemical genetics profiles of thousands of gene knockout mutants. In terms of previously measured drug pairs ($n = 206$), our metric agreed with 90 and disagreed with 116 of the previously identified interactions. 85/116 were previously identified as neutral interactions (Extended Data Fig. 2a–c) and we reasoned that these may be potential false negatives—akin to what was observed when comparing drug pairs across studies (Fig. 2a). This increased the total number of inferred drug-pair relationships to 840 (404 XR, 267 CS and 169 neutral) and expanded the number of known XR and CS interactions by three and six times, respectively (Extended Data Fig. 2d). A user-friendly Shiny app available at https://shiny-portal.embl.de/shinyapps/app/21_xrcs allows the user to browse the XR and CS interaction data per drug pair, class-based pair or genes of interest and includes views of drug class interactions.

## Chemical genetics-based metric accurately infers XR and CS

To benchmark our chemical genetics-based metric (OCDM) and cutoff decisions, we selected a subset of 38 newly inferred and 32 previously tested drug pairs (for 21/32, we predicted a different interaction than one previously reported) and measured their interactions using experimental evolution. In our setup we evolved resistance to 23 antibiotics in 12 lineages for up to about 50 generations (population bottleneck, approximately $2 \times 10^6$ cells; Methods) and tested resistant lineages for changes in susceptibility to a second antibiotic (Supplementary Table 3, Fig. 3a and Methods). Drug pairs were chosen to cover a wide OCDM range and to have low initial MICs to be able to evolve several-fold resistance. The number of antibiotic pairs belonging to the same chemical

class was limited ($n = 3$) to avoid inflating the prediction accuracy of XR predictions, as same-class drug pairs are likely to share resistance mechanisms. Evolving resistance to both drugs of each pair allowed us to assess the (bi)directionality of interactions, something that the OCDM score cannot assess. By definition, XR interactions are bidirectional and failure to detect them in both ways in experimental evolution experiments exemplifies the limitations of the method. In contrast, CS interactions can be directional, as resistance mechanisms for each drug of the pair can be different and not bear a fitness cost to the other drug. Hence, most of the previously detected CS pairs have been unidirectional. To decrease false negatives (that is, the failure to detect an interaction), we evolved resistance to a large number of lineages ($n = 12$), probed interactions in both directions and avoided strict cutoffs on the number of lineages required to exhibit an interaction to call drug pairs CS or XR (one was enough). As in our OCDM score, we considered XR interactions dominant to CS and non-monochromatic drug pairs (with lineages exhibiting both XR and CS) were deemed to be XR.

In total, we validated all but six of the inferred interactions, which amounts to a validation rate of 91.4%. Not only did we confirm all interactions for which previous studies and our metric agreed ($n = 11$) but also 18/21 interactions for which our predictions contradicted previous studies (Fig. 3b–d). This implies that several more of the 116 interactions that the OCDM metric classified differently from previous reports may be correct (Extended Data Fig. 2a–d). This high validation rate could be positively influenced by loss-of-function mutations typically dominating short evolution experiments, as the one we performed here (50 generations), and the OCDM score being based on chemical genetics data of an *E. coli* single-gene deletion library. To test whether longer experimental evolution would influence precision, we continued the evolution for a subset of drugs for 100 generations and probed 14 drug pairs for XR and CS, including three drug pairs that our shorter evolution experiment could not validate. Eleven of the 14 interactions agreed with the chemical genetics-based inferences despite individual lineages changing interactions with time and overall CS interactions decreasing during longer evolution (Extended Data Fig. 2e). Overall, chemical genetics could capture the results of experimental evolution regarding XR and CS well, although observed frequencies change with duration and strength of selective pressure.

The four published studies contained only 25 CS interactions. Here we inferred and validated 23 further CS interactions as well as two known ones (Fig. 3b). The majority of the validated CS interactions ($n = 19/25$) were bidirectional. The two non-monochromatic interactions that exhibited single instances of XR were classified as XR per our definition (Fig. 3b). This illustrates the power of chemical genetics to identify new CS interactions, especially the rare bidirectional ones, which are the most promising for cycling/combination therapies[8–15]. In contrast to CS drug pairs, about one-third of the tested XR pairs ($n = 11/38$), including those that were previously known, were non-monochromatic (Fig. 3d)—that is, some evolved lineages were sensitive, instead of resistant, to the second antibiotic. We failed to detect

**Fig. 3 | Inferred XR and CS interactions are validated with high accuracy by experimental evolution. a**, Schematic of benchmarking conducted for 70 drug pairs by experimental evolution and IC$_{90}$ measurements. Twelve lineages were evolved in parallel for five passages in increasing concentrations of 23 antibiotics. At each passage, the culture growing at the highest concentration was transferred to a new antibiotic gradient. The IC$_{90}$ of the final resistant population was then measured for all lineages in the relevant antibiotics. **b**–**d**, Heatmaps of 70 new and known drug-pair interactions, split depending on whether they were inferred as CS (**b**), neutral (**c**) or XR (**d**). Interactions were tested in both directions, with the drug for which selection occurred shown first and the drug for which MIC/IC$_{90}$ was tested shown second. In each interaction, all tested lineages are shown ($n = 9$–12). Coloured boxes denote the interaction observed for a given lineage. The three columns on the right of the lineage results represent the summary for all lineages. We considered an interaction as

validated if the log$_2$-transformed IC$_{90}$ fold change was >1 for XR and <−1 for CS in any direction tested for at least one lineage compared with the wild type. An interaction of a drug pair was deemed to be XR if there was at least one lineage showing XR despite any CS for other lineages. Interaction monochromaticity (that is, whether the interaction is exclusively CS or XR; neutral lineages do not affect this call and were labelled as not applicable (N/A)) and directionality (drug pair interacting consistently in both directions) are shown. Interactions referred to as reclassified in the text are those for which our inference and validation agree but previous reports have reported differently. The interaction in red (least monochromatic interaction) is used in Fig. 5 to understand the mechanisms in play. The interactions in bold are used later in Fig. 6 to test resistance evolution in drug combinations. The interaction in italics (drug pair 14), which was conflicting across studies (XR in one study and CS in another), has been inferred and validated to be CS.

the expected bidirectionality in nine XR cases and failed to detect the interaction after 50 generations of experimental evolution in four further cases (Fig. 3c); however, we detected the XR interaction for three out of four cases after 100 generations (Extended Data Fig. 2e). Overall,

the discovery of XR/CS using evolution experiments strongly depends on the experimental design and, given that frequencies change with experimental setup, calls are sensitive to strict thresholds and low numbers of lineages probed.

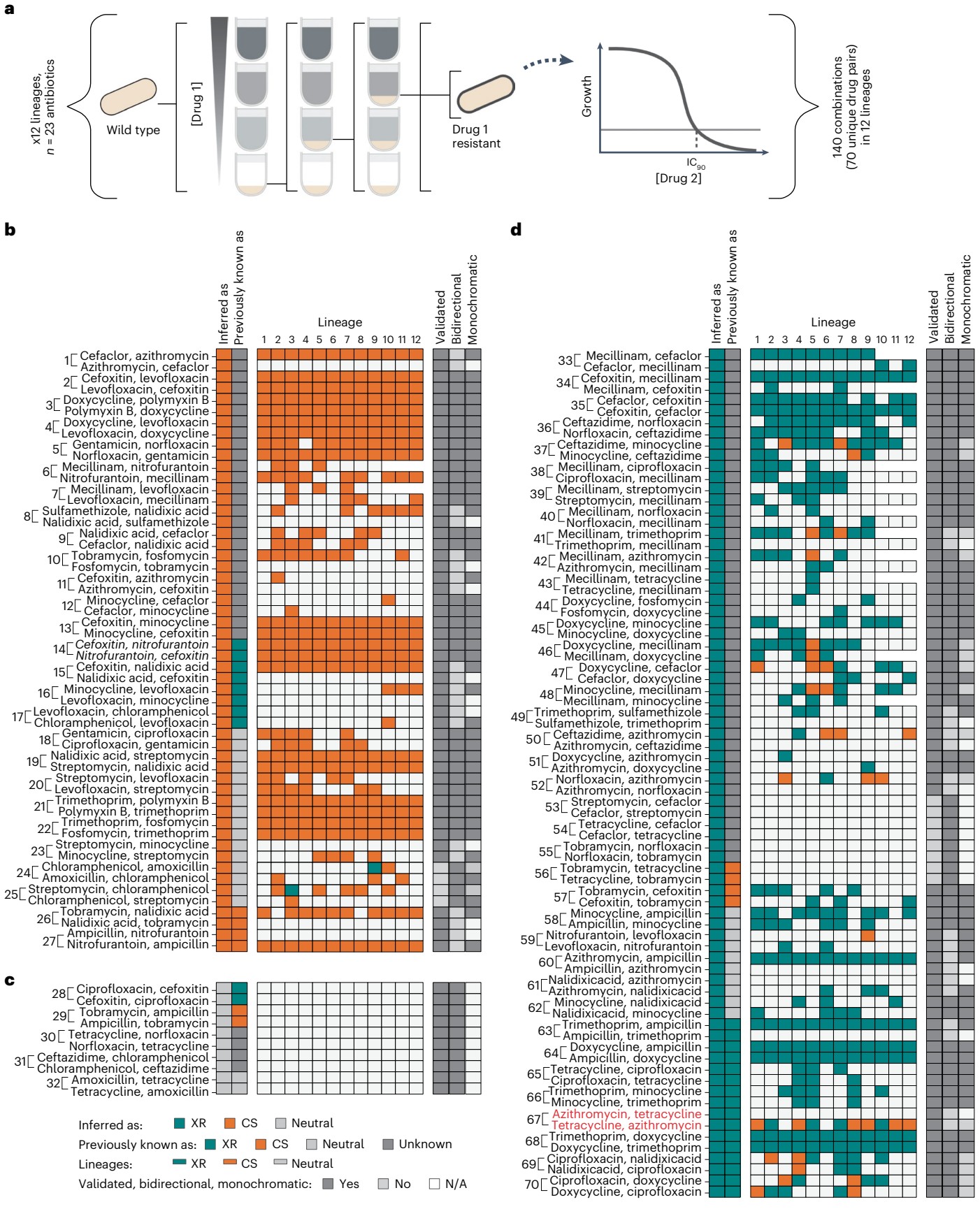

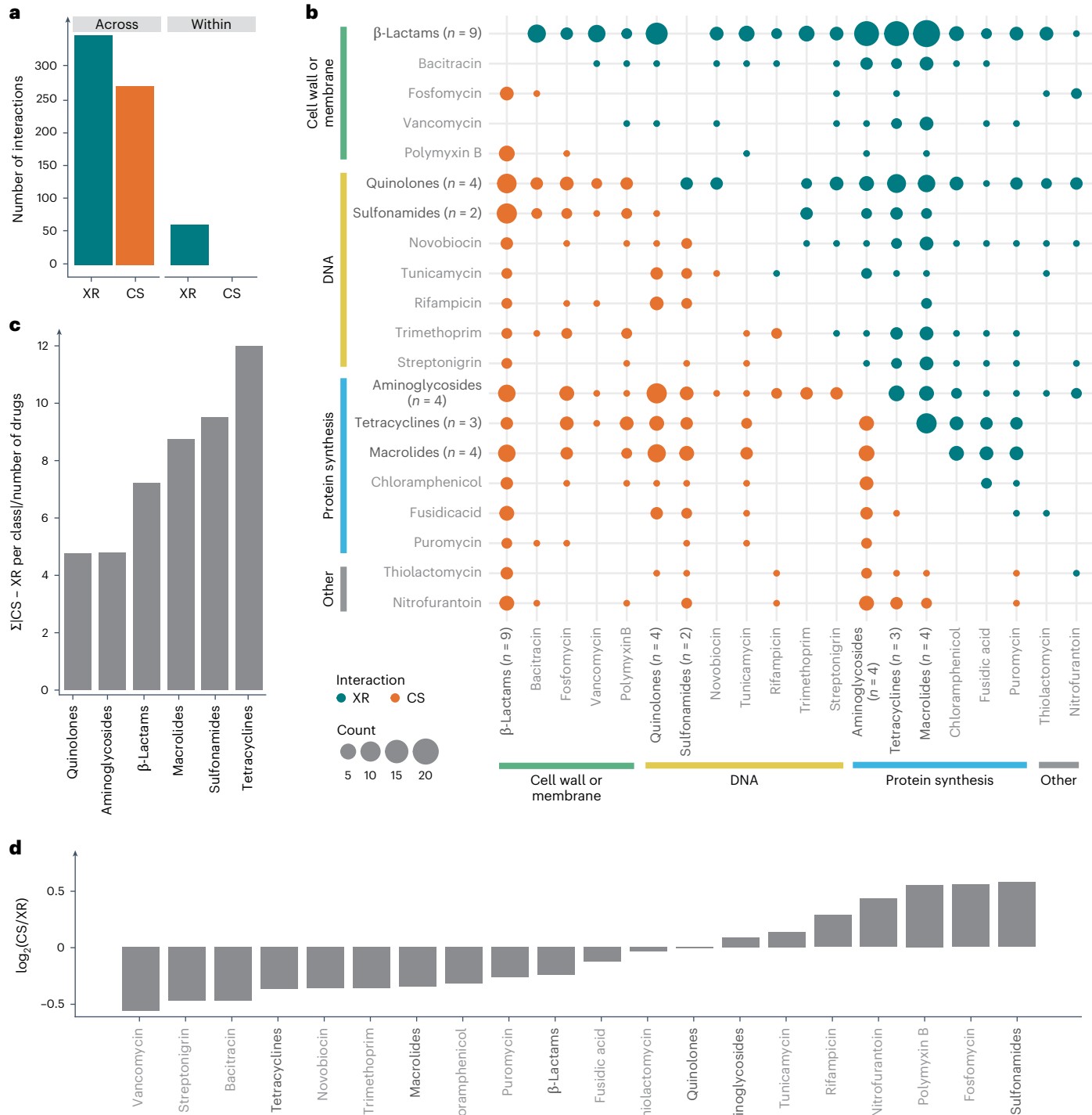

**Fig. 4 | CS and XR interactions between and within antibiotic classes.**
**a**, Interactions between members of the same antibiotic class (within class) are exclusively inferred as XR. The within-class group includes classes with more than one member probed—that is, β-lactams, aminoglycosides, quinolones, macrolides, tetracyclines and sulfonamides. **b**, Overview of all inferred and known drug interactions in *E. coli* at the class level. When a class has only one representative the antibiotic is named and shown in grey. The heat map sums XR and CS interactions across drug classes inferred by the OCDM metric. Within-class interactions are not displayed in the plot but are all exclusively classified as XR. Antibiotics are grouped according to their modes of action. Dot size represents the count of interactions between classes (or single antibiotics).

**c**, Coherency of interactions of each class with all other classes—that is, if all members of the class interact the same with other classes—calculated as the sum of the absolute differences between the number of XR and the number of CS interactions with each other class normalized to the number of drugs in the class. The higher the number, the more coherently the class is behaving. **d**, Interaction preference of each class (single- or multi-membered), calculated as the $\log_2$-transformed ratio of the number of CS and XR interactions with all other antibiotics from other classes. Antibiotic classes with a ratio of >0 are considered predominantly CS ($n = 8$), whereas those with a ratio of <0 as predominantly XR ($n = 12$). Antibiotic classes in bold are classes with more than one antibiotic tested.

## Antibiotic classes with extensive XR or CS

In contrast to other studies looking into CS and XR, where mostly one antibiotic per class is tested, here we could assess antibiotic class behaviours, as for some classes, several members were profiled in the chemical genetics data[31]. Antibiotics from the same class exhibited exclusively XR interactions, as they largely shared the mode of action and mechanisms of resistance (Discussion). In contrast, as previously reported[39], antibiotics of different chemical classes exhibited both XR and CS interactions (Fig. 4a), the former often driven by promiscuous resistance mechanisms (for example, efflux pumps) and the latter by mutations that lead to modifications of the outer-membrane composition (Extended Data Fig. 3). We next investigated whether antibiotic classes behaved coherently, that is, whether members of two classes interacted predominantly in the same way. Although this was true for antibiotic classes with members that share cellular target(s) and/or transport mechanisms (for example, tetracyclines and macrolides), this was less the case for classes with distinct targets (β-lactams) or transport mechanisms (quinolones of different generations; Fig. 4c). Interestingly, protein synthesis-inhibitor classes did not only act coherently but also exhibited mostly XR interactions between them (Fig. 4b), with the exception of aminoglycosides, which are known to be CS with drugs of different classes[8,17,19].

Besides aminoglycosides, the only other class reported to be enriched in CS interactions are polymyxins[8,17]. In addition to these two classes and nitrofurantoin, for which CS interactions have been reported before[17], we identified sulfonamides and several single drugs (fosfomycin, rifampicin and tunicamycin) with extensive CS interactions (Fig. 4b,d). Sulfonamides were largely CS to macrolides and β-lactams, driven by lipopolysaccharide (LPS)- and nucleotide biosynthesis-related mechanisms (Fig. 4b and Extended Data Fig. 3a). In contrast, protein-synthesis inhibitors (apart from aminoglycosides) were enriched in XR interactions, probably because of shared efflux resistance mechanisms (AcrAB–TolC; Fig. 4b,d and Extended Data Fig. 3b).

## Chemical genetics unravel CS and XR mechanisms

Understanding the mechanisms of XR and CS interactions from sequences of experimentally evolved strains is challenging, as passenger mutations occur in parallel to the causal mutation(s), and indirect mutations can also affect the expression/activity of causal resistance elements. The situation is even less obvious for CS interactions, for which very few mechanisms are known to date[7,17,19,26]. Chemical genetics make it easier to disentangle causality as all genes contributing to resistance or sensitivity to a certain drug are identified. To explore this, we first investigated how known CS interactions were represented in chemical genetics. For example, the decrease in proton motive force across the inner membrane decreases aminoglycoside uptake and makes cells more resistant to aminoglycosides, but also collaterally sensitive to drugs whose efflux is driven by proton motive force-dependent pumps, such as AcrAB–TolC[17,19]. Mutations in *trkH*, which encodes a proton-potassium symporter, were previously shown to cause this phenotype, in particular for the CS interaction between the aminoglycoside tobramycin and nalidixic acid or tetracycline[17,39]. The *trkH*

mutant, as well as mutants in subunits of the respiratory complexes[17,39], indeed exhibited discordant s-scores for these known CS drug pairs in chemical genetics (Extended Data Fig. 4a). Using the same logic, we tried to deduce the unknown mechanism of the recently described CS interaction between cefoxitin and novobiocin[26]. Genes involved in adding polarity to the LPS core—*waaG*, *waaP* and *waaQ*—were strongly discordant for this drug pair, leading to cefoxitin resistance and novobiocin sensitivity (Extended Data Fig. 4b). The outer-membrane penetration of novobiocin, a large lipophilic antibiotic, is known to be affected by LPS modifications[40,41]. At the same time, these mutations lower the levels of the outer-membrane porins OmpC and OmpF[42], allowing less cefoxitin and other cephalosporins to enter the cell[43].

Drug interactions can be non-monochromatic, as multiple resistance mechanisms exist for a given drug. Given that chemical genetics systematically explore the mutational space (of single loss-of-function mutations), we assumed that they should capture the dynamics of such interactions better. To assess this, we focused on XR drug pairs that exhibited non-monochromaticity in our validation experiment ($n = 11/38$; Fig. 3d). Antibiotic pairs with non-monochromatic XR interactions exhibited significantly stronger discordance scores in chemical genetics than drug pairs with monochromatic XR ($P = 1.00 \times 10^{-5}$; Extended Data Fig. 4c). Hence, chemical genetics can capture monochromaticity of XR interactions and potentially identify the antibiotic pairs that can evolve both XR and CS relationships (Extended Data Fig. 4d–g). We then investigated the most non-monochromatic pair in more detail, that is, tetracycline and azithromycin, which showed XR, CS and neutral interactions in four, six and two lineages, respectively (Fig. 3d). For each of our 12 tetracycline-evolved lineages, we measured changes in susceptibility to both antibiotics at each of the ten passages (Fig. 5a and Methods). Almost all lineages exhibited increased neutrality with time and as resistance to tetracycline increased, except for three lineages (lineages 1, 4 and 12), which evolved low resistance to tetracycline and remained CS to azithromycin (Fig. 5a). First, and as noted earlier, this could partially explain the low rates of CS and XR discovery in previous studies (Fig. 2a), given that XR and CS is typically assessed using final populations with high resistance to one drug. Second, it suggests that with time cells evolve more specific resistance mechanisms—for example, target- compared with intracellular concentration-related mechanisms.

To understand the mechanisms driving changes in the tetracycline–azithromycin relationship over time, we sequenced all 12 lineage populations from days 3, 5 and 7 (Extended Data Fig. 5). Lineages with neutral interactions carried either point mutations in tetracycline target genes (for example, lineage 3 with *rpsJ* V57L, coding for the S10 ribosomal protein[44]) or a combination of CS and XR strains in the population (for example, linage 7 with mutations in *hldE* and *marR*; Fig. 5a and Extended Data Fig. 5). Mutations in *marR*, which encodes a repressor of efflux pumps in *E. coli* and is a known modulator of antibiotic resistance[45,46], were behind all XR interactions observed in different lineages (lineages 2, 5, 7 and 10; Fig. 5a and Extended Data Fig. 5). This was consistent with *marR* deletion ($\Delta marR$) increasing resistance to both drugs in chemical genetics data (Fig. 5b). In contrast, all lineages with stable and strong CS interactions had promoter or deletion mutations

**Fig. 5 | Chemical genetics recapitulate the dynamics and explain the mechanisms of non-monochromatic interactions. a**, Changes in azithromycin susceptibility during the evolution of 12 lineages in tetracycline (100 generations, Methods). Resistance levels of 12 lineages to both antibiotics are shown for days 2, 3, 5, 7 and 10. Lineages are grouped according to whether they exhibited CS, neutrality or XR on day 5 (same as Fig. 3d). Dashed lines indicate the neutral threshold. **b**, Chemical genetic profiles of the *E. coli* deletion library in tetracycline and azithromycin[31]. Mutants with concordant (XR-related) and discordant (CS-related) profiles are highlighted. Dots in grey represent mutants that do not have s-scores within the 3% extreme cutoff for both drugs. Lines at $x = 0$ and $y = 0$ are shown to separate concordant and discordant zones of the plot. **c**, Mutations of lineage 11 during evolution. Genome sequencing of the lineage population

reveals a succession of two point mutations in genes that both lead to CS—first in *hldE*, which is then replaced by mutations in *waaF*, a slightly less detrimental gene for azithromycin resistance according to chemical genetics data in **b**. For the other 11 lineages see Extended Data Fig. 5. **d**, The fold change in tetracycline and azithromycin IC$_{90}$ of knockout mutants compared with the wild type confirms that both *hldE* and *waaF* contribute to resistance to tetracycline and sensitivity to azithromycin, whereas *ompF* deletion leads only to resistance to tetracycline; $n = 6$ biological replicates. **e**, Tetracycline uptake is reduced in a *waaF* deletion ($\Delta waaF$) mutant. Tetracycline fluorescence was measured in cell pellets and the signal was normalized to the optical density at 600 nm (OD$_{600nm}$); $n = 3–6$ biological replicates. **d**,**e**, Data are the mean ± s.e.m. **f**, OmpF, a major tetracycline importer, is the most downregulated protein in $\Delta waaF$[42]. FC, fold change.

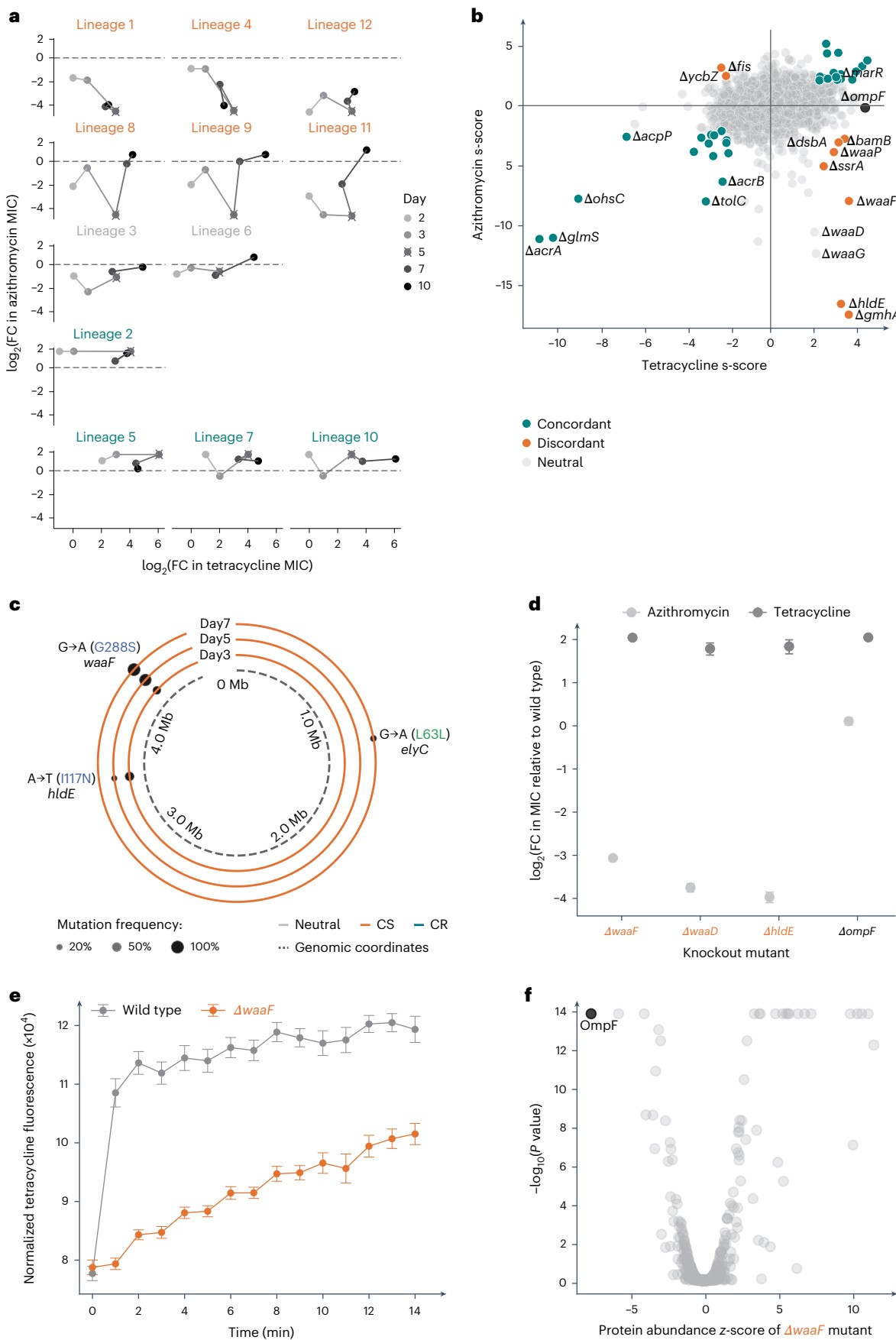

in *waaD* (Extended Data Fig. 5), one of the most sensitive mutants to azithromycin and resistant to tetracycline in chemical genetics data[31,47] (Fig. 5b). Lineages that were initially CS but became neutral (lineages 8, 9 and 11) carried initially strong CS mutations on *waaD* or *hldE* (both are involved in synthesis of the ADP–heptose precursor of core LPS), which were then replaced by strains with mutations in genes with milder CS or XR phenotypes, such as *waaF* and *marR* (Fig. 5b,c and Extended Data Fig. 5). We confirmed the slightly milder CS (lower azithromycin sensitivity) for the *waaF* deletion mutant ($\Delta waaF$), a gene encoding a protein that adds the second heptose sugar to the LPS inner core, compared with the *hldeE* and *waaD* deletion mutants (Fig. 5d). We postulated that the increased tetracycline resistance of all LPS core mutants is due to reduced uptake compared with the wild type and confirmed this by measuring intracellular tetracycline fluorescence in $\Delta waaF$ cells (Fig. 5e and Source Data Fig. 5e). This lower intracellular tetracycline concentration is probably due to low OmpF levels in $\Delta waaF$ cells (Fig. 5f)[42], as OmpF is the major tetracycline importer[43,48,49]. This is in agreement with chemical genetics data, where $\Delta ompF$ is tetracycline-resistant but not azithromycin-sensitive (Fig. 5b,d). Hence, loss-of-function mutations in *waaF* (or in other LPS core genes such as *hldE*, *waaD* and *waaP*) reduced the OmpF levels in the outer membrane and increased tetracycline resistance. At the same time, cells became more sensitive to azithromycin (and macrolides) because their outer membrane became less polar and thereby more permeable to hydrophobic antibiotics[50].

Overall, we confirmed that chemical genetics data can pinpoint CS and XR mechanisms that emerge and get selected during experimental evolution, thereby helping us to rationalize the dynamics of non-monochromatic antibiotic interactions.

## Combining CS antibiotic pairs reduces resistance evolution

Combination, sequential use and cycling of CS drug pairs reduce the rate of resistance evolution[8–15] and re-sensitize resistant strains[16] in laboratory settings. This has also been observed for a *Pseudomonas aeruginosa* infection in clinics[23]. Considering the therapeutic potential of CS antibiotic combinations, we tested the degree to which our newly identified CS pairs reduced resistance evolution in combination when compared with single drugs (Fig. 6a). We selected four CS pairs, two neutral pairs and one XR pair involving nine antibiotics. We evolved seven *E. coli* lineages to single drugs or combinations (using a 1:1 ratio compared with drug MICs) for seven days and measured the $IC_{90}$ of the evolved populations (Fig. 6a and Methods). For each antibiotic combination, we calculated 2,401 evolvability indices ($7^4$ combinations), that is, the degree by which resistance to any of the single drugs increases ($\log_2$(evolvability index) > 0) or decreases ($\log_2$(evolvability index) < 0) in the drug combination (Methods)[21]. As expected, lineages evolved in the presence of the ceftazidime–ciprofloxacin XR combination reached higher resistance to each drug

compared with lineages evolved with single antibiotic treatments (Fig. 6b and Source Data Fig. 6). In contrast, most lineages treated with CS or neutral combinations evolved lower resistance than those treated with single antibiotics (Fig. 6b). The strongest reduction in resistance evolution occurred for combinations of bidirectional CS pairs (Figs. 3c and 6b). For example, six of seven lineages evolved full resistance towards mecillinam alone (256-fold increase in MIC), whereas the combination of mecillinam with nitrofurantoin or levofloxacin led to almost no mecillinam resistance (average fold change in $IC_{90}$ < 2). For the cefoxitin–levofloxacin pair, resistance evolved in combination was lower just for cefoxitin (Fig. 6b and Extended Data Fig. 6), despite the pair showing bidirectional CS during experimental evolution (Fig. 3c). Together, we demonstrate that reciprocal CS antibiotic pairs hold a great potential for diminishing resistance evolution when used in combination.

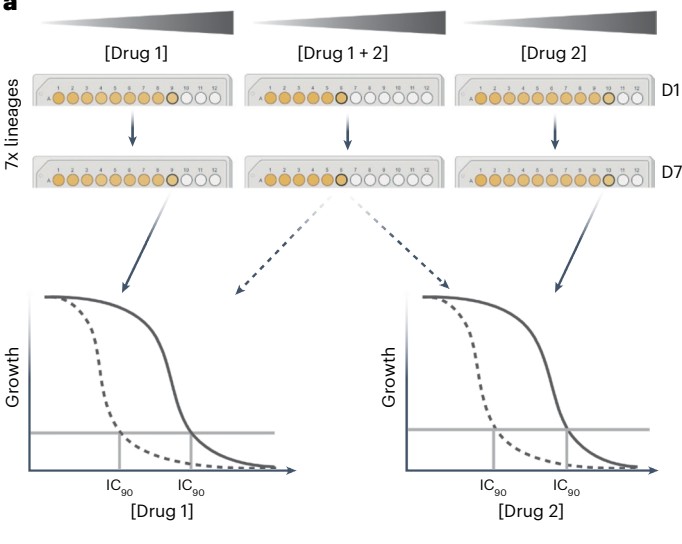

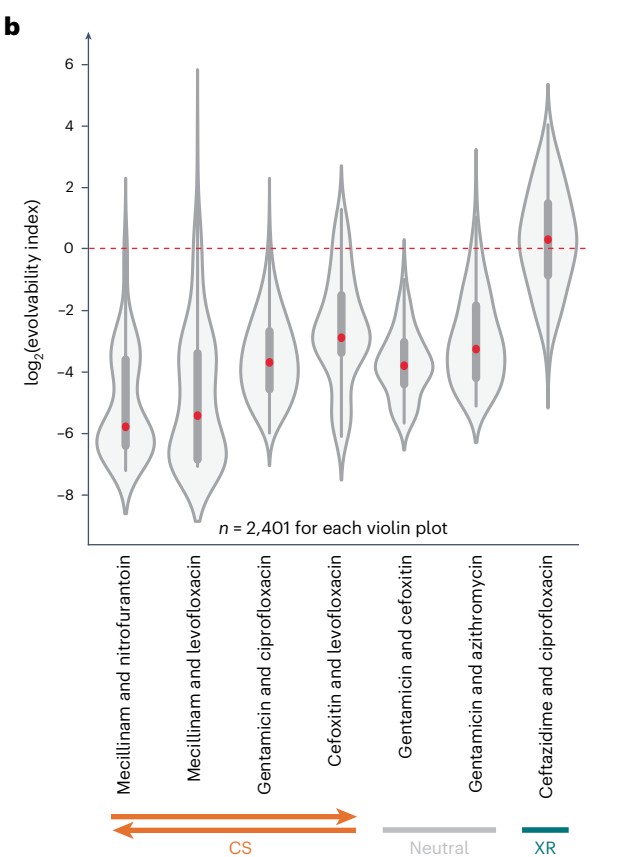

**Fig. 6 | Combinations of reciprocal CS antibiotic pairs reduce resistance evolution. a**, Experimental design. After evolving resistance to single antibiotics or their combination (seven lineages for each, passaged every 24 h for 7 d; 70 generations in total), the $IC_{90}$ of both antibiotics was determined for the evolved mutants. In each passage mutants growing (coloured yellow) at the highest concentration (well marked by a thick circle) were transferred (Methods). **b**, The measured $IC_{90}$ values were used to calculate the evolvability index (equation (2), Methods; data using slightly different original evolvability indices (equation (3), Methods) in Extended Data Fig. 6a). The red line represents the cutoff ($\log_2$(evolvability index) = 0; the evolvability index was $\log_2$-transformed to make data symmetrical) below which the antibiotic pair is considered to reduce resistance evolution compared with single antibiotics. Red dots on the violin plots represent the median. The box boundaries represent the first and third quartiles, with the median indicated. The whiskers extend to the furthest data points within 1.5× the interquartile range. Non-XR antibiotic combinations led to lower collective resistance and in the case of reciprocal CS to lower evolvability indices and lower resistance to each of the antibiotics combined (Extended Data Fig. 6b).

## Discussion

A better understanding of how resistance to one antibiotic limits treatment with others (XR) or opens new opportunities (CS) is imperative in the context of the ongoing antimicrobial resistance crisis. In the last decade such drug interactions have been assessed for several pathogens[8,12,16–18,21–25,51]. However, the main detection method, experimental evolution, has limitations. First, it has low sensitivity, which leads to different studies reporting different interactions for the same drug pairs in the same species (Fig. 2a). This is because only limited numbers of lineages and resistance mechanisms are probed. What further augments the problem is that resistance mechanisms largely depend on the amount and time of selective pressure applied, as we show for the tetracycline–azithromycin pair (Fig. 5a) and for the 13 further drug pairs probed for both 50 and 100 generations (Extended Data Fig. 2e). In addition to these inherent limitations each study uses different selection pressures, metrics and number of lineages to assess interactions. Although within-species comparisons are possible when the metric and selection pressure are standardized[52], cross-species comparisons become challenging as it is unclear whether the differences in interactions stem from biological (different resistance mechanisms) or technical reasons (false negatives, more difficult to standardize selective pressure across species). Second, experimental evolution is laborious and limits the number of drug pairs that can be tested. As a result, the monochromaticity of interactions (especially for drug classes) has been challenging to assess properly in the past. Last, it is very hard to identify the underlying mechanism for CS and XR interactions by only sequencing resistant lineages and without additional tailored experiments.

By assessing the impact of thousands of individual mutations at once on resistance or sensitivity to different drugs, chemical genetics can bypass most of these limitations. As we show here chemical genetics offer a way to systematically and quantitatively assess all chromosomal resistance mechanisms (independent of selective pressure) and can dramatically increase the throughput of bacterial species and drugs tested. In addition, it can provide insights into how monochromatic or conserved such interactions are as well as a basis to dissect the driving mechanisms. As proof-of-principle we focused on published chemical genetics data from *E. coli*[31] because of the large number of antibiotics screened at different concentrations and the extensive benchmarking. In the future similar analyses can be expanded to other available datasets in the same or other species[34,47,53–56], but the OCDM metric may need to be fine-tuned and/or retrained, especially if the fitness metric and dynamic range of the data are different. Such datasets will inevitably increase with time as genome-wide mutant libraries are becoming available for tens of species and even more strains[57,58]; these can be arrayed or pooled[29,59] and constructed by targeted deletions[60–62], transposon insertions[59,63] or CRISPRi knockdowns[53,64]. Including such libraries will allow probing of the role of essential genes and/or gene overexpression when mapping antibiotic resistance and XR/CS relationships. An obvious limitation of our current metric is that it is based on single loss-of-function (deletion) mutations of non-essential genes. During evolution (in the laboratory or in patients) resistance does not only arise by frequent loss-of-function mutations in non-essential genes but also by less frequent gain-of-function mutations (via point mutations, insertions or duplications) and by mutating essential genes (for example, antibiotic target). Moreover, epistatic relationships between multiple mutations can affect both resistance and XR/CS to other drugs. As global epistasis maps[65] become more common in bacteria in the future, such data could make XR and CS inferences even more robust.

In this study we devised an approach and metric to map CS and XR in *E. coli* using available chemical genetics data for 40 antibiotics. We thereby increased the number of known interactions by several-fold, validated previous conflicts in literature and proposed different interactions for 116 drug pairs reported mostly neutral (*n* = 85) by single studies

before (18 were further validated by experimental evolution). Beyond this we obtained unique insights into within-class interactions, unravelling that all antibiotic classes are dominated by XR interactions between their members. Although this is largely expected, some classes have members with non-overlapping targets and/or resistance mechanisms. Specifically for β-lactams, their use in combination has been reported to constrain resistance evolution during fast-switching regimens[66] or for specific pairs and resistance mechanisms[67]. Moreover, we identified many new bidirectional CS interactions and used a handful to show that the evolution of antibiotic resistance to combinations of such antibiotics is harder. In the past evolutionary variability and non-monochromaticity of CS interactions has been identified as a bottleneck for their use in clinics[68,69]. It remains to be seen if the ability to identify monochromatic and bidirectional CS drug pairs alleviates some of these limitations. Finally, we mechanistically rationalized CS interactions and explained why some drug interactions can be non-monochromatic. In the case of tetracycline–azithromycin, the mechanisms that played a role in experimental evolution were a small subset of the possible mechanisms revealed by chemical genetics. This is probably because only 12 lineages were probed but also likely to be due to the fitness costs of some of these resistance mechanisms. Interestingly, the interaction changed non-monotonically over time and longer/stronger selection on one drug (tetracycline) led to more neutral interactions with the second drug (azithromycin). This means that long-term, bacterial populations may opt for target mutations or low/neutralized fitness-cost resistance mechanisms, neutralizing also CS/XR interactions. Hence, fast-switching or combinatorial treatments may be more efficient than sequential antibiotic treatments for CS drug pairs.

The increased ability to map XR and CS interactions between drugs opens the path for future expansion of such endeavours to non-antibiotics with antibacterial or adjuvant activity[70–72] and to probing interactions in different environments—such as in bile, different pHs[73], urine media, biofilms[74] or gut microbiome communities—as fitness costs are known to change with the environment[75]. Moreover, the systematic nature of chemical genetics limits false negatives and metric biases and can allow for comprehensive comparisons across species and strains using corresponding genome-wide mutant libraries. Cross-species studies have been conducted previously to map drug synergies and antagonisms[35,76]. Knowledge on how drugs interact at multiple levels—resistance evolution, efficacy, long-term clearance effects[77] and host cytotoxicity—will open the path for designing better combinations for the clinical setting.

## Methods

### Data sources and preprocessing

The *E. coli* chemical genetics data were obtained from a previous study[31] in which the fitness of 3,979 non-essential single-gene knockout mutants and essential gene hypomorphs was evaluated in 324 different conditions (114 unique stresses and drugs tested in different concentrations). Fitness effects were quantified as s-scores, that is, a modified *t*-statistic on the deviation of the colony size of one mutant in one condition from the median colony size of the mutant across all conditions[38,78]. We reprocessed the data to exclude the following: (1) strains from the hypomorphic mutant collection and mutants that had ≥10 missing values for the conditions, reaching a final number of 3,904 mutants, and (2) environmental stresses (for example, different temperatures, pH, heavy metals, amino acids, dyes and alternative carbon sources), non-antibiotic drugs and drug combinations. Antibiotics with a narrow range of s-scores (no extreme s-scores, that is, <−6.9 or >3.9) were also excluded from the analysis (*n* = 7). This left us with 40 antibiotics that were further used in this study (Supplementary Table 1). For those antibiotics tested in multiple concentrations, the highest was selected.

Previously reported XR and CS interactions were collected from four studies. Lazar and colleagues[17,18] measured XR and CS in *E. coli*

BW25113 using 12 antibiotics where interactions were defined based on a difference of at least 10% in the growth of more than 50% evolved lineages compared with control lineages. Oz and colleagues[19], and Imamovic and Sommer[8] compared the MICs of evolved populations with the wild type to define XR and CS in *E. coli* MG1655 using 22 and 23 antibiotics, respectively. We kept the original definitions and assessments of XR and CS used in the respective studies. When integrating these datasets, interactions of overlapping antibiotic pairs were annotated as 'XR and neutral', 'CS and neutral', 'XR and CS' and 'XR and CS and neutral' if conflicting interactions were observed in different studies. Interactions with 'XR and CS' and 'XR and CS and neutral' annotations were removed ($n = 6$) and 'XR and neutral' and 'CS and neutral' were re-annotated as 'XR' and 'CS', respectively, because evolution experiments are prone to false negatives. Directionality was reduced (keeping drug 1–drug 2 but removing the reciprocal) by removing one pair (if XR/CS was bidirectional) or the 'neutral pair' (if the interaction was unidirectional). After the preprocessing steps, only conditions for which chemical genetics data were available were selected as the training set ($n = 24$), amounting to 111 neutral, 70 XR and 25 CS drug-pair relationships (Supplementary Table 3).

### Assessment of correlation metrics

Given that the first attempts at combining chemical genetics profiles and XR/CS interactions found associations between the chemical genetics profile similarity and XR/CS[17,18], we assessed several correlation methods from SciPy (v1.12.0)[79] to compute various correlation coefficients between two drugs (drugs 1 and 2; Extended Data Fig. 1a). The correlation functions were applied to drug pairs with known interactions for which chemical genetics data are available. For each drug pair in this dataset, the correlation coefficient was computed using the four methods (Pearson, Spearman, Kendall's tau and weighted tau). We plotted ROC curves to evaluate the performance of the computed correlation coefficients in distinguishing between interaction types (XR ($n = 70$) versus non-XR ($n = 136$) and CS ($n = 25$) versus non-CS ($n = 181$)). The correlation coefficients served as the predictor values and the interaction types (either XR or CS) were the true labels. The ROC AUC was computed for each correlation method (Extended Data Fig. 1a).

### Feature generation and interpretation of decision trees

For each condition in the chemical genetics data, 3% extreme positive and negative s-scores were chosen after assessment of different cutoffs (Extended Data Fig. 1d). Six features were generated by antibiotics pairwise calculation: sum of positive concordant s-scores, sum of negative concordant s-scores, sum of discordant s-scores, count of positive concordant s-scores, count of negative concordant s-scores and count of discordant s-scores. Using these features, machine-learning algorithms (based on decision trees[80]) were used and models were trained to classify XR ($n = 70$) versus non-XR ($n = 136$) and CS ($n = 25$) versus non-CS ($n = 181$).

To address the class imbalance, the minority class was oversampled to match the size of the majority class. A search space for hyperparameters was defined for the decision tree classifier, including the function to measure the quality of a split, the maximum depth of the tree, the minimum number of samples required to split an internal node and the minimum number of samples required to be at a leaf node. A fivefold grid search cross-validation always excluding the test set from the training set, stratified to maintain the same proportion of the target class as the entire dataset, was used to find the best hyperparameters for the decision tree classifier based on the F1 score. The resulting classifier was trained and again evaluated on the balanced dataset using cross-validation. The best classifier according to the F1 score, precision, recall and ROC AUC was then fitted to the balanced dataset.

The trained decision tree classifier was graphed, showing the decision paths and splits. The tree visualization was limited to a depth of three for clarity (Extended Data Fig. 1c). We learned from decision tree

classifiers that if the count of concordant negative s-scores was higher, the level of discordance was not important to classify interactions. The sum and count of concordant negative s-scores were found to be the most important features, followed by the sum of discordant s-scores. This information was used to generate the OCDM metric, described in detail in the following section. Classifier training, hyperparameter tuning and visualization were implemented using the scikit-learn package (v1.1.3)[81].

### Metric generation and interaction measurement

Among the correlation methods, six chemical genetics-derived features and their engineered combinations, we identified the OCDM as the best metric to separate statistically significantly XR, neutral and CS interactions (Fig. 2c). The OCDM metric is defined as the difference between the sum of concordant s-scores and the sum of discordant s-scores if the count of concordant s-scores ($N_c$) is lower than the median count as shown below. Otherwise, OCDM is simply the sum of concordant s-scores.

$$\text{OCDM} = \begin{cases} \sum C - \sum D & , \text{if } N_c < \text{median} N_c \\ \sum C & , \text{else} \end{cases} \tag{1}$$

where $C$ represents concordant s-scores and $D$ represents discordant s-scores. To identify optimal threshold determination (cutoffs) of OCDM, the false-positive (FPR) and true-positive (TPR) rates were used to calculate the true factor (TF = TPR − (1 − FPR) = sensitivity − specificity), which was computed for each threshold. This threshold represents the best trade-off between sensitivity (TPR) and specificity (1 − FPR), which are >105.159057 (to define XR) and <27.224792 (to define CS).

All data analyses were performed in Python (v3.9.17).

### Bacterial strains and growth conditions

For all experiments, and unless otherwise specified, *E. coli* (strain BW25113) or single-gene knockouts in this strain[60] were cultured in LB Lennox broth (tryptone 10 g l$^{-1}$, yeast extract 5 g l$^{-1}$ and sodium chloride 5 g l$^{-1}$) at 37 °C and fully aerobically (850 rpm) or on agar (2%) plates (same medium and temperature).

### MIC (IC$_{90}$) determination

Overnight cultures of *E. coli* BW25113 were diluted to OD$_{600mn}$ = 0.001 and cultured with antibiotics (Supplementary Table 1) at eight concentrations in a twofold dilution gradient, in two technical replicates in microtiter plates (U-bottomed 96-well plates; Greiner Bio-One, 268200) at 37 °C with continuous shaking (850 rpm; orbital microplate shaking). The plates were sealed with breathable membrane (Breathe-Easy; Sigma-Aldrich, Z380059-1PAK) and the OD$_{600nm}$ was measured every 30 min for 24 h using the BioTek Gen5 (v3.02.2) and SoftMax Pro 7.1 software. The liquid handler Biomek FX (Beckman Coulter) was used to prepare plates. All MIC tests were performed in a total volume of 100 µl per well. Controls included 'no cell + no drug' controls to assess contamination, 'no drug' controls to assess maximal growth and 'no cell' controls to assess artefacts (OD$_{600mn}$ change) of the drugs alone or their interaction with medium components. The AUC was calculated using the simps function from SciPy (v1.12.0)[79] and divided by the no drug control. Across the study, the MICs were defined as the IC$_{90}$, which was calculated using the drc (v0.5.8) package in R (v.4.1.2)[82].

### Experimental evolution and XR/CS measurements

Overnight cultures of wild-type *E. coli* were diluted 1:1,000 and exposed to eight concentrations −from 0.5× IC$_{90}$ to 64× IC$_{90}$−of 23 antibiotics in 12 lineages using the same volumes and plates as for MIC determination. Every 24 h the lineages growing in the highest concentration (OD$_{600nm}$ > 0.3) were back-diluted to OD$_{600nm}$ = 0.01 and the volume needed to reach a final dilution of 1:1,000 (3–10 µl) was transferred to the next plate with the same concentration gradients. Once the

evolution experiment was completed (five passages for a total of five days; approximately 50 generations in total), the lineages were tested for antibiotic susceptibility for 70 of the 634 predicted interactions (11%; 30 novel XR, eight known XR, 25 novel CS, two known CS, four novel neutral and one known neutral interaction; Fig. 3b–d and Source Data Fig. 3). The $IC_{90}$ values were determined as in 'MIC determination' (12 lineages or populations × 140 combinations (70 unique drug pairs) × 2 technical replicates = 3,360 $IC_{90}$ values; Source Data Fig. 3c–e). Changes in $IC_{90}$ were compared with the ancestor strain. Interactions were defined as XR or CS if $\log_2$(fold change) > +1 or −1, respectively. For 14 drug pairs, we performed five more passages (total of ten passages; approximately 100 generations) and measured the changes in $IC_{90}$ again (Extended Data Fig. 2e). In the case of the azithromycin–tetracycline pair, we tracked changes both in tetracycline resistance and azithromycin susceptibilities across multiple generations.

### Whole-genome sequencing and analysis

A clone from the wild type and from populations of 12 lineages from days 3, 5 and 7 were sequenced to determine mutations responsible for the given phenotype. Genomic DNA was extracted using a Macherey Nagel DNA extraction kit and sequenced using single-end Illumina NextSeq 2000 (P1; length of 122 bp). Mutations were identified by mapping sequences to the reference genome from the NCBI database (*E. coli* BW25113 strain K-12 chromosome; GCF_000750555.1)[83] using Breseq[84] with the following parameters: -p -l 80 -j 8 -b 5 -m 30. Mutations present in the wild-type clone compared with the NCBI reference genome were eliminated to only identify mutations that are associated with resistance/sensitivity.

### P1 transduction

Single colonies of *E. coli* wild type (BW25113) and the corresponding Keio mutants[60] were used for P1 transduction. P1 lysate preparation and transduction were performed as previously described[85]. We confirmed the transduction success with colony PCR.

### Tetracycline fluorescence assay

Wild-type *E. coli* and *waaF*- *waaD*- and *hldE*-knockout mutants were cultured in 5 ml LB with continuous shaking at 37 °C until they reached an $OD_{600nm}$ of 0.5. Aliquots (1 ml) of each culture were centrifuged at 3,500 rpm (1,300*g*) for 10 min and the supernatants were discarded. The pellets were washed three times with 0.5 ml of 137 mM PBS, resuspended in 50 µl of 137 mM PBS and transferred to black-walled, clear- and flat-bottomed 96-well plates (Greiner Bio-One, 655096) containing three concentrations of twofold serially diluted tetracycline (highest final concentration, 16 µg ml$^{-1}$; final volume, 100 µl per well). Both the $OD_{600nm}$ and fluorescence (excitation λ, 405 nm; emission λ, 535 nm) were measured with an Infinite M1000 PRO plate reader (Tecan i-control (v1.10)) for 15 min, with readings taken every minute. Three to six biological replicates were conducted for each experiment.

### Experimental evolution against antibiotic combinations

The $IC_{90}$ values for individual antibiotics ($n = 8$) and drug combinations at a 1:1 $IC_{90}$ ratio ($n = 7$) were measured as in 'MIC determination'. The evolution experiment was carried out in the same way as described in 'Experimental evolution and XR/CS measurements' with the following changes: the initial wild-type culture was exposed to 11 concentrations—from 0.125× $IC_{90}$ to 128× $IC_{90}$—of eight single antibiotics and seven antibiotic combinations for seven lineages. At the end of the experiment (seven passages for a total of seven days; approximately 70 generations), the $IC_{90}$ values of drugs 1 and 2 were measured in drug 1-, drug 2- and drug 1 + 2-resistant lineages as described in 'MIC measurements'. To compare the evolution of resistance to single drugs versus drug combinations, evolvability indices were calculated using the average of the $\log_2$-transformed $IC_{90}$ ratios of two drugs for each possible pair (2,401 values per antibiotic combination) as:

$$\text{Evolvability index} = \frac{1}{2} \times \left( \log_2 \left( \frac{IC_{90}(\text{drug1})_{\text{drug1+2}}}{IC_{90}(\text{drug1})_{\text{drug1}}} \right) + \log_2 \left( \frac{IC_{90}(\text{drug2})_{\text{drug1+2}}}{IC_{90}(\text{drug2})_{\text{drug2}}} \right) \right), \quad (2)$$

where $IC_{90}(\text{drug1})_{\text{drug1+2}}$ corresponds to the $IC_{90}$ of drug 1 for the lineage evolved against the drug 1 + 2 combination. We calculated evolvability indices using the modified equation, an average of the $\log_2$-transformed $IC_{90}$ ratios of two drugs, different from previously defined (equation (3))[21] as it assesses the effects of combining drugs on resistance to each drug separately.

$$\text{Evolvability index} = \frac{1}{2} \times \left( \frac{IC_{90}(\text{drug1})_{\text{drug1+2}}}{IC_{90}(\text{drug1})_{\text{drug1}}} + \frac{IC_{90}(\text{drug2})_{\text{drug1+2}}}{IC_{90}(\text{drug2})_{\text{drug2}}} \right) \quad (3)$$

### Reporting summary

Further information on research design is available in the Nature Portfolio Reporting Summary linked to this article.

### Data availability

All supplementary data are provided in Supplementary Tables 1–3. A reference genome from the NCBI database (*E. coli* BW25113 strain K-12 chromosome, GCF_000750555.1) was used. Raw reads of sequenced samples (file names describe samples) are available via Zenodo at https://doi.org/10.5281/zenodo.10572857 (ref. 86). All data are included in the Shiny app at https://shiny-portal.embl.de/shinyapps/app/21_xrcs. Source data are provided with this paper.

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

## Acknowledgements

We thank M. Galardini, A. Koumoutsi and the EMBL Gene Core for help with sequencing experiments; A. Koumoutsi, L. Y. Yong and S. Bassler for experimental advice as well as the Typas laboratory, especially K. Mitosch, for fruitful discussions. This work was supported by the ERC consolidator grant uCARE (ID 819454) and EMBL internal funding to A.T.

## Author contributions

A.T. and N.S. conceived and designed the study. A.T., C.V.G., E.C., P.B. and J.M. supervised the project. All scripts were written by N.S., with advice on data preprocessing from F.H., machine learning from A.O. and MIC determination from V.V. All experiments were carried out by N.S. with advice from E.C. and C.V.G. Figures were designed and plotted by N.S. with input from E.C., A.O., C.V.G. and A.T. N.S. and A.T. wrote the manuscript with input from all authors. All authors approved the final version.

## Funding

## Competing interests

The authors declare no competing interests.

## Additional information

**Extended data** is available for this paper at https://doi.org/10.1038/s41564-024-01857-w.

**Correspondence and requests for materials** should be addressed to Camille V. Goemans or Athanasios Typas.

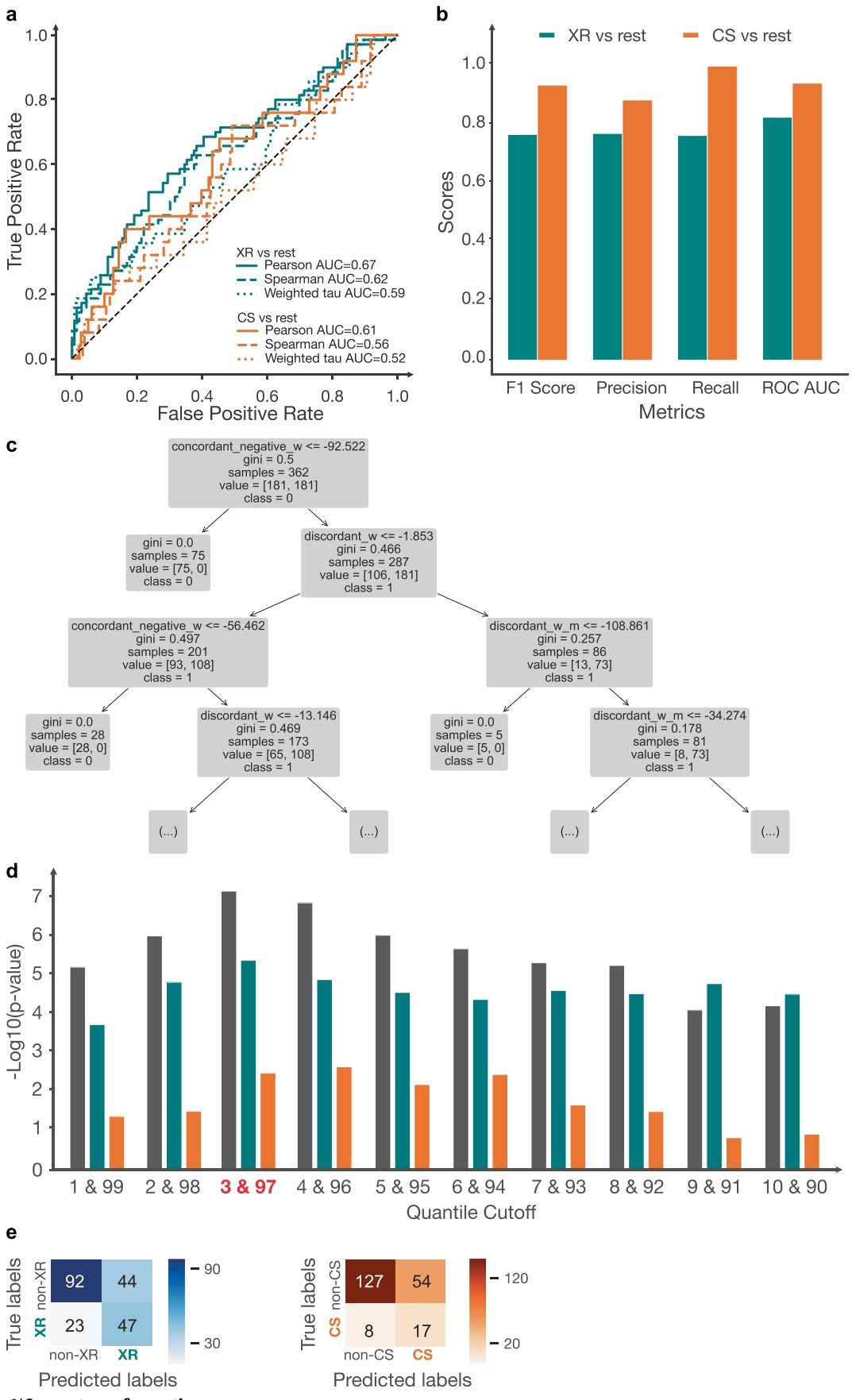

**Extended Data Fig. 1 | See next page for caption.**

**Extended Data Fig. 1 | Performance of different metrics and models in capturing XR and CS antibiotic interactions from chemical genetics data.** **a**, Receiver operating characteristic (ROC) curves for classification of XR (positive class) vs non-XR (negative class), and CS (positive class) vs non-CS (negative class), using simple linear and non-linear correlation metrics. AUC is the area under the curve. **b**, The performance of the decision tree model on balanced classes shows that both XR and CS interactions can be well classified. **c**, Decision tree with classes CS (class 1) versus the rest (class 0), where a maximum depth of 3 is shown for visualization, illustrates the hierarchy of decisions to discriminate classes. Each node in the tree represents a decision point based on the value of a particular feature, and branches represent the outcome of the decision. The root node divides the data based on the 'concordant_negative_w' feature, which is the sum of s-scores (as weights) of hits on the negative concordant site of a scatterplot. The tree branches out to 'discordant_w' feature, which is the sum of s-scores (as weights) of hits on the discordant site of a scatterplot, while 'discordant_w_m' is the sum of products of s-scores (as weights) of hits on the discordant site of a scatterplot. **d**, *P* values from a paired Mann–Whitney U-test (two-sided) are depicted across quantile cutoffs for extreme s-scores to differentiate XR/CS/neutral interactions based on OCDM values. Q3 and Q97 perform the best. **e**, Confusion matrix of results based on Q3 and Q97. Most interactions inferred as non-XR/non-CS were previously reported neutral. For more information, see also Extended Data Fig. 2a-c.

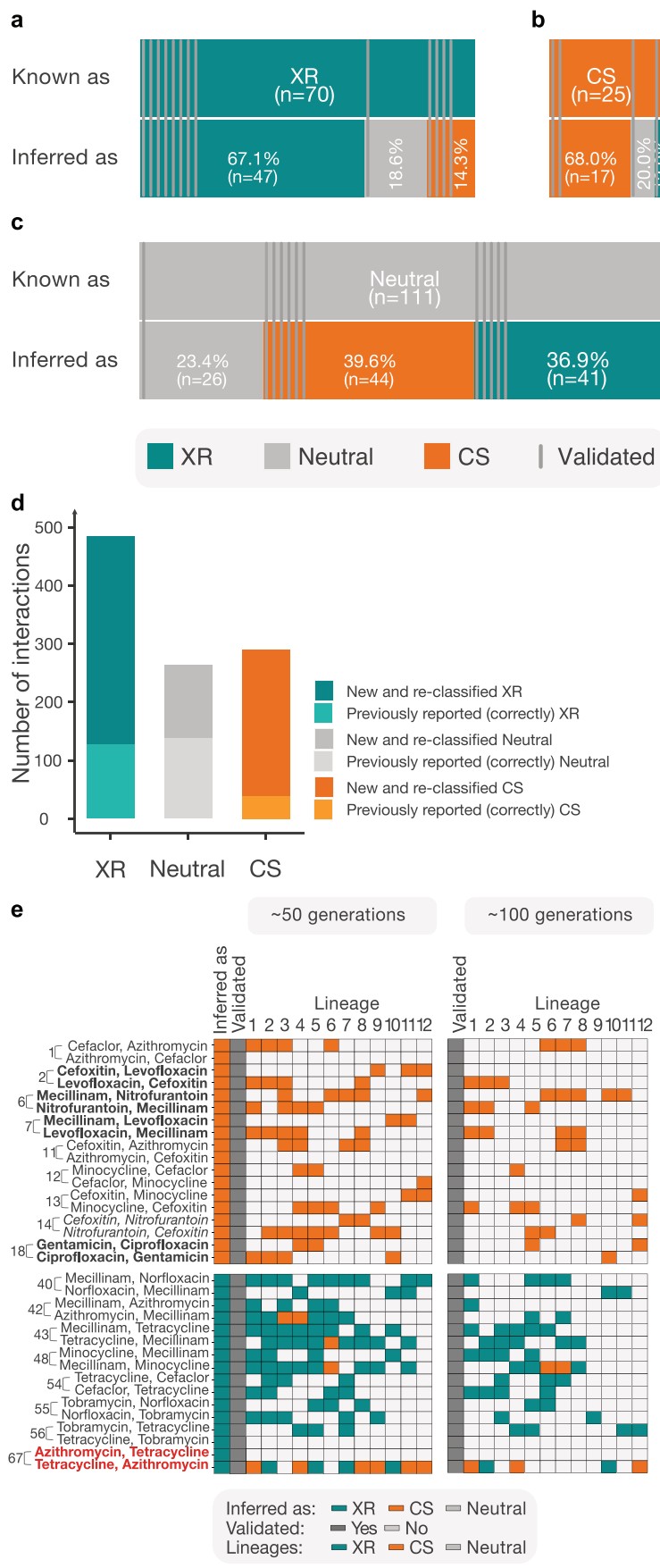

**Extended Data Fig. 2 | See next page for caption.**

**Extended Data Fig. 2 | Chemical genetics metric captures well prior information and can be used to reclassify a subset of prior interactions.**
**a,b**, Comparison of previously reported XR (**a**) and CS (**b**) interactions with our inferences based on our chemical genetics metric (OCDM) show an agreement of 67–68% for CS (n=17) and XR (n=47) - 10 such interactions were validated experimentally during our benchmarking (Fig. 3b,d). The rest is inferred as neutral or the opposite interaction by OCDM, including seven interactions (4 CS, 2 neutral & 1 XR) that we experimentally validated that OCDM inference was correct (Fig. 3b,d). **c**, In contrast to CS or XS, there is less agreement for neutral interactions with previous studies. This is consistent with the high false negative rates when comparing prior studies between them (Fig. 2a). The majority of previously reported neutral interactions (76.6%, n=85) are inferred as CS/XR by chemical genetics. 11/13 we included in the benchmarking set were confirmed as inferred by OCDM. The other two were inferred CS, but although most lineages exhibited CS, a single lineage exhibited XR, and hence called XR (Fig. 3b–d). **d**, New XR, neutral, and CS pairs inferred by chemical genetics and the OCDM cutoff are 2.8- and 6.4-fold more than currently known XR and CS antibiotic interactions in *E. coli*, after reclassifying interactions (n = 116) we infer differently than previously reported. The plot includes known interactions for which chemical genetics data is not available. **e**, Resistance against 12 antibiotics was evolved again for up to ~100 generations in 12 lineages. The MIC of evolved populations was measured at ~50 and ~100 generations for the same lineages in different antibiotics, allowing us to assess XR/CS for 17 drug pairs in both directions. Data are represented and drug pairs are numbered as in Fig. 3b,d. All inferred interactions were validated at both ~50 and ~100 generations. The length of experimental evolution affected the XR/CS of individual lineages and to a lower degree the cumulative call of the drug pair.

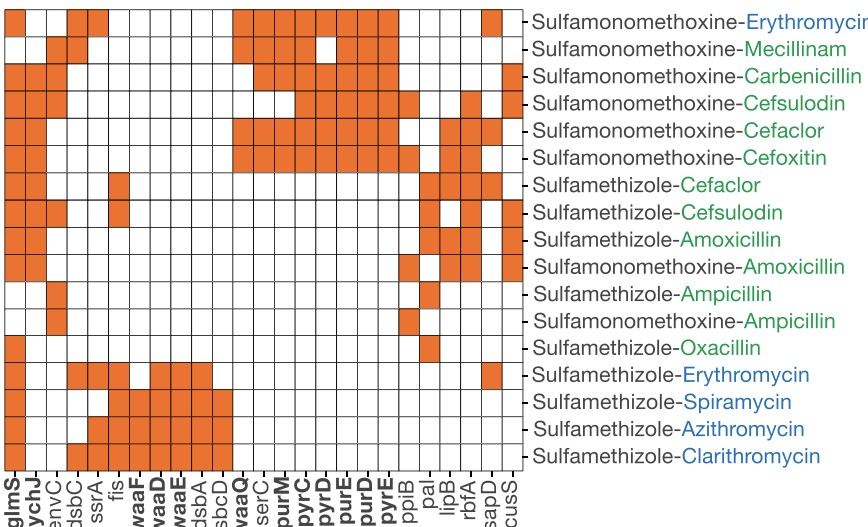

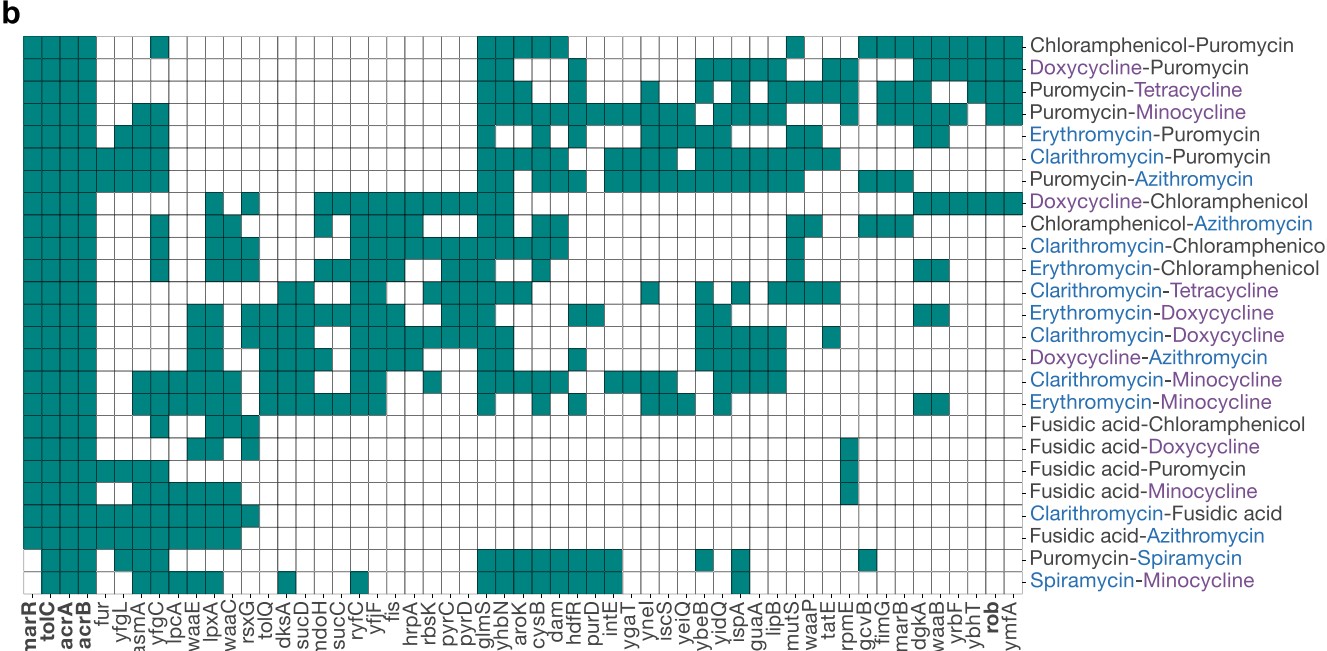

**Extended Data Fig. 3 | Chemical genetics can uncover the biological processes that drive interactions between antibiotic classes. a**, Clustered heatmap of discordant mutants that are part of CS interactions between sulfonamides and macrolides (blue) or beta-lactams (green). Genes in bold are involved in LPS or nucleotide biosynthesis. **b**, Clustered heatmap of concordant mutants that are part of XR interactions between tetracyclines (violet), macrolides (blue), and other protein synthesis inhibitors. Genes in bold regulate or are part of the major efflux pump in *E. coli* (AcrAB-TolC).

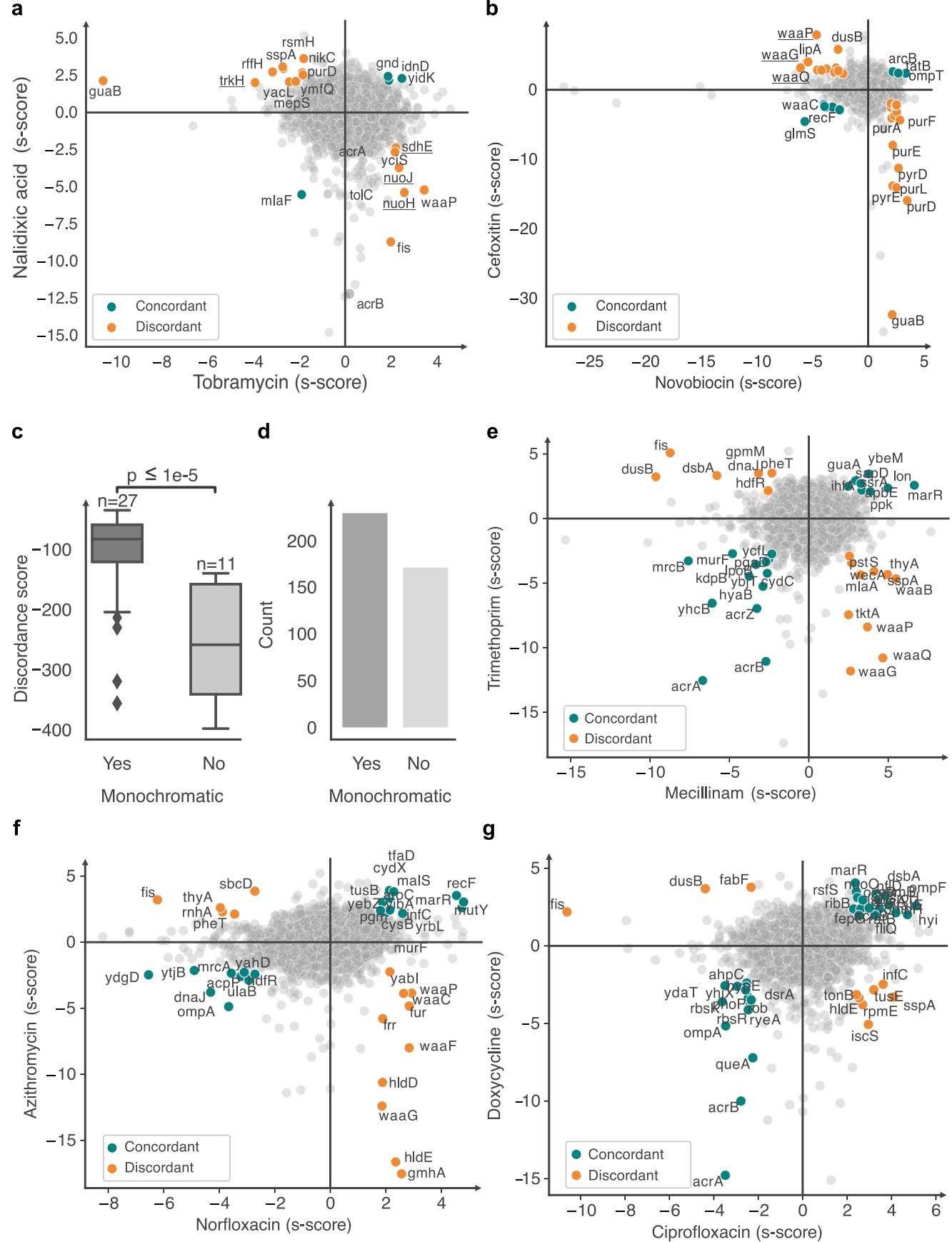

**Extended Data Fig. 4 | See next page for caption.**

**Extended Data Fig. 4 | Chemical genetics can infer mechanisms and monochromaticity of XR and CS drug interactions. a**, Scatter plot of chemical genetic profiles of the *E. coli* deletion library in tobramycin and nalidixic acid[31]. Mutants with concordant (XR-related) and discordant (CS-related) profiles are highlighted. Dots in grey represent mutants that do not have s-scores within the 3% extreme values for both drugs. The underlined knockout mutants are known causal genes of this CS interaction[17,19]. **b**, Chemical genetic profiles for novobiocin and cefoxitin, presented as in **a**. Underlined knockout mutants indicate that the changes in polarity of the lipopolysaccharide (LPS) core can drive resistance to cefoxitin while providing sensitivity to the large and non-polar novobiocin. **c**, Non-monochromatic XR interactions (n=11) have higher absolute discordance scores than their monochromatic counterparts (n=27) (two-sided Mann–Whitney U-test; P = 3.758e-07) - monochromaticity was defined

in the validation experiment. This means that chemical genetics can infer the monochromaticity of XR interactions. The box boundaries represent the first and third quartiles, with the median indicated. The whiskers extend to the furthest data points within 1.5 times the interquartile range (IQR). **d**, The highest discordance score of -133.8481 based on the 11 non-monochromatic XR interactions from **c** was used to separate the remaining inferred XR interactions (excluding the 38 validated) into monochromatic (n=225) or non-monochromatic (n=168). **e**–**g**. Scatter plots of chemical genetic profiles of the *E. coli* deletion library[31] for examples of other pairs of drugs with both high concordance and discordance (in addition to azithromycin and tetracycline shown in Fig. 5b). As the azithromycin-tetracycline pair, those are expected to be non-monochromatic. Data are depicted as in **a,b**. For all data, see the relevant Shiny app at https://shiny-portal.embl.de/shinyapps/app/21_xrcs.

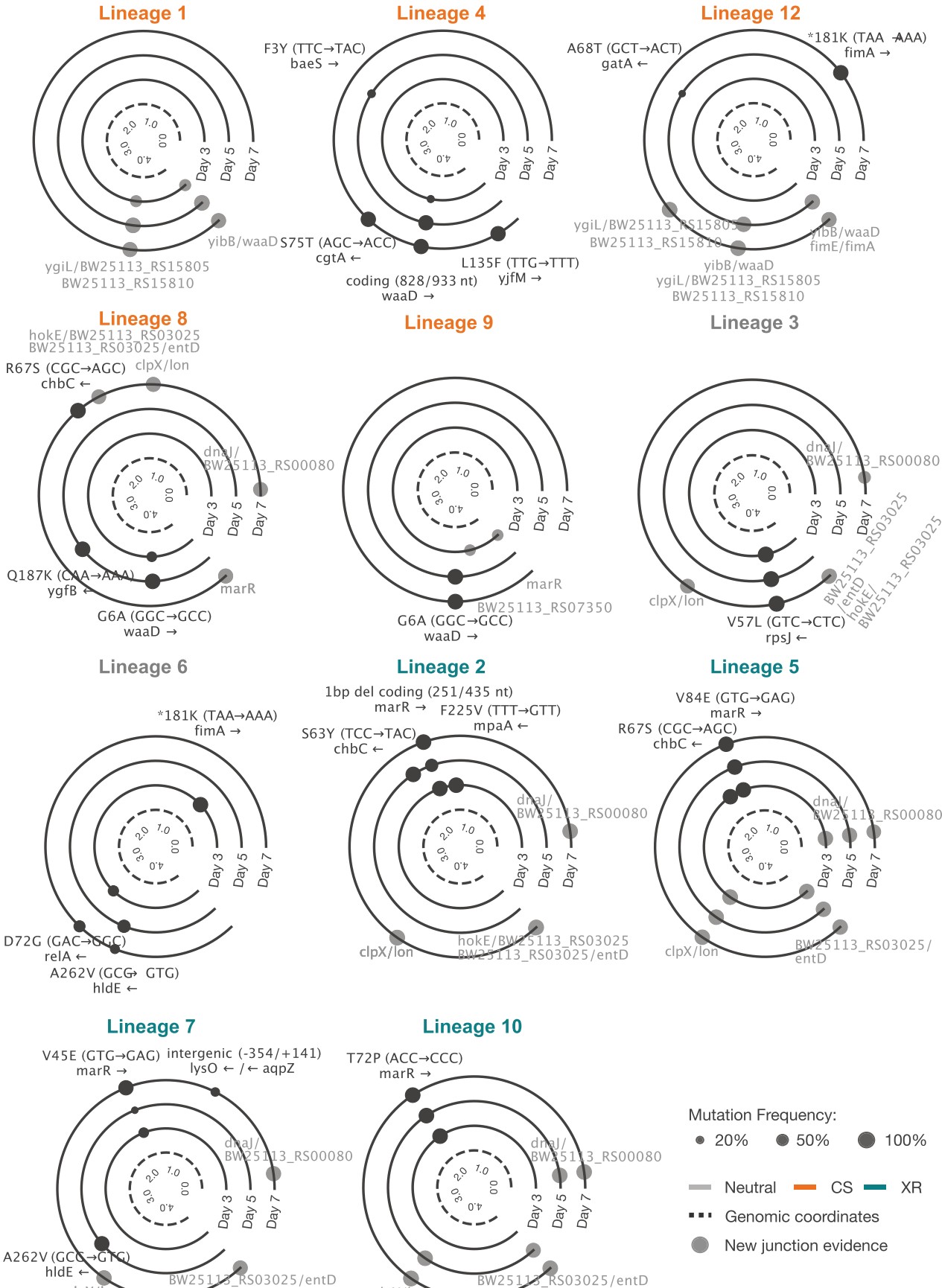

**Extended Data Fig. 5 | Genome sequencing of lineage populations evolved in tetracycline.** Results of the remaining 11 lineages from days 3, 5, and 7. Results are shown as in Fig. 5c, and lineages grouped in XR, CS, and neutral according to classification in Fig. 5a.

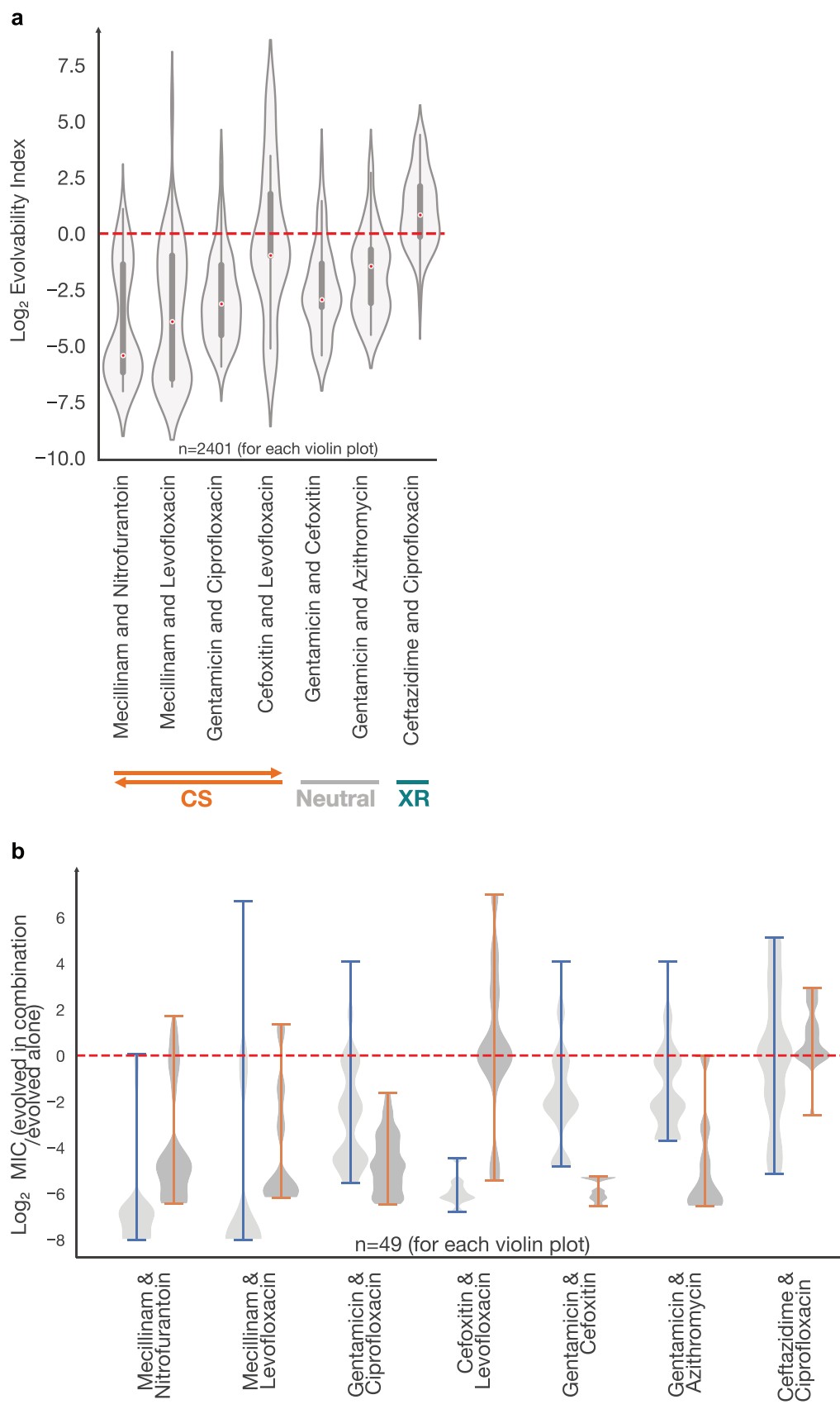

**Extended Data Fig. 6 | See next page for caption.**

**Extended Data Fig. 6 | CS antibiotic combinations constrain resistance evolution to one or both compounds. a**, $\log_2$ transformed evolvability Index as originally proposed in the literature (equation (3), Methods) confirms observations made using the slightly modified Evolvability Index (equation (2), Methods) – Fig. 6b. Red dots on the violin plots represent the median. The box boundaries represent the first and third quartiles, with the median indicated. The whiskers extend to the furthest data points within 1.5 times the interquartile range (IQR). **b**, The $\log_2$ of MIC (IC90) of the evolved population in both drugs compared evolved population of the drug itself is used to identify whether and how well combining drugs reduces resistance to each drug compared to single-drug treatments. Reciprocal CS drug pairs do this efficiently. The red dashed line shows the no-effect when combining drugs does not change resistance evolution to single drug treatments. The bars in the violin plots represent the distributions of $\log_2$ MIC ratios for each antibiotic combination.

# Reporting Summary

## Statistics

For all statistical analyses, confirm that the following items are present in the figure legend, table legend, main text, or Methods section.

| n/a | Confirmed | |
|---|---|---|
| ☐ | ☒ | The exact sample size (*n*) for each experimental group/condition, given as a discrete number and unit of measurement |
| ☐ | ☒ | A statement on whether measurements were taken from distinct samples or whether the same sample was measured repeatedly |
| ☐ | ☒ | The statistical test(s) used AND whether they are one- or two-sided *Only common tests should be described solely by name; describe more complex techniques in the Methods section.* |
| ☒ | ☐ | A description of all covariates tested |
| ☐ | ☒ | A description of any assumptions or corrections, such as tests of normality and adjustment for multiple comparisons |
| ☐ | ☒ | A full description of the statistical parameters including central tendency (e.g. means) or other basic estimates (e.g. regression coefficient) AND variation (e.g. standard deviation) or associated estimates of uncertainty (e.g. confidence intervals) |
| ☐ | ☒ | For null hypothesis testing, the test statistic (e.g. *F*, *t*, *r*) with confidence intervals, effect sizes, degrees of freedom and *P* value noted *Give P values as exact values whenever suitable.* |
| ☒ | ☐ | For Bayesian analysis, information on the choice of priors and Markov chain Monte Carlo settings |
| ☒ | ☐ | For hierarchical and complex designs, identification of the appropriate level for tests and full reporting of outcomes |
| ☐ | ☒ | Estimates of effect sizes (e.g. Cohen's *d*, Pearson's *r*), indicating how they were calculated |

*Our web collection on statistics for biologists contains articles on many of the points above.*

## Software and code

Policy information about availability of computer code

| Data collection | Biomek FX (Beckman Coulter); SoftMax Pro 7.1; BioTek Gen5 (v.3.02.2); TECAN i-control (v1.10); |
|---|---|
| Data analysis | R v.4.1.2; R packages drc (v0.5.8), Bioconductor (v3.16); Python (v3.9.17); Python package scikit-learn (v1.1.3), Python library SciPy (v1.12.0) |

For manuscripts utilizing custom algorithms or software that are central to the research but not yet described in published literature, software must be made available to editors and reviewers. We strongly encourage code deposition in a community repository (e.g. GitHub). See the Nature Portfolio guidelines for submitting code & software for further information.

## Data

Policy information about availability of data

All manuscripts must include a data availability statement. This statement should provide the following information, where applicable:

- Accession codes, unique identifiers, or web links for publicly available datasets
- A description of any restrictions on data availability
- For clinical datasets or third party data, please ensure that the statement adheres to our policy

All supplementary data is provided in supplementary tables. A reference genome from the NCBI database (E. coli BW25113 strain K-12 chromosome; GCF_000750555.1) was used. Raw reads of sequenced samples (file names are describe samples) are in Zenodo DOI: 10.5281/zenodo.10572857 All data is included in the Shiny app: https://shiny-portal.embl.de/shinyapps/app/21_xrcs

## Research involving human participants, their data, or biological material

Policy information about studies with human participants or human data. See also policy information about sex, gender (identity/presentation), and sexual orientation and race, ethnicity and racism.

| | |
|---|---|
| Reporting on sex and gender | N/A |
| Reporting on race, ethnicity, or other socially relevant groupings | N/A |
| Population characteristics | N/A |
| Recruitment | N/A |
| Ethics oversight | N/A |

Note that full information on the approval of the study protocol must also be provided in the manuscript.

# Field-specific reporting

Please select the one below that is the best fit for your research. If you are not sure, read the appropriate sections before making your selection.

☒ Life sciences ☐ Behavioural & social sciences ☐ Ecological, evolutionary & environmental sciences

For a reference copy of the document with all sections, see nature.com/documents/nr-reporting-summary-flat.pdf

# Life sciences study design

All studies must disclose on these points even when the disclosure is negative.

| | |
|---|---|
| Sample size | For all experiments, no prior assumptions were made regarding sample size. In the validation experiment (Fig. 3b-d) we used 12 lineages per antibiotic and 7 lineages in evolution against drug combinations (Fig. 6). Experiments were performed in at least two biological (independent overnight cultures) and maximum of six biological replicates (Fig. 5). Given the number of lineages and number of biological replicates per lineages, the sample size is sufficient for evolution experiments. For number of replicates for each experiment, please see figure legends and "Methods". |
| Data exclusions | No exclusions were performed for experimental data obtained in this project (for details please see "Methods"). |
| Replication | Experiments were always conducted in at least two and at max six biological (independent overnight cultures). For number of replicates for each experiment, please see figure legends and "Methods". |
| Randomization | Randomization was not relevant to our study. Randomization is less relevant in experimental evolution of antibiotic resistance, the main experimental method of our study, because the primary focus is on how a specific selective pressure, like antibiotics, drives predictable evolutionary changes. Since bacterial populations are often clonal and mutations arise spontaneously, the outcomes are largely determined by this selective pressure rather than random variation. As a result, randomization of experimental conditions is unnecessary, as the experiment is designed to observe resistance development in response to a controlled environment. Controls were always processed in parallel. |
| Blinding | For in vitro experiments, no blinding was performed, since it was necessary to keep track of which samples are used in each experiments. Controls were always processed in parallel. |

# Reporting for specific materials, systems and methods

We require information from authors about some types of materials, experimental systems and methods used in many studies. Here, indicate whether each material, system or method listed is relevant to your study. If you are not sure if a list item applies to your research, read the appropriate section before selecting a response.

## Materials & experimental systems

| n/a | Involved in the study |
|---|---|
| ☒ | Antibodies |
| ☒ | Eukaryotic cell lines |
| ☒ | Palaeontology and archaeology |
| ☒ | Animals and other organisms |
| ☒ | Clinical data |
| ☒ | Dual use research of concern |
| ☒ | Plants |

## Methods

| n/a | Involved in the study |
|---|---|
| ☒ | ChIP-seq |
| ☒ | Flow cytometry |
| ☒ | MRI-based neuroimaging |

## Plants

| Seed stocks | N/A |
|---|---|
| Novel plant genotypes | N/A |
| Authentication | N/A |

nature portfolio | reporting summary

April 2023

