## [Peer Review File · Nature Microbiology]

Systematic mapping of antibiotic cross-resistance and collateral sensitivity with chemical genetics

Corresponding Author: Dr Athanasios Typas

Version 0:

Reviewer comments:

Reviewer #1

(Remarks to the Author)

In the manuscript "Systematic mapping of antibiotic cross-resistance and collateral sensitivity with chemical genetics", Sakenova et al. developed a metric to reveal a detailed network of collateral sensitivity (CS) and cross-resistance (XR) using a combination of computational analysis, previously known interactions from the literature, and chemogenomic profiling.

The Authors utilized a white-box machine-learning model, namely the Decision Tree model, combined with their modified t-test (s-score) to infer CS and XR interactions in *Escherichia coli* strain BW25113. They then used an evolutionary experiment model to verify their inferred results to establish a vast network of evolutionary interactions. The Authors also confirmed that chemical genetics data can pinpoint CS and XR mechanisms that emerge and get selected during experimental evolution, thereby helping us to rationalize the dynamics of non-monochromatic antibiotic interactions - all without the need of genome sequencing. Lastly, the Authors also convincingly showed that newly identified CS pairs (especially the bidirectional ones) used in combination could reduce resistance evolution compared to single drugs.

Mapping the network of interactions between resistance mutations is imperative - their clinical relevance is unquestionable - but our current collective knowledge of such interactions is extremely limited due to experimental constraints in their assessment. This fact alone elevates the importance of such studies that provides a systematic framework to map XR/CS interactions and their mechanisms.

However, it is important to focus on one of the key limitations of the study: the evolution experiment used 5 passages over 5 days, with a small population size. Such experimental setups tend to limit the results to loss of function mutations; while these results align with the Authors' inferred interactions and their chemogenomic profiling experiments, it would be beneficial to expand on the limitations of these experiments to include gain of function mutations as well. The shortness of the evolutionary experiments is emphasized even by the Authors on Figure 5a. While these important initial steps of bacterial evolution are well represented by their evolutionary approach, these early loss of function mutants are only stepping stones before the emergence of target mutations providing greater resistance and possibly different interactions with other antibiotics.

Major points:

1) In line # 114-115, #346-347 and throughout the text, the Authors claim that the XR/CS detection via experimental evolution is prone to false negatives, as previous studies (showing a high level of discrepancy - 56 out of the 91 tested antibiotic pairs) were under sampling the antibiotic resistance solution space.

It is important to note, however, that they used evolutionary experiments to validate their inferred XR/CS interactions from the chemogenomic profiling. This evolutionary experiment showed a high 93% accuracy with their inferred interactions - thus adequately validating it. It is partly because in their short-term evolution experiment (using a small bacterial population size $\sim 10^6$, 5 passages, 50 generations) the fixed resistance mutations are most likely loss of function mutations more resembling the results of the chemogenomic screens. A long-term evolution experiment (with large bacterial population sizes $\sim 10^8$, 20 passages, 200 generations) is more likely to select for gain of function resistance mutations (on the target gene of the drug). I would suggest for the Authors to further elucidate and rationalize their chosen experimental evolution protocol in the text.

The observed large discrepancy between previously published XR/CS results are mainly the consequence of other possible explanations listed by the Authors as well in line #109-113. These are as follows: (a) selection biases in evolution experiments (e.g. different selection pressure, drug resistance level cutoffs), (b) slightly different criteria used in each study to define XR/CS, (c) low power to call interactions (limited number of lineages tested), and (d) population complexity (resistance or sensitivity assessment is done for lineage populations).

2) Why do the Authors claim the superior accuracy of chemical genetics simply from the fact that there were 12 out of 13 interactions (8 false negative and 4 false positive cases) for which their predictions (and the results of their evolution experiments) contradicted previous studies (in #187-195 lines)?

Does it not rather mean that their short-term evolution experiment better resembles the inferred XR/CS derived from the chemogenomic screens than the long-term evolutionary experiments performed in other previous studies? Do the Authors have additional relevant arguments to convince the readers that their chemogenomic/evolutionary approach is superior and should be used to resolve more than a hundred cases of prior conflicts and/or misclassifications reported in the literature (as stated in #377-378)?

3) In lines of 204-205: "In seven XR cases we failed to detect the expected bidirectionality, and in 4 further cases, we failed to detect the interaction overall. Overall, this highlights again that experimental evolution experiments are prone to false negative calls (even with large number of lineages being evolved), and uncovers an unexpected tendency for XR interactions to be non-monochromatic."

Have the authors validated the 4 cases where they failed to detect CR interactions following the evolution experiments? One of these methods of validation could be used:

- a) The Authors know which gene deletions predict the CR interactions. So one option would be to newly create the responsible gene knockouts and validate resistance profile and the XR phenotype by measuring the MIC changes.
- b) Increase the duration of the evolution experiments (~20 passages) and remeasure MIC.
- c) Sequencing the evolved line to verify the mutations responsible for resistance.

Without one of these possible validations it would be inappropriate to claim that these interactions are false negatives as there can be possible false positive results originating from the data of the chemogenomic screen or from the nature of the decision tree algorithm, from which the OCDM metric was devised.

4) Somewhere between lines #197- 208 it would be also important to highlight that the model failed to detect 8 unidirectional CS as well.

5) The Authors should explain why the model is not suitable for directionality analysis and emphasize it more on Figure 3 since the heatmaps (on Figure 3c-e) are a bit confusing, especially the validated and bidirectional columns. If the interactions are both neutral, why is the bidirectional value no and how can the validated column be yes if the XR/CS interaction is unidirectional (e.g. Azithromycin-Ceftazidime where none of the evolved azithromycin lines showed XR to Ceftazidime, however the inferred interaction was XR and the validation is labeled as yes). It is also important to note, that based on these Lineage heatmaps, it would seem that the Authors classified any combination that showed both XR and CS as XR following the evolution experiments, even when the possibility of CS seemed to outweigh XR; it would be of some importance to emphasize this either in text or in the figure legend.

6) The Authors state (in line #115-117) that: "...we designated as XR or CS drug pairs that exhibited an interaction in at least one study, even if they were neutral in other(s)". But how did they account for possible unidirectional interactions of the previous studies? It would be important to clarify this as well.

7) It would be also greatly beneficial to quantify the likelihood/extent of XR and CS based on the interaction results of the evolutionary experiments. If there is only one resistant line showing CS interactions (out of 24) to a drug pair versus all of the 24 evolved lines showing CS in another pair, the biological and clinical relevance might be also similarly heterogeneous.

8) It would be interesting to know the Author's opinion whether or not their current OCDM metric could be applied across different bacterial species (on the chemogenomic data of different strains/species), or whether it would be necessary to begin from the decision tree phase. The Authors could place more emphasis on the table they provided with the unique knockout mutants tested for each drug pair. That way, the scientific community might more readily turn its attention to these genes or the homologs of these genes for further analysis rather than testing the whole arrayed knockout collections.

Minor points:

A) In line # 192-195 "This highlights the superior accuracy of chemical genetics (compared to limited/biased experimental evolution efforts) in mapping CS and XR interactions, and supports that the 103 further drug pair relationships (n=116 total) from our training set warrant reclassification (Extended Data Fig. 2).

B) The numbers (103 and 116) are confusing. Where do they originate from? It would be useful to clarify them for the readers in more detail.

C) On Figure 3C what do the white rectangles represent in the lineages matrix (I guess no interactions, but there is no information about this in the figure legends)?

D) The matrix seems to be slightly confusing. It seems to infer directionality when in reality, it represents the amount of XR and CS interactions across drug classes. This issue can be easily resolved by improving the quality of labeling on the image. How was inconsistency represented when the inferred data differed from the known interactions?

E) In lines #257-259 the Authors state that "at the same time these mutations lower the levels of the OM porins OmpC and OmpF42, allowing cefoxitin and other cephalosporins to enter the cell (43)". It should be mentioned that it is true only in the presence of PhoE otherwise the deletion of both of these porins reduce the accumulation of the cefoxitin (based on the reference #43).

F) The Authors state that chemical genomics limits biases and false negatives (line #401-402). Based on Figure 3c-e and my previous points (#5 above), they do arrive to this resolution via preferentially labeling any interaction as XR, even if their results on lineages present a majority of CS (for example in the cases of Tetracycline and Azithromycin, Norfloxacin and Azithromycin, or Azithromycin and Ceftazidime). This selection bias shouldn't be ignored - it would be more precise to say that the Authors put more emphasis on XR based on its potential dangers in the clinic.

Reviewer #2

(Remarks to the Author)

In this manuscript, Sakenova et al. introduce a computational and experimental framework to study the genetic basis of pleiotropy in the evolution of antibiotic resistance, focusing on the evolution of antibiotic cross-resistance (XR) and collateral sensitivity (CS). They employ four previously collected datasets that reported on cross-resistance and collateral sensitivity for their study. Additionally, they utilize their prior "chemical genetic" datasets to develop a predictive tool to identify pairs of antibiotics that are candidates for evolving cross-resistance and collateral sensitivity phenotypes. Upon identifying several hundred such antibiotic pairs, they experimentally test their predictions for a subset of these pairs by evolving a laboratory strain of *E. coli* in the presence of each antibiotic pair, either individually or in combination. This study is compelling and will likely attract considerable attention from both the infectious disease field and the general public. The merit of the study makes it a strong candidate for publication in a high-profile journal such as *Nature Microbiology*. However, I believe that the manuscript would significantly benefit from improvements, particularly in the computational aspects, which should be revised for more rigorous analysis and clarity. I have listed some of these issues below.

1. I congratulate the authors for assembling such a dataset and for comparing the phenotypic changes reported by different studies. In my view, the authors have done a great job in elucidating the sources of discordance between studies. Indeed, in most evolutionary studies, the sample size is limited, experimental conditions vary, and different operational definitions are used to classify cross-resistance and collateral sensitivity. Consequently, it is challenging to synthesize the results of all these studies and provide a definitive metric for cross-resistance and collateral sensitivity. However, throughout the manuscript, the authors depend on chemical genetic data as a more reliable source, despite the acknowledged limitations of this methodology. The gene deletion library employed only included nonessential gene deletions, omitting approximately 300 essential genes. Moreover, this library does not account for other mutations, such as point mutations, copy number changes, upregulation of genes, and multiple coexisting mutations, which are prevalent in both laboratory evolution experiments and clinical isolates of antibiotic-resistant bacteria. Therefore, comparing two methodologies that are both prone to errors will inevitably lead to some discrepancies, and favoring one over the other for analysis could be misleading. This perspective could be adjusted in the manuscript.

2. Utilizing s-scores to predict cross-resistance and collateral sensitivity profiles is a powerful approach. However, the experimental conditions for obtaining s-scores, such as growth on solid agar and measuring colony size, differ from those commonly used in clinical microbiology laboratories. It raises the question of whether s-scores derived from gene deletion mutants on solid agar versus liquid broth would differ. For example, in our experience, the same *del-trkH* strain (acquired from Yale *E. coli* stock) does not exhibit a collateral sensitivity phenotype in liquid media (LB or minimal M9 media), whereas *TrkH* point mutations in the same gene (reported by Lazar et al. and Oz et al.) become hypersensitive to several non-aminoglycoside antibiotics. To address this discrepancy, the authors could select a set of mutants from their study (approximately 10-20 strains) and compare their susceptibility profiles using both liquid media and solid agar. Should there be no significant difference, it would bolster the credibility of their methodology.

3. The operational definition employed by the authors to determine cross-resistance (XR) or collateral sensitivity (CS) is not clearly defined in the manuscript. It is essential to specify what is considered as cross-resistance or collateral sensitivity. Furthermore, the sensitivity of the analysis to different thresholds should be addressed to provide clarity on how changes in these thresholds may affect the classification of XR or CS. This detail is crucial for the reproducibility and robustness of the results.

4. The evolvability index formula presented in the manuscript appears to have an issue. It may be more appropriate to use fold changes, expressed as the logarithm base 10 of the ratio, instead of using the ratios directly. For example, in cases where there is no cross-resistance, the metric would be $0.5 * (\log_{10}(1) + \log_{10}(1)) = 0$. Calculating the sum of the ratios and then taking the logarithm base 2, as shown in Figure 6b, is likely to be problematic. Therefore, Figure 6b should be revised to reflect the use of log-transformed fold changes.

5. My primary concern relates to the linear classifier model; the cross-validation process is unclear. A confusion matrix has not been provided. It is essential to know whether the authors tested a dataset that was not utilized for training. The Receiver Operating Characteristic (ROC) curve shown in Figure 2c, with an Area Under the Curve (AUC) of approximately 0.7, suggests that there is a discernible pattern the model is capable of learning, provided that this is not a result of overfitting. However, the sensitivity and specificity appear suboptimal. For instance, at a 20 percent false positive rate, the true positive rate is only around 30-35 percent. Additionally, the section describing the Optimal Class Distribution Model (OCDM) is confusing; it seems that it was used to enhance the signal when concordance is high and penalize discordance. I am uncertain whether this introduces bias. Nevertheless, the data from experimental validation seem more promising than the prediction model.

(Remarks to the Author)

Sakenova and co-workers used published chemical genomics data and machine learning to make predictions about collateral sensitivity (CS) and cross-resistance (XR) between antibiotics. The rationale is that the more similar the chemical genomic footprints of two different antibiotics are, the more likely it is that evolving resistance to one of the drugs will also lead to resistance to the other (and vice versa for collateral sensitivity). The authors use a new formula to quantify the difference in chemical genomic footprints, predict many new XR and CS interactions, and perform experimental evolution to test these predictions. They claim that many of the newly predicted interactions are thus confirmed. The specific mutants that lead to the prediction of XR or CS from chemical genomics data can indicate the underlying mechanism. Using evolution experiments, the authors further confirm the general finding of previous studies (PMID: 24068739 and others) that CS can slow down resistance evolution for several new drug pairs. A key conclusion is that the approach based on chemical genomics data can systematically address XR and CS, while experimental evolution as used in previous studies is limited in this respect.

The ability to predict XR and CS based on chemical genomics or other data on individual drugs would be of great interest. In principle, this would solve the combinatorial explosion problem and the technical challenge of running longer evolution experiments, making evolution predictable. Unfortunately, the present work overpromises on these points. Key parts of the study are overstated due to basic flaws in the analysis, and several key conclusions are misleading. The most promising aspect of this work is the mechanistic insight into the underlying mechanism of CS and XR for specific drug pairs.

Specific concerns:

1. The criteria for validating the XR and CS interactions predicted from chemical genomics data are not stringent enough. Figure 3 illustrates this point: Several cases where only one or a few of the 12 lineages showed CS are scored as CS according to Fig. 3c. The figure legend explicitly states that an interaction was considered to be validated if at least one (out of 12) lineages showed the predicted interaction (lines 609-611). Accordingly, in Fig. 3d, several cases where the majority of interactions showed CS but XR was predicted based on the chemical genomics data are shown as validated. The azithromycin-tetracycline pair highlights this issue: Here, 4, 6, and 2 lineages show XR, CS, and neutral interactions, respectively, but according to Fig. 3d, the XR prediction is considered validated. According to the criteria given in the legend, if the prediction had been CS (or neutral), it would also have been validated for this drug pair. There are many other similarly problematic examples. It is clear that these criteria are not useful: A minority of events or a single event consistent with a hypothesis in twelve replicates of an experiment is better described as an outlier, not a validation of the hypothesis. Earlier papers likely found fewer such interactions because they used stringent criteria to validate them.
2. Closely related to the previous point, the threshold for considering a change in MIC to be meaningful seems to be too low. According to the methods, XR and CS were defined based on " \log_2 fold-change $> +1$ or -1 ", which probably means that a twofold difference in MIC in either direction was considered sufficient to classify an interaction as XR or CS. Since concentration gradients with twofold dilutions were used for MIC measurements, a twofold change is the imprecision of this measurement, and much smaller actual changes in MIC than twofold could meet this threshold; i.e., tiny effects (which are irrelevant for drug resistance in practice) or experimental variability could end up being scored as XR or CS. This may be another reason why previous studies have found far fewer such interactions.
3. The selection of drug pairs used to validate XR and CS interactions in the evolution experiments seems biased. It is not clearly explained how the subset of 59 drug pairs was selected. There is a considerable risk of confirmation bias. The exclusion of drug pairs belonging to the same chemical class in this validation is particularly problematic: The authors argue that these are very likely to be XR. However, a common way of evolving beta-lactam resistance is via point mutations in genomic beta-lactamases, which are thus optimized for different beta-lactams, essentially leading to CS (see PMID: 16601193 and many subsequent papers), to give just one example. Such phenomena based on mutations other than loss-of-function are thus excluded from the validation process, resulting in a strong bias in favor of the chemical genomics approach (which by design can only provide information on the effects of loss-of-function mutations).
4. The claim that the approach based on chemical genomics data is "superior" to laboratory evolution for mapping CS and XR interactions (which permeates the entire manuscript, e.g. lines 192-195) is misleading. First, it is paradoxical to conclude that any approach is superior to the technique used to validate it (validation paradox). Second, mutations other than loss-of-function mutations are (by design) completely neglected in chemical genomics. For example, gain-of-function mutations in resistance genes (e.g. beta-lactamases) or drug targets (e.g. DHFR) are known to play a key role in resistance evolution and cannot be captured by this approach. Third, higher-level resistance via spontaneous mutations generally requires multiple mutations. These generally exhibit epistatic interactions, which again are not captured by the chemical genomics approach. All of these phenomena are captured by evolution experiments. The chemical genomics approach is certainly promising in cases where loss-of-function mutations are the key to resistance; these do exist, but the limitations outlined above are severe. Therefore, the argument that chemical genomics "systematically explores the mutational space" and should better capture XR and CS interactions (lines 262-263) is misleading. Beyond these fundamental issues, the technical and other limitations of chemical genomics would also need to be considered and discussed.
5. Closely related to the previous point, the conclusion that "evolution experiments are prone to false negative calls" (line 206) is

problematic on several levels. First, it presumes that the correct calls for XR and CS interactions are known (and given by the prediction based on the chemical genomics data), which they are not. This problem is prominent in the analysis in Figure 4, which appears to simply take the inferred XR and CS interactions (including the unvalidated ones) at face value. Second, this conclusion assumes that the results of the evolution experiments and the analysis in the present paper are correct, while the published results of several other groups are incorrect (e.g., lines 617-619). Given the questionable way in which interactions are called in the present study (points 1 and 2 above), it seems that previous studies simply found many fewer XR and CS interactions because they used rigorous criteria for calling them. In any case, the claim that the results of the authors' evolution experiments are valid while those of previous papers are not would require a detailed explanation of the arguments on which this claim is based.

6. Quantitative aspects are critical to the outcome of evolution experiments, but are not discussed. Factors such as population size, bottleneck at dilution, and selection pressure (in this case, drug concentration) can significantly affect the dynamics and outcome of evolution experiments, including which resistance mutations are selected and how variable the results are. For instance, in the clonal interference regime, deterministic outcomes are more likely. The main text should briefly explain the gist of the experimental protocol, including key factors such as population size, and provide an explanation of the evolutionary regime (clonal interference or other) and the rationale for conducting the experiments in this regime. Demonstrating the careful design of these experiments is particularly important because the authors claim (explicitly) that their chemical genomics-based approach is superior to evolution experiments, and (implicitly) that their evolution experiments (and subsequent analysis) are superior to those in previous work (see previous points).

7. It should be noted that previous publications have identified evolutionary variability as a significant challenge in designing treatments based on CS. The disappearance of the initial CS over time in the evolution experiment (lines 273-275) reinforces these concerns. The conclusions drawn from these observations in lines 277-279 seem overly strong and would require experimental data to support them.

Reviewer #4

(Remarks to the Author)

The manuscript of Sakenova et al., describes a comprehensive analysis of the classifications of pairs of drugs showing cross resistance (XR) versus collateral sensitivity (CS). While combination or sequential use of drugs with XR should be prevented, drugs pairs identified as CS may have a benefit in the clinic. The Typas lab is leading the high throughput characterization of drug interactions, and this work provide an invaluable resource for a broad range of applications involving drug combinations, as well as for designing new drugs.

One of the most important results of this extensive analysis is in pointing out the difficulties in identifying pairs of drugs as resulting in XR. The high level of non monochromatic results, i.e. when the same pair may show XR, neutral or CS in different evolutionary lines, show that the whole concept of attributing XR to a pair of drugs is problematic. The work then follows up of some of these non-monochromatic pairs in evolutionary experiments.

The ability to collect the comprehensive data on many classes of drugs now draws an extremely useful classification of drug classes according to their XR or CS propensity. The authors then demonstrate the usefulness of their analysis in showing the different propensity to evolve resistance in various drug pairs. They identify bidirectional CS interactions as the most promising combinations for preventing the evolution of resistance. This finding, together with the plethora of new CS interactions identified provides an extremely useful playground for devising clever drug combinations. In summary, the work is extremely interesting and well carried out. It provides a comprehensive platform and methodology for characterizing drug interactions, as well as many avenues for further investigations. I enthusiastically support its publication in Nature Microbiology.

Minor comments:

1. I may have missed something but wouldn't we expect the level of discordance to correlate with non-monochromaticity? For example, if there are very resistant mutants to tet that happen to show collateral sensitivity and others with cross resistance to azi, then we could expect that in an evolution experiment, both XR and CS may be attained. This would mean that a drug pair should not only be classified as neutral, CS or XS but also as discordant.
2. Figure 1: I understand that the figure is meant as an illustration but it would still be useful to mention the names of the drugs used in the diagrams of Fig. 1c, as it shows real data of mutants.
3. It would be fascinating to follow up of the classification of antibiotic classes as CS or XR into the results of clinical trials of antibiotic combinations or cycling. Can the authors relate their findings with previous clinical results?
4. So far, the method works only for drug concentrations below the MIC, which may not reflect the clinical setting. It may be useful to mention this limitation and maybe suggest ways to expand the range to above MIC in the future.

Decision Letter:

8th March 2024

Dear Nassos,

Thank you for your patience while your manuscript "Systematic mapping of antibiotic cross-resistance and collateral sensitivity with chemical genetics" was under peer-review at Nature Microbiology. It has now been seen by 4 referees, whose expertise and comments you will find at the end of this email. Although they find your work of some potential interest, they have raised a number of concerns that will need to be addressed before we can consider publication of the work in Nature Microbiology.

In particular, referee #1 is concerned that the study used a relatively low number of passages with a small population size. This referee also says it would be beneficial to expand on the limitations of these experiments to include gain of function mutations, asks for some more validation and is not convinced that the approach is superior to previous methods. Referee #2 has important concerns regarding some of the bioinformatics used. Referee #3 feels that some of the claims made are currently not supported by the data, and questions whether the approach is superior to other methods. Referee #4 has some minor points to be addressed.

Please include a data availability statement as a separate section after Methods but before references, under the heading "Data Availability". This section should inform readers about the availability of the data used to support the conclusions of your study. This information includes accession codes to public repositories (data banks for protein, DNA or RNA sequences, microarray, proteomics data etc...), references to source data published alongside the paper, unique identifiers such as URLs to data repository entries, or data set DOIs, and any other statement about data availability. At a minimum, you should include the following statement: "The data that support the findings of this study are available from the corresponding author upon request", mentioning any restrictions on availability. If DOIs are provided, we also strongly encourage including these in the Reference list (authors, title, publisher (repository name), identifier, year). For more guidance on how to write this section please see: <http://www.nature.com/authors/policies/data/data-availability-statements-data-citations.pdf>

* If you have not done so already we suggest that you begin to revise your manuscript so that it conforms to our Article format instructions at <http://www.nature.com/nmicrobiol/info/final-submission>. Refer also to any guidelines provided in this letter.

When submitting the revised version of your manuscript, please pay close attention to our [href="https://www.nature.com/nature-portfolio/editorial-policies/image-integrity">Digital Image Integrity Guidelines](https://www.nature.com/nature-portfolio/editorial-policies/image-integrity) and to the following points below:

Link Redacted

Note: This url links to your confidential homepage and associated information about manuscripts you may have submitted or be reviewing for us. If you wish to forward this e-mail to co-authors, please delete this link to your homepage first.

Nature Microbiology is committed to improving transparency in authorship. As part of our efforts in this direction, we are now requesting that all authors identified as 'corresponding author' on published papers create and link their Open Researcher and Contributor Identifier (ORCID) with their account on the Manuscript Tracking System (MTS), prior to acceptance. This applies to primary research papers only. ORCID helps the scientific community achieve unambiguous attribution of all scholarly contributions. You can create and link your ORCID from the home page of the MTS by clicking on 'Modify my Springer Nature account'. For more information please visit [please visit www.springernature.com/orcid](http://www.springernature.com/orcid).

If you wish to submit a suitably revised manuscript we would hope to receive it within 6 months. If you cannot send it within this time, please let us know. We will be happy to consider your revision, even if a similar study has been accepted for publication at Nature Microbiology or published elsewhere (up to a maximum of 6 months).

Reviewer Expertise:

Referee #1: Cross-resistance, collateral sensitivity
Referee #2: Bioinformatics, AMR
Referee #3: Drug combinations
Referee #4: Systems biology

Reviewer Comments:

Reviewer #1 (Remarks to the Author):

In the manuscript "Systematic mapping of antibiotic cross-resistance and collateral sensitivity with chemical genetics", Sakenova et al. developed a metric to reveal a detailed network of collateral sensitivity (CS) and cross-resistance (XR) using a combination of computational analysis, previously known interactions from the literature, and chemogenomic profiling.

The Authors utilized a white-box machine-learning model, namely the Decision Tree model, combined with their modified t-test (s-score) to infer CS and XR interactions in *Escherichia coli* strain BW25113. They then used an evolutionary experiment model to verify their inferred results to establish a vast network of evolutionary interactions. The Authors also confirmed that chemical genetics data can pinpoint CS and XR mechanisms that emerge and get selected during experimental evolution, thereby helping us to rationalize the dynamics of non-monochromatic antibiotic interactions - all without the need of genome sequencing. Lastly, the Authors also convincingly showed that newly identified CS pairs (especially the bidirectional ones) used in combination could reduce resistance evolution compared to single drugs.

Mapping the network of interactions between resistance mutations is imperative - their clinical relevance is unquestionable - but our current collective knowledge of such interactions is extremely limited due to experimental constraints in their assessment. This fact alone elevates the importance of such studies that provides a systematic framework to map XR/CS interactions and their mechanisms.

However, it is important to focus on one of the key limitations of the study: the evolution experiment used 5 passages over 5 days, with a small population size. Such experimental setups tend to limit the results to loss of function mutations; while these results align with the Authors' inferred interactions and their chemogenomic profiling experiments, it would be beneficial to expand on the limitations of these experiments to include gain of function mutations as well. The shortness of the evolutionary experiments is emphasized even by the Authors on Figure 5a. While these important initial steps of bacterial evolution are well represented by their evolutionary approach, these early loss of function mutants are only stepping stones before the emergence of target mutations providing greater resistance and possibly different interactions with other antibiotics.

Major points:

1) In line # 114-115, #346-347 and throughout the text, the Authors claim that the XR/CS detection via experimental evolution is prone to false negatives, as previous studies (showing a high level of discrepancy - 56 out of the 91 tested antibiotic pairs) were under sampling the antibiotic resistance solution space.

It is important to note, however, that they used evolutionary experiments to validate their inferred XR/CS interactions from the chemogenomic profiling. This evolutionary experiment showed a high 93% accuracy with their inferred interactions - thus adequately validating it. It is partly because in their short-term evolution experiment (using a small bacterial population size $\sim 10^6$, 5 passages, 50 generations) the fixed resistance mutations are most likely loss of function mutations more resembling the results of the chemogenomic screens. A long-term evolution experiment (with large bacterial population sizes $\sim 10^8$, 20 passages, 200 generations) is more likely to select for gain of function resistance mutations (on the target gene of the drug). I would suggest for the Authors to further elucidate and rationalize their chosen experimental evolution protocol in the text.

The observed large discrepancy between previously published XR/CS results are mainly the consequence of other possible explanations listed by the Authors as well in line #109-113. These are as follows: (a) selection biases in evolution experiments (e.g. different selection pressure, drug resistance level cutoffs), (b) slightly different criteria used in each study to define XR/CS, (c) low power to call interactions (limited number of lineages tested), and (d) population complexity (resistance or sensitivity assessment is done for lineage populations).

2) Why do the Authors claim the superior accuracy of chemical genetics simply from the fact that there were 12 out of 13 interactions (8 false negative and 4 false positive cases) for which their predictions (and the results of their evolution experiments) contradicted previous studies (in #187-195 lines)? Does it not rather mean that their short-term evolution experiment better resembles the inferred XR/CS derived from the chemogenomic screens than the long-term evolutionary experiments performed in other previous studies? Do the Authors have additional relevant arguments to convince the readers that their chemogenomic/evolutionary approach is superior and should be used to resolve more than a hundred cases of prior conflicts and/or misclassifications reported in the literature (as stated in #377-378)?

3) In lines of 204-205: "In seven XR cases we failed to detect the expected bidirectionality, and in 4 further cases, we failed to detect the interaction overall. Overall, this highlights again that experimental evolution experiments are prone to false negative calls (even with large number of lineages being evolved), and uncovers an unexpected tendency for XR interactions to be non-monochromatic."

Have the authors validated the 4 cases where they failed to detect CR interactions following the evolution experiments? One of these methods of validation could be used:

- a) The Authors know which gene deletions predict the CR interactions. So one option would be to newly create the responsible gene knockouts and validate resistance profile and the XR phenotype by measuring the MIC changes.
- b) Increase the duration of the evolution experiments (~20 passages) and remeasure MIC.
- c) Sequencing the evolved line to verify the mutations responsible for resistance.

Without one of these possible validations it would be inappropriate to claim that these interactions are false negatives as there can be possible false positive results originating from the data of the chemogenomic screen or from the nature of the decision tree algorithm, from which the OCDM metric was devised.

- 4) Somewhere between lines #197- 208 it would be also important to highlight that the model failed to detect 8 unidirectional CS as well.
- 5) The Authors should explain why the model is not suitable for directionality analysis and emphasize it more on Figure 3 since the heatmaps (on Figure 3c-e) are a bit confusing, especially the validated and bidirectional columns. If the interactions are both neutral, why is the bidirectional value no and how can the validated column be yes if the XR/CS interaction is unidirectional (e.g. Azithromycin-Ceftazidime where none of the evolved azithromycin lines showed XR to Ceftazidime, however the inferred interaction was XR and the validation is labeled as yes). It is also important to note, that based on these Lineage heatmaps, it would seem that the Authors classified any combination that showed both XR and CS as XR following the evolution experiments, even when the possibility of CS seemed to outweigh XR; it would be of some importance to emphasize this either in text or in the figure legend.
- 6) The Authors state (in line #115-117) that: "...we designated as XR or CS drug pairs that exhibited an interaction in at least one study, even if they were neutral in other(s)". But how did they account for possible unidirectional interactions of the previous studies? It would be important to clarify this as well.
- 7) It would be also greatly beneficial to quantify the likelihood/extent of XR and CS based on the interaction results of the evolutionary experiments. If there is only one resistant line showing CS interactions (out of 24) to a drug pair versus all of the 24 evolved lines showing CS in another pair, the biological and clinical relevance might be also similarly heterogeneous.
- 8) It would be interesting to know the Author's opinion whether or not their current OCDM metric could be applied across different bacterial species (on the chemogenomic data of different strains/species), or whether it would be necessary to begin from the decision tree phase. The Authors could place more emphasis on the table they provided with the unique knockout mutants tested for each drug pair. That way, the scientific community might more readily turn its attention to these genes or the homologs of these genes for further analysis rather than testing the whole arrayed knockout collections.

Minor points:

- A) In line # 192-195 "This highlights the superior accuracy of chemical genetics (compared to limited/biased experimental evolution efforts) in mapping CS and XR interactions, and supports that the 103 further drug pair relationships (n=116 total) from our training set warrant reclassification (Extended Data Fig. 2).
- B) The numbers (103 and 116) are confusing. Where do they originate from? It would be useful to clarify them for the readers in more detail.
- C) On Figure 3C what do the white rectangles represent in the lineages matrix (I guess no interactions, but there is no information about this in the figure legends)?
- D) The matrix seems to be slightly confusing. It seems to infer directionality when in reality, it represents the amount of XR and CS interactions across drug classes. This issue can be easily resolved by improving the quality of labeling on the image. How was inconsistency represented when the inferred data differed from the known interactions?
- E) In lines #257-259 the Authors state that "at the same time these mutations lower the levels of the OM porins OmpC and OmpF42, allowing cefoxitin and other cephalosporins to enter the cell (43)". It should be mentioned that it is true only in the presence of PhoE otherwise the deletion of both of these porins reduce the accumulation of the cefoxitin (based on the reference #43).
- F) The Authors state that chemical genomics limits biases and false negatives (line #401-402). Based on Figure 3c-e and my previous points (#5 above), they do arrive to this resolution via preferentially labeling any interaction as XR, even if their results on lineages present a majority of CS (for example in the cases of Tetracycline and Azithromycin, Norfloxacin and Azithromycin, or Azithromycin and Ceftazidime). This selection bias shouldn't be ignored - it would be more precise to say that the Authors put more emphasis on XR based on its potential dangers in the clinic.

Reviewer #2 (Remarks to the Author):

In this manuscript, Sakenova et al. introduce a computational and experimental framework to study the genetic basis of pleiotropy in the evolution of antibiotic resistance, focusing on the evolution of antibiotic cross-resistance (XR) and collateral sensitivity (CS). They employ four previously collected datasets that reported on cross-resistance and collateral sensitivity for their study. Additionally, they utilize their prior "chemical genetic" datasets to develop a predictive tool to identify pairs of antibiotics that are candidates for evolving cross-resistance and collateral sensitivity phenotypes. Upon identifying several hundred such antibiotic pairs, they experimentally test their predictions for a subset of these pairs by evolving a laboratory strain of *E. coli* in the presence of each antibiotic pair, either individually or in combination. This study is compelling and will likely attract considerable attention from both the infectious disease field and the general public. The merit of the study makes it a strong candidate for publication in a high-profile journal such as *Nature Microbiology*. However, I believe that the manuscript would significantly benefit from improvements, particularly in the computational aspects, which should be revised for more rigorous analysis and clarity. I have listed some of these issues below.

1. I congratulate the authors for assembling such a dataset and for comparing the phenotypic changes reported by different studies. In my view, the authors have done a great job in elucidating the sources of discordance between studies. Indeed, in most evolutionary studies, the sample size is limited, experimental conditions vary, and different operational definitions are used to classify cross-resistance and collateral sensitivity. Consequently, it is challenging to synthesize the results of all these studies and provide a definitive metric for cross-resistance and collateral sensitivity. However, throughout the manuscript, the authors depend on chemical genetic data as a more reliable source, despite the acknowledged limitations of this methodology. The gene deletion library employed only included nonessential gene deletions, omitting approximately 300 essential genes. Moreover, this library does not account for other mutations, such as point mutations, copy number changes, upregulation of genes, and multiple coexisting mutations, which are prevalent in both laboratory evolution experiments and clinical isolates of antibiotic-resistant bacteria. Therefore, comparing two methodologies that are both prone to errors will inevitably lead to some discrepancies, and favoring one over the other for analysis could be misleading. This perspective could be adjusted in the manuscript.

2. Utilizing s-scores to predict cross-resistance and collateral sensitivity profiles is a powerful approach. However, the experimental conditions for obtaining s-scores, such as growth on solid agar and measuring colony size, differ from those commonly used in clinical microbiology laboratories. It raises the question of whether s-scores derived from gene deletion mutants on solid agar versus liquid broth would differ. For example, in our experience, the same *del-trkH* strain (acquired from Yale *E. coli* stock) does not exhibit a collateral sensitivity phenotype in liquid media (LB or minimal M9 media), whereas *TrkH* point mutations in the same gene (reported by Lazar et al. and Oz et al.) become hypersensitive to several non-aminoglycoside antibiotics. To address this discrepancy, the authors could select a set of mutants from their study (approximately 10-20 strains) and compare their susceptibility profiles using both liquid media and solid agar. Should there be no significant difference, it would bolster the credibility of their methodology.

3. The operational definition employed by the authors to determine cross-resistance (XR) or collateral sensitivity (CS) is not clearly defined in the manuscript. It is essential to specify what is considered as cross-resistance or collateral sensitivity. Furthermore, the sensitivity of the analysis to different thresholds should be addressed to provide clarity on how changes in these thresholds may affect the classification of XR or CS. This detail is crucial for the reproducibility and robustness of the results.

4. The evolvability index formula presented in the manuscript appears to have an issue. It may be more appropriate to use fold changes, expressed as the logarithm base 10 of the ratio, instead of using the ratios directly. For example, in cases where there is no cross-resistance, the metric would be $0.5 * (\log_{10}(1) + \log_{10}(1)) = 0$. Calculating the sum of the ratios and then taking the logarithm base 2, as shown in Figure 6b, is likely to be problematic. Therefore, Figure 6b should be revised to reflect the use of log-transformed fold changes.

5. My primary concern relates to the linear classifier model; the cross-validation process is unclear. A confusion matrix has not been provided. It is essential to know whether the authors tested a dataset that was not utilized for training. The Receiver Operating Characteristic (ROC) curve shown in Figure 2c, with an Area Under the Curve (AUC) of approximately 0.7, suggests that there is a discernible pattern the model is capable of learning, provided that this is not a result of overfitting. However, the sensitivity and specificity appear suboptimal. For instance, at a 20 percent false positive rate, the true positive rate is only around 30-35 percent. Additionally, the section describing the Optimal Class Distribution Model (OCDM) is confusing; it seems that it was used to enhance the signal when concordance is high and penalize discordance. I am uncertain whether this introduces bias. Nevertheless, the data from experimental validation seem more promising than the prediction model.

Reviewer #3 (Remarks to the Author):

Sakenova and co-workers used published chemical genomics data and machine learning to make predictions about collateral sensitivity (CS) and cross-resistance (XR) between antibiotics. The rationale is that the more similar the chemical genomic footprints of two different antibiotics are, the more likely it is that evolving resistance to one of the drugs will also lead to resistance to the other (and vice versa for collateral sensitivity). The authors use a new formula to quantify the difference in chemical genomic footprints, predict many new XR and CS interactions, and perform experimental evolution to test these predictions. They claim that many of the newly predicted interactions are thus confirmed. The specific mutants that lead to the prediction of XR or CS from chemical genomics data can indicate the underlying mechanism. Using evolution experiments, the authors further confirm the general finding of previous studies (PMID: 24068739 and others) that CS can slow down resistance evolution for several new drug pairs. A key conclusion is that the approach based on chemical genomics data can systematically

address XR and CS, while experimental evolution as used in previous studies is limited in this respect.

The ability to predict XR and CS based on chemical genomics or other data on individual drugs would be of great interest. In principle, this would solve the combinatorial explosion problem and the technical challenge of running longer evolution experiments, making evolution predictable. Unfortunately, the present work overpromises on these points. Key parts of the study are overstated due to basic flaws in the analysis, and several key conclusions are misleading. The most promising aspect of this work is the mechanistic insight into the underlying mechanism of CS and XR for specific drug pairs.

Specific concerns:

1. The criteria for validating the XR and CS interactions predicted from chemical genomics data are not stringent enough. Figure 3 illustrates this point: Several cases where only one or a few of the 12 lineages showed CS are scored as CS according to Fig. 3c. The figure legend explicitly states that an interaction was considered to be validated if at least one (out of 12) lineages showed the predicted interaction (lines 609-611). Accordingly, in Fig. 3d, several cases where the majority of interactions showed CS but XR was predicted based on the chemical genomics data are shown as validated. The azithromycin-tetracycline pair highlights this issue: Here, 4, 6, and 2 lineages show XR, CS, and neutral interactions, respectively, but according to Fig. 3d, the XR prediction is considered validated. According to the criteria given in the legend, if the prediction had been CS (or neutral), it would also have been validated for this drug pair. There are many other similarly problematic examples. It is clear that these criteria are not useful: A minority of events or a single event consistent with a hypothesis in twelve replicates of an experiment is better described as an outlier, not a validation of the hypothesis. Earlier papers likely found fewer such interactions because they used stringent criteria to validate them.
2. Closely related to the previous point, the threshold for considering a change in MIC to be meaningful seems to be too low. According to the methods, XR and CS were defined based on " \log_2 fold-change $> +1$ or -1 ", which probably means that a twofold difference in MIC in either direction was considered sufficient to classify an interaction as XR or CS. Since concentration gradients with twofold dilutions were used for MIC measurements, a twofold change is the imprecision of this measurement, and much smaller actual changes in MIC than twofold could meet this threshold; i.e., tiny effects (which are irrelevant for drug resistance in practice) or experimental variability could end up being scored as XR or CS. This may be another reason why previous studies have found far fewer such interactions.
3. The selection of drug pairs used to validate XR and CS interactions in the evolution experiments seems biased. It is not clearly explained how the subset of 59 drug pairs was selected. There is a considerable risk of confirmation bias. The exclusion of drug pairs belonging to the same chemical class in this validation is particularly problematic: The authors argue that these are very likely to be XR. However, a common way of evolving beta-lactam resistance is via point mutations in genomic beta-lactamases, which are thus optimized for different beta-lactams, essentially leading to CS (see PMID: 16601193 and many subsequent papers), to give just one example. Such phenomena based on mutations other than loss-of-function are thus excluded from the validation process, resulting in a strong bias in favor of the chemical genomics approach (which by design can only provide information on the effects of loss-of-function mutations).
4. The claim that the approach based on chemical genomics data is "superior" to laboratory evolution for mapping CS and XR interactions (which permeates the entire manuscript, e.g. lines 192-195) is misleading. First, it is paradoxical to conclude that any approach is superior to the technique used to validate it (validation paradox). Second, mutations other than loss-of-function mutations are (by design) completely neglected in chemical genomics. For example, gain-of-function mutations in resistance genes (e.g. beta-lactamases) or drug targets (e.g. DHFR) are known to play a key role in resistance evolution and cannot be captured by this approach. Third, higher-level resistance via spontaneous mutations generally requires multiple mutations. These generally exhibit epistatic interactions, which again are not captured by the chemical genomics approach. All of these phenomena are captured by evolution experiments. The chemical genomics approach is certainly promising in cases where loss-of-function mutations are the key to resistance; these do exist, but the limitations outlined above are severe. Therefore, the argument that chemical genomics "systematically explores the mutational space" and should better capture XR and CS interactions (lines 262-263) is misleading. Beyond these fundamental issues, the technical and other limitations of chemical genomics would also need to be considered and discussed.
5. Closely related to the previous point, the conclusion that "evolution experiments are prone to false negative calls" (line 206) is problematic on several levels. First, it presumes that the correct calls for XR and CS interactions are known (and given by the prediction based on the chemical genomics data), which they are not. This problem is prominent in the analysis in Figure 4, which appears to simply take the inferred XR and CS interactions (including the unvalidated ones) at face value. Second, this conclusion assumes that the results of the evolution experiments and the analysis in the present paper are correct, while the published results of several other groups are incorrect (e.g., lines 617-619). Given the questionable way in which interactions are called in the present study (points 1 and 2 above), it seems that previous studies simply found many fewer XR and CS interactions because they used rigorous criteria for calling them. In any case, the claim that the results of the authors' evolution experiments are valid while those of previous papers are not would require a detailed explanation of the arguments on which this claim is based.
6. Quantitative aspects are critical to the outcome of evolution experiments, but are not discussed. Factors such as population size, bottleneck at dilution, and selection pressure (in this case, drug concentration) can significantly affect the dynamics and outcome

of evolution experiments, including which resistance mutations are selected and how variable the results are. For instance, in the clonal interference regime, deterministic outcomes are more likely. The main text should briefly explain the gist of the experimental protocol, including key factors such as population size, and provide an explanation of the evolutionary regime (clonal interference or other) and the rationale for conducting the experiments in this regime. Demonstrating the careful design of these experiments is particularly important because the authors claim (explicitly) that their chemical genomics-based approach is superior to evolution experiments, and (implicitly) that their evolution experiments (and subsequent analysis) are superior to those in previous work (see previous points).

7.

It should be noted that previous publications have identified evolutionary variability as a significant challenge in designing treatments based on CS. The disappearance of the initial CS over time in the evolution experiment (lines 273-275) reinforces these concerns. The conclusions drawn from these observations in lines 277-279 seem overly strong and would require experimental data to support them.

Reviewer #4 (Remarks to the Author):

The manuscript of Sakenova et al., describes a comprehensive analysis of the classifications of pairs of drugs showing cross resistance (XR) versus collateral sensitivity (CS). While combination or sequential use of drugs with XR should be prevented, drugs pairs identified as CS may have a benefit in the clinic. The Typas lab is leading the high throughput characterization of drug interactions, and this work provide an invaluable resource for a broad range of applications involving drug combinations, as well as for designing new drugs.

One of the most important results of this extensive analysis is in pointing out the difficulties in identifying pairs of drugs as resulting in XR. The high level of non monochromatic results, i.e. when the same pair may show XR, neutral or CS in different evolutionary lines, show that the whole concept of attributing XR to a pair of drugs is problematic. The work then follows up of some of these non-monochromatic pairs in evolutionary experiments.

The ability to collect the comprehensive data on many classes of drugs now draws an extremely useful classification of drug classes according to their XR or CS propensity. The authors then demonstrate the usefulness of their analysis in showing the different propensity to evolve resistance in various drug pairs. They identify bidirectional CS interactions as the most promising combinations for preventing the evolution of resistance. This finding, together with the plethora of new CS interactions identified provides an extremely useful playground for devising clever drug combinations. In summary, the work is extremely interesting and well carried out. It provides a comprehensive platform and methodology for characterizing drug interactions, as well as many avenues for further investigations. I enthusiastically support its publication in Nature Microbiology.

Minor comments:

1. I may have missed something but wouldn't we expect the level of discordance to correlate with non-monochromaticity? For example, if there are very resistant mutants to tet that happen to show collateral sensitivity and others with cross resistance to azi, then we could expect that in an evolution experiment, both XR and CS may be attained. This would mean that a drug pair should not only be classified as neutral, CS or XS but also as discordant.
2. Figure 1: I understand that the figure is meant as an illustration but it would still be useful to mention the names of the drugs used in the diagrams of Fig. 1c, as it shows real data of mutants.
3. It would be fascinating to follow up of the classification of antibiotic classes as CS or XR into the results of clinical trials of antibiotic combinations or cycling. Can the authors relate their findings with previous clinical results?
4. So far, the method works only for drug concentrations below the MIC, which may not reflect the clinical setting. It may be useful to mention this limitation and maybe suggest ways to expand the rage to above MIC in the future.

Version 1:

Reviewer comments:

Reviewer #1

(Remarks to the Author)

The Authors have substantially strengthened their results with additional long-term evolution experiments and improved the manuscript by further expanding the limitations of their study. They also provided a shiny app for improve data availability/visualization of their main data (XR/CS interaction inferences based on chemical genetics) making it accessible for the scientific community. I believe that this study provides an important systematic framework to map XR/CS interactions and their mechanisms.

Reviewer #2

(Remarks to the Author)

I appreciate the extensive work that has gone into improving the manuscript. All of my concerns have been addressed except for the evolvability formula. While I understand the desire to adhere to previous publications that were not criticized or assumed to

be accepted, I believe this formula has inherent issues that need to be considered.

Additionally, I am unsure why the new figure was included in the rebuttal letter without a thorough discussion or comparison with the original analysis, beyond stating that the variance was less. However, I will respect the decision of the authors and the editor if they are satisfied with this representation.

Reviewer #4

(Remarks to the Author)

The manuscript is clearer, and I have no further comments. I would have been glad to see directly the real data of Fig. 1c, but understand that the authors intend it only as illustration.

Decision Letter:

Our ref: NMICROBIOL-24010241A

21st June 2024

Dear Nassos,

Thank you for submitting your revised manuscript "Systematic mapping of antibiotic cross-resistance and collateral sensitivity with chemical genetics" (NMICROBIOL-24010241A). It has now been seen by the original referees and their comments are below. The reviewers find that the paper has improved in revision, and therefore we'll be happy in principle to publish it in Nature Microbiology, pending minor revisions to satisfy the referees' final requests and to comply with our editorial and formatting guidelines.

As mentioned before, we consulted with another referee (referee #4) to comment on the concerns by referee #2 and #3 (from the first round of review). I'm copying here the comments by referee #4 which we will also need you to address:

"About the evolvability index raised by reviewer #2 : it seems to me that both definitions may be used, with a slight preference for the formula proposed by the reviewer. Comparing the graph in Fig. 6b with the one from the rebuttal letter shows that the reviewer's formula fits their claim even better. Therefore, I would let the authors choose which formula they prefer to use. However, what appears to have been missed is that the claims in lines 359-361 require a significance test: " In contrast, most lineages treated with CS or neutral 360 combinations evolved lower resistance than those treated with single antibiotics (Fig. 6b). The strongest reduction in resistance evolution occurred for combinations of bidirectional CS pairs". About the response to reviewer #3: the authors have done quite some efforts to tone down their claims and added data to further support their approach."

Editorially, we will require you to add a caveat regarding the evolvability index noting the other approach. Ideally, you should add the result with the other index to the Supplementary Information.

Thank you again for your interest in Nature Microbiology. Please do not hesitate to contact me if you have any questions.

Sincerely,
Francois

Francois Mayer, PhD
Senior Editor
Nature Microbiology

Reviewer #1 (Remarks to the Author):

The Authors have substantially strengthened their results with additional long-term evolution experiments and improved the manuscript by further expanding the limitations of their study. They also provided a shiny app for improve data availability/visualization of their main data (XR/CS interaction inferences based on chemical genetics) making it accessible for the scientific community. I believe that this study provides an important systematic framework to map XR/CS interactions and their mechanisms.

Reviewer #2 (Remarks to the Author):

I appreciate the extensive work that has gone into improving the manuscript. All of my concerns have been addressed except for the evolvability formula. While I understand the desire to adhere to previous publications that were not criticized or assumed to be accepted, I believe this formula has inherent issues that need to be considered.

Additionally, I am unsure why the new figure was included in the rebuttal letter without a thorough discussion or comparison with the original analysis, beyond stating that the variance was less. However, I will respect the decision of the authors and the editor if they are satisfied with this representation.

Reviewer #4 (Remarks to the Author):

The manuscript is clearer, and I have no further comments. I would have been glad to see directly the real data of Fig. 1c, but understand that the authors intend it only as illustration.

Version 2:

Decision Letter:

13th October 2024

Dear Nassos,

I am pleased to accept your Article "Systematic mapping of antibiotic cross-resistance and collateral sensitivity with chemical genetics" for publication in *Nature Microbiology*. Thank you for having chosen to submit your work to us and many congratulations.

Over the next few weeks, your paper will be copyedited to ensure that it conforms to *Nature Microbiology* style. We look particularly carefully at the titles of all papers to ensure that they are relatively brief and understandable.

Please note that *Nature Microbiology* is a Transformative Journal (TJ). Authors may publish their research with us through the traditional subscription access route or make their paper immediately open access through payment of an article-processing charge (APC). Authors will not be required to make a final decision about access to their article until it has been accepted. Find out more about Transformative Journals

Authors may need to take specific actions to achieve compliance with funder and institutional open access mandates. If your research is supported by a funder that requires immediate open access (e.g. according to Plan S principles) then you should select the gold OA route, and we will direct you to the compliant route where possible. For authors selecting the subscription publication route, the journal's standard licensing terms will need to be accepted, including self-archiving policies. Those licensing terms will supersede any other terms that the author or any third party may assert apply to any version of the manuscript.

Congratulations once again and I look forward to seeing the article published.

P.S. Click on the following link if you would like to recommend Nature Microbiology to your librarian
<http://www.nature.com/subscriptions/recommend.html#forms>

** Visit the Springer Nature Editorial and Publishing website at http://editorial-jobs.springernature.com?utm_source=ejP_NMicro_email&utm_medium=ejP_NMicro_email&utm_campaign=ejp_NMicro for more information about our career opportunities. If you have any questions please click [here](mailto:editorial.publishing.jobs@springernature.com).

We thank the reviewers for the time and effort they put into reviewing our manuscript. Their comments have helped us identify and address the limitations of the first submission. In this revised version, we have addressed all main points raised by the reviewers, which has strengthened the manuscript. Many of the points raised were concentrated on the large validation experiment we performed in order to benchmark our chemical-genetics-based metric. We have now defined better how we assessed XR and CS in these experiments, increased the drug pairs we validated, and for a subset did a longer experimental evolution. All the data point to the metric being very robust. We also explained why our chemical-genetics-based metric agrees even better with our experimental evolution experiments than with previous ones, but we avoided focusing too much on this part – as this study is not about devising the most accurate ways to measure XR/CS by experimental evolution. Second, we presented better the definitions of XR and CS from the chemical genetics metric, and to allow the reader to grasp better who the OCDM score works and the underlying data, we generated a user-friendly shiny app with all our data. Last, we outlined the pros and the cons of our metric, and how this compares with experimental evolution, but avoided superiority statements about its accuracy, since these raised earlier some concerns by reviewers. It is clear from the data we present that the method is very accurate, and we find this is enough to make the method appealing to many others in the future, especially as chemical genetics data become increasingly available in different organisms. The method has other obvious pros (mechanism insights, throughput, systematic nature) and cons (not surveying gain-of-function mutations or epistatic interactions) compared to experimental evolution.

Reviewer #1 (Remarks to the Author):

In the manuscript “Systematic mapping of antibiotic cross-resistance and collateral sensitivity with chemical genetics”, Sakenova et al. developed a metric to reveal a detailed network of collateral sensitivity (CS) and cross-resistance (XR) using a combination of computational analysis, previously known interactions from the literature, and chemogenomic profiling. The Authors utilized a white-box machine-learning model, namely the Decision Tree model, combined with their modified t-test (s-score) to infer CS and XR interactions in *Escherichia coli* strain BW25113. They then used an evolutionary experiment model to verify their inferred results to establish a vast network of evolutionary interactions. The Authors also confirmed that chemical genetics data can pinpoint CS and XR mechanisms that emerge and get selected during experimental evolution, thereby helping us to rationalize the dynamics of non-monochromatic antibiotic interactions - all without the need of genome sequencing. Lastly, the Authors also convincingly showed that newly identified CS pairs (especially the bidirectional ones) used in combination could reduce resistance evolution compared to single drugs. Mapping the network of interactions between resistance mutations is imperative - their clinical relevance is unquestionable - but our current collective knowledge of such interactions is extremely limited due to experimental constraints in their assessment. This fact alone elevates the importance of such studies that provides a systematic framework to map XR/CS interactions and their mechanisms.

However, it is important to focus on one of the key limitations of the study: the evolution experiment used 5 passages over 5 days, with a small population size. Such experimental setups tend to limit the results to loss of function mutations; while these results align with the Authors' inferred interactions and their chemogenomic profiling experiments, it would be beneficial to expand on the limitations of these experiments to include gain of function mutations as well. The shortness of the evolutionary experiments is emphasized even by the Authors on Figure 5a. While these important initial steps of bacterial evolution are well represented by their evolutionary approach, these early loss of function mutants are only stepping stones before the emergence of target mutations providing greater resistance and possibly different interactions with other antibiotics.

We thank the reviewer for the thorough evaluation of our manuscript and insightful comments and suggestions. The questions regarding the validation experiment (experimental evolution) have helped us to explain better the concepts and definitions of classifying XR/CS interactions used in this study. They also prompted us to improve the data availability/visualization of our main data (XR/CS interaction inferences based on chemical genetics) and to create a Shiny app that makes it easier for the user to link the inferred interactions to the underlying data.

Major points:

1) In line # 114-115, #346-347 and throughout the text, the Authors claim that the XR/CS detection via experimental evolution is prone to false negatives, as previous studies (showing a high level of discrepancy - 56 out of the 91 tested antibiotic pairs) were under sampling the antibiotic resistance solution space.

It is important to note, however, that they used evolutionary experiments to validate their inferred XR/CS interactions from the chemogenomic profiling. This evolutionary experiment showed a high 93% accuracy with their inferred interactions - thus adequately validating it. It is partly because in their short-term evolution experiment (using a small bacterial population size $\sim 10^6$, 5 passages, 50 generations) the fixed resistance mutations are most likely loss of function mutations more resembling the results of the chemogenomic screens. A long-term evolution experiment (with large bacterial population sizes $\sim 10^8$, 20 passages, 200 generations) is more likely to select for gain of function resistance mutations (on the target gene of the drug). I would suggest for the Authors to further elucidate and rationalize their chosen experimental evolution protocol in the text.

The observed large discrepancy between previously published XR/CS results are mainly the consequence of other possible explanations listed by the Authors as well in line #109-113. These are as follows: (a) selection biases in evolution experiments (e.g. different selection pressure, drug resistance level cutoffs), (b) slightly different criteria used in each study to define XR/CS, (c) low power to call interactions (limited number of lineages tested), and (d) population complexity (resistance or sensitivity assessment is done for lineage populations).

We thank the reviewer for this comment. Indeed, shorter evolution times make it hard for target or other gain-of-function point mutations to emerge, and favor the more common loss-of-function mutations that lead to resistance. In contrast, longer evolution experiments allow for target mutations to appear (when those possible), or for multiple loss-of-function mutations when those have additive effects in resistance. Since the chemical genetics data we used systematically assess the impact of single-gene deletions on fitness, they would naturally agree more with results from shorter rather longer-evolution experiments.

Although all this is true, we are unsure if longer evolution experiments and larger population sizes are necessarily more physiologically relevant. Defining the experimental evolution conditions that are more physiologically relevant for resistance emergence in patients is not easy (besides the obvious inherent caveats of mono-cultures of pathogens in LB). This is perhaps the reason that all previous studies we collected XR/CS data from (or other studies done in other microbes) have used very different conditions of experimental evolution (population size, bottleneck, and number of generations). This difference is one of the reasons data from different studies disagree, something we had noted from the beginning, but now emphasized further.

In the first version, we had one drug pair (tetracycline - azithromycin), for which we had performed a longer evolution experiment to exactly show that interactions can change with the duration of the evolution experiment (and pressure applied) for individual lineages, but not overall. To answer the reviewer's question, which is about this point, and to also address more broadly whether chemical genetics can also capture longer evolution experimental data, we

performed a “long-term” evolution experiment (~100 generations) for 13 further drug pairs in addition to the tetracycline-azithromycin pair. Within the 13 drug pairs, we included 3 drug pairs for which the short experimental evolution did not match our chemical genetic inference. 11/14 drug pairs validated the chemical genetic inference in both the 100 and 50 generations-long evolution (note that the 11 pairs do not overlap 100%). Although things do change for individual lineages with time and interactions become more neutral (or XR), the overall drug interaction, taking into account a large enough number of lineages and as we define it (see also answer in comment 5), remains rather robust and is captured well by chemical genetics.

Action points: We explained why a “short-term” experimental evolution could match better the results of chemical genetics (lines 214-217). We performed a longer experimental evolution experiment for 14 drug pairs as validation for the chemical genetics inferences - this data is presented in **Extended Data Fig. 2e**, illustrating that chemical genetics and OCDM also agree well with XR/CS measurements from longer evolution experiments. We highlighted that the interactions can change with the duration of the evolution experiment (and pressure applied) and that there is a tendency for less CS and more neutrality or XR as resistance to one drug increases (lines 221-224; 303-305). We also emphasized more that one of the reasons for discrepancies in outcomes of previous studies has to do with differences in evolution experiments (lines 111-115), and that one of the limitations of the current approach is that it uses chemical genetics data from loss-of-function library in discussion (lines 409-416).

2) Why do the Authors claim the superior accuracy of chemical genetics simply from the fact that there were 12 out of 13 interactions (8 false negative and 4 false positive cases) for which their predictions (and the results of their evolution experiments) contradicted previous studies (in #187-195 lines)? Does it not rather mean that their short-term evolution experiment better resembles the inferred XR/CS derived from the chemogenomic screens than the long-term evolutionary experiments performed in other previous studies? Do the Authors have additional relevant arguments to convince the readers that their chemogenomic/evolutionary approach is superior and should be used to resolve more than a hundred cases of prior conflicts and/or misclassifications reported in the literature (as stated in #377-378)?

Our chemical genetics metric infers a different interaction for 116 drug pairs than what was previously reported from experimental evolution. The vast majority (85) were called neutral in (one of) these studies. We previously included 13 drug pairs in our validation experiment (experimental evolution) and confirmed that 12 evolved the same interaction as predicted from the chemical genetics' inferences. Based on this, and the fact that previously published experimental evolution studies had more interactions on which they disagreed rather than agreed (exactly because of many neutrals - false negatives), we decided to keep the chemical-genetic metric as the correct call for presenting all the data in **Fig 4**. Although we don't agree that the chemical genetics inferences are just better because we have done a tailored short-experimental evolution experiment (please see the answer to your previous point, and please also note that previous longer-term evolution experiments disagree between each other, to begin with for common pairs; mostly because of missing interactions), we do agree with the reviewer that we cannot be confident that all other 93 interactions are correctly called by chemical genetics metric. To address this point we have first toned down the notion that the chemical genetics metric is the ground truth and all reclassified or new interactions are true - and explained better how OCDM classifies interactions. Second to be more confident that our OCDM thresholds for separating between CS, neutral, and XR are robust, we tested 8 more reclassified interactions with even lower OCDM scores by experimental evolution - this data is incorporated in **Fig. 3b-d**; pairs 16, 17, 24, 25, 57, 58, 59, 60). We validated 6/8, and the 2 non-validated (CS) had the majority of lineages as CS but had one XR lineage, hence we classified them as XR by our definition (note that both were deemed as neutral in previous reports). So, in total, we validated 18/21 from the reclassified interactions.

Action points: We removed the statements about the superiority of our chemical genetic data in terms of accuracy and explained caveats of the metric (lines 409-416). We also removed emphasis on reclassification (e.g. removed reference to this in the abstract). We also explained more clearly how we define XR and CS based on chemical genetics (lines 168-175) - and that priority to XR is given in both our OCDM metric (drug pairs are defined as XR if there is a high concordance, despite any discordance in chemical genetics; CS pairs are ones with only discordance signal) and in the validation experiment. Last we expanded the number of misclassified interactions we validated (**Fig. 3b-d**), adding more confidence in our metrics and cutoffs.

3) In lines of 204-205: "In seven XR cases we failed to detect the expected bidirectionality, and in 4 further cases, we failed to detect the interaction overall. Overall, this highlights again that experimental evolution experiments are prone to false negative calls (even with large number of lineages being evolved), and uncovers an unexpected tendency for XR interactions to be non-monochromatic."

Have the authors validated the 4 cases where they failed to detect CR interactions following the evolution experiments? One of these methods of validation could be used:

- a) The Authors know which gene deletions predict the CR interactions. So one option would be to newly create the responsible gene knockouts and validate resistance profile and the XR phenotype by measuring the MIC changes.
- b) Increase the duration of the evolution experiments (~20 passages) and remeasure MIC.
- c) Sequencing the evolved line to verify the mutations responsible for resistance.

Without one of these possible validations it would be inappropriate to claim that these interactions are false negatives as there can be possible false positive results originating from the data of the chemogenomic screen or from the nature of the decision tree algorithm, from which the OCDM metric was devised.

The reviewer is right. These 4 interactions are inferred as XR from our chemical genetics data, but not captured as such in the validation experiment using experimental evolution, and most have never been tested before. So, calling them false negatives of the validation experiment may seem like jumping to conclusions. We did this because the validation experiment, like any other experimental evolution, had other bona fide false negatives, as it failed to detect 7 other XR interactions in one of the two directions (XR are by definition bidirectional). In any case, we have now included 3/4 pairs (unfortunately, the streptomycin-resistant lineages from the streptomycin-cefaclor pair lost viability after freezing) in the longer evolution experiment (100 generations), and all 3 drug pairs were validated as XR. So, the chemical genetics-based inference was correct.

Action points: We tested these interactions in a longer evolution experiment and validated that our OCDM metric inferred them correctly - we included the data in **Extended Data Fig. 2e**.

4) Somewhere between lines #197- 208 it would be also important to highlight that the model failed to detect 8 unidirectional CS as well.

We are sorry for the misunderstanding here - all references to directionality (lines 197-208 of the first submission) come from the validation experiment (experimental evolution). We did not attempt to infer directionality with our current model cannot infer directionality. We have now clearly stated this in the text and also improved **Fig. 3b-d** to make it clearer that calls on monochromaticity and directionality are based on validation/evolution experiments.

Action point: We have clarified that the current model does not infer directionality in line 197, and also improved the representation of **Fig. 3b-d** to make this clearer (moving inferences

from chemical genetics to the left of lineages, and results from the validation experiment to the right of lineages).

5) The Authors should explain why the model is not suitable for directionality analysis and emphasize it more on Figure 3 since the heatmaps (on Figure 3c-e) are a bit confusing, especially the validated and bidirectional columns. If the interactions are both neutral, why is the bidirectional value no and how can the validated column be yes if the XR/CS interaction is unidirectional (e.g. Azithromycin-Ceftazidime where none of the evolved azithromycin lines showed XR to Ceftazidime, however the inferred interaction was XR and the validation is labeled as yes). It is also important to note, that based on these Lineage heatmaps, it would seem that the Authors classified any combination that showed both XR and CS as XR following the evolution experiments, even when the possibility of CS seemed to outweigh XR; it would be of some importance to emphasize this either in text or in the figure legend.

We thank the reviewer for pointing out the confusing parts of **Fig. 3**. The data displayed in this figure come from experimental validation. To avoid confusion, we have moved 3 columns (“Validated”, “Directionality” and “Monochromatic”) to the right side of the plot as they represent the result summary for the 12 lineages. The validation column assesses whether the experimental evolution results agreed with what the model inferred from the chemical genetics data. As we probed the interactions in both directions during the validation experiment, we got information on the directionality and this is what is indicated in the “Directionality” column. We did not infer directionality from the model (see next paragraph for an explanation). We marked drug pairs as “validated” even if we only experimentally validated XR/CS in one direction because the interaction was inferred without predicting directionality.

XR interactions are by definition bidirectional. We did not attempt to predict directionality for CS interactions, as we had very few high-confidence data CS directional interactions from previous studies to benchmark our predictions. Theoretically, inferring the directionality of CS interactions is possible based on the dominance of mutants on one side of the discordant quadrants. However, to validate the directionality inference of CS interactions, we would have had to test a prohibitively high number of drug pairs and lineages by experimental evolution.

Finally, one important point to clarify here (also came up from other reviewers) is our decision to call XR all non-monochromatic interactions, even if the lineages with XR were fewer than those of CS. The main reason is that in heterogeneous populations of non-monochromatic interactions, even small frequency mutants that are XR will dominate the population when it is exposed to the second drug, and CS interactions will not matter. Hence such drug pairs will behave as XR in drug combinatorial or alternating therapies. Only monochromatic CS (allowing neutral) combinations are useful for combination therapies.

Action point: We have improved **Fig. 3** and its legend to clarify data that come from the validation set. We have also explained better our criteria for calling XR/CS interactions from the validation experiment (experimental evolution) in lines 202-207.

6) The Authors state (in line #115-117) that: “..we designated as XR or CS drug pairs that exhibited an interaction in at least one study, even if they were neutral in other(s)”. But how did they account for possible unidirectional interactions of the previous studies? It would be important to clarify this as well.

As our model does not predict directionality, we did not take into account the information on directionality from previous studies.

Action point: We have clarified in the text that the directionality of published data was not taken into account when combining datasets in the legend of **Fig 2**.

7) It would be also greatly beneficial to quantify the likelihood/extent of XR and CS based on the interaction results of the evolutionary experiments. If there is only one resistant line showing CS interactions (out of 24) to a drug pair versus all of the 24 evolved lines showing CS in another pair, the biological and clinical relevance might be also similarly heterogeneous.

We thank the reviewer for this suggestion. Although our validation data would indeed allow us to calculate frequencies/likelihood of XR and CS (and underlying data are easily discernible in **Fig. 3b-d** and **ED Fig. 2e**, as each lineage is displayed individually), we would rather not give too much attention to this information in the main text for several reasons. First, because the main part of this work is the inference of interactions based on chemical genetics data - not on the validation set we did - hence we would avoid placing more emphasis on this data. Second, we know that those frequencies are not robust, as they are based on small numbers of lineages and depend on how experimental evolution is done (as discussed in point 1). Third, even if more lineages show CS, but some show XR, then in heterogeneous large populations and for alternating or combinatorial therapies, only XR will dominate.

Action point: We have added this information in the source table of **Fig. 3b-d** as a value (proportion of lineages showing the interaction).

8) It would be interesting to know the Author's opinion whether or not their current OCDM metric could be applied across different bacterial species (on the chemogenomic data of different strains/species), or whether it would be necessary to begin from the decision tree phase. The Authors could place more emphasis on the table they provided with the unique knockout mutants tested for each drug pair. That way, the scientific community might more readily turn its attention to these genes or the homologs of these genes for further analysis rather than testing the whole arrayed knockout collections.

We thank the reviewer for this point. We have discussed how this data can be applied to other species, but retraining of the model may be needed, especially if fitness metrics and dynamic ranges are different. To make the data more accessible to the scientific community, we have built a Shiny app, where users can see drug class-class interactions, select specific drug pairs or pairs based on drug classes, and select specific genes of interest that cause XR/CS.

Action point: We discuss the potential of using the OCDM metric in other bacterial species (lines 401-404) and provide the reader with a user-friendly Shiny app to browse XR/CS interaction data, including viewing interactions by drug class, selecting specific drug pairs or class-based pairs or genes of interest related to XR/CS - the app is now hosted at https://shiny-portal.embl.de/shinyapps/app/21_xrcs .

Minor points:

A) In line # 192-195 "This highlights the superior accuracy of chemical genetics (compared to limited/biased experimental evolution efforts) in mapping CS and XR interactions, and supports that the 103 further drug pair relationships (n=116 total) from our training set warrant reclassification (Extended Data Fig. 2).

We believe this point goes together with B, so we answered below.

B) The numbers (103 and 116) are confusing. Where do they originate from? It would be useful to clarify them for the readers in more detail.

Of the 206 previously measured drug pairs, our metric agreed with 90 and disagreed with 116. The vast number of drug pairs our metric called differently were deemed neutral in previous

studies (85/116). 13 drug pairs (now extended to 21) of those 116 were further tested by experimental evolution, and 12 (now 18) confirmed our inferences from the OCDM metric. Although this implied that many of the now remaining $116 - 21 = 95$ drug interactions are likely predicted accurately from the OCDM metric, we have toned down the language on this.

Action point: We have explained better in lines 176-177 where the 116 number comes from (please note number 103 is now obsolete since we validated more drug pairs from the 116). We have also removed the emphasis of reclassified interactions (e.g. removed reference to this in the abstract), and toned down the claim that this means that the interactions of the remaining 95 ($116 - 21 = 95$) drug pairs are inferred correctly from the OCDM model (lines 212-214)

C) On Figure 3C what do the white rectangles represent in the lineages matrix (I guess no interactions, but there is no information about this in the figure legends)?

The missing boxes mean that there is “no data” for this lineage. This could be for different reasons - most often because we could not evolve resistance for all lineages. In contrast, the white/light grey rectangles refer to neutral interactions, as now stated in the legend.

Action point: We have explained better in the legend of **Fig. 3** legend what missing and white boxes for lineages represent.

D) The matrix seems to be slightly confusing. It seems to infer directionality when in reality, it represents the amount of XR and CS interactions across drug classes. This issue can be easily resolved by improving the quality of labeling on the image. How was inconsistency represented when the inferred data differed from the known interactions?

The matrix that the reviewer refers to is likely **Fig 4b**. In order to avoid confusion, we improved the labelling on the new **Fig. 4** and explained that we only plotted the inferred data from chemical genetics and OCDM metric (with reclassified interactions according to the definitions we used in this study).

Action point: We have improved the labeling of **Fig. 4b** and explained in the legend that the inferred data from chemical genetics and OCDM metric are plotted here.

E) In lines #257-259 the Authors state that “at the same time these mutations lower the levels of the OM porins OmpC and OmpF⁴², allowing cefoxitin and other cephalosporins to enter the cell (43)”. It should be mentioned that it is true only in the presence of PhoE otherwise the deletion of both of these porins reduce the accumulation of the cefoxitin (based on the reference #43).

Thanks for spotting this mistake.

Action point: We have amended the sentence to specify that these mutations lower the levels of the OM porins, OmpC, and OmpF⁴², allowing **less** cefoxitin and other cephalosporins to enter the cell (lines 287-288).

F) The Authors state that chemical genomics limits biases and false negatives (line #401-402). Based on Figure 3c-e and my previous points (#5 above), they do arrive to this resolution via preferentially labeling any interaction as XR, even if their results on lineages present a majority of CS (for example in the cases of Tetracycline and Azithromycin, Norfloxacin and Azithromycin. or Azithromycin and Cefotaxime). This selection bias shouldn't be ignored - it would be more precise to say that the Authors put more emphasis on XR based on its potential dangers in the clinic.

We agree with the reviewer that it is an important point to clarify. As we mentioned before (main point 5), we decided to call XR all non-monochromatic interactions, even if the lineages with XR were fewer than those of CS. The main reason for this is that in heterogeneous populations of non-monochromatic interactions, less frequent mutants that provide XR will dominate over CS relationships when the population is exposed to the second drug. Hence such drug pairs will behave as XR in drug combinations or alternating therapies. We believe this is an important consideration not to inflate the numbers of CS drug pairs. At the end of the day, this decision seems to be also warranted by the fact that some of the CS interactions in lineages are temporal, and become neutral (or XR) with longer evolution times (**ED Fig. 2e**).

Action point: We have also explained better our criteria to call XR/CS interactions in the text (lines 202-207).

Reviewer #2 (Remarks to the Author):

In this manuscript, Sakenova et al. introduce a computational and experimental framework to study the genetic basis of pleiotropy in the evolution of antibiotic resistance, focusing on the evolution of antibiotic cross-resistance (XR) and collateral sensitivity (CS). They employ four previously collected datasets that reported on cross-resistance and collateral sensitivity for their study. Additionally, they utilize their prior "chemical genetic" datasets to develop a predictive tool to identify pairs of antibiotics that are candidates for evolving cross-resistance and collateral sensitivity phenotypes. Upon identifying several hundred such antibiotic pairs, they experimentally test their predictions for a subset of these pairs by evolving a laboratory strain of *E. coli* in the presence of each antibiotic pair, either individually or in combination. This study is compelling and will likely attract considerable attention from both the infectious disease field and the general public. The merit of the study makes it a strong candidate for publication in a high-profile journal such as *Nature Microbiology*. However, I believe that the manuscript would significantly benefit from improvements, particularly in the computational aspects, which should be revised for more rigorous analysis and clarity. I have listed some of these issues below.

We thank the reviewer for the kind words and the feedback on the computational aspects of this work. We have addressed the questions as follows below.

1. I congratulate the authors for assembling such a dataset and for comparing the phenotypic changes reported by different studies. In my view, the authors have done a great job in elucidating the sources of discordance between studies. Indeed, in most evolutionary studies, the sample size is limited, experimental conditions vary, and different operational definitions are used to classify cross-resistance and collateral sensitivity. Consequently, it is challenging to synthesize the results of all these studies and provide a definitive metric for cross-resistance and collateral sensitivity. However, throughout the manuscript, the authors depend on chemical genetic data as a more reliable source, despite the acknowledged limitations of this methodology. The gene deletion library employed only included nonessential gene deletions, omitting approximately 300 essential genes. Moreover, this library does not account for other mutations, such as point mutations, copy number changes, upregulation of genes, and multiple coexisting mutations, which are prevalent in both laboratory evolution experiments and clinical isolates of antibiotic-resistant bacteria. Therefore, comparing two methodologies that are both prone to errors will inevitably lead to some discrepancies, and favoring one over the other for analysis could be misleading. This perspective could be adjusted in the manuscript.

We thank the reviewer for bringing up this point. We had acknowledged before that a main limitation of chemical genetics is the fact that we can only assess the role of loss-of-function mutations of non-essential genes. We have now made this point clearer - i.e. that it does not

address other types of less frequent mutations that can arise during evolution. That being said, we still believe chemical genetics are probing more systematically mutational space than experimental evolution - at least when the latter is performed in a few tens of lineages and with specific experimental conditions.

Action point: We have added the types of mutations we don't survey using chemical genetics in the Discussion (lines 410-415).

2. Utilizing s-scores to predict cross-resistance and collateral sensitivity profiles is a powerful approach. However, the experimental conditions for obtaining s-scores, such as growth on solid agar and measuring colony size, differ from those commonly used in clinical microbiology laboratories. It raises the question of whether s-scores derived from gene deletion mutants on solid agar versus liquid broth would differ. For example, in our experience, the same del-trkH strain (acquired from Yale E. coli stock) does not exhibit a collateral sensitivity phenotype in liquid media (LB or minimal M9 media), whereas TrkH point mutations in the same gene (reported by Lazar et al. and Oz et al.) become hypersensitive to several non-aminoglycoside antibiotics. To address this discrepancy, the authors could select a set of mutants from their study (approximately 10-20 strains) and compare their susceptibility profiles using both liquid media and solid agar. Should there be no significant difference, it would bolster the credibility of their methodology.

The reviewer is indeed correct that not all mutations have the same fitness effects in liquid and agar. But the vast majority actually do. We say this because we (and many others) have repeatedly used the data from the Nichols *et al.* study (2011) to follow up on specific genes and functional complexes data - the vast majority of effects reproduce in liquid. A more common source of error from genome-wide library data comes from secondary mutations that single-gene deletion libraries accumulate. We already knew this when we did the chemical genetics, and came up with 3 ways to deal with it: a) we used a copy of the Keio library from the Mori lab that came straight from the original library (minimizing thus secondary mutations that arise during passaging); b) we were strict in the way we propagated and assayed the library; and c) we included 2 independent clones for each Keio mutant, and when clones behaved differently, we flagged the bad clones, whereas the s-score penalized the irreproducibility of the data (leading to neutral s-scores). So, clones with secondary mutations do not drive any of these data. In contrast, mutants maintained in stock collections may have accumulated secondary compensatory mutations that mask the fitness effect (and phenotypes) of the original mutant.

Nevertheless, to satisfy the reviewer and not leave anything open, we took 10 deletion mutants from the library (*trkH*, *fre*, *lpxA*, *waaF*, *waaP*, *gmhA*, *hldE*, *ompF*, *ompR*, and *lon*), and P1 transduced the mutation to the wildtype (we always do this to make sure we get rid of any non-linked secondary mutations), except for *gmhA* and *hmdE*, which are P1 resistant. We then measured their MICs in liquid in 8 different antibiotics, and results agreed very well with s-scores from chemical genetics (solid), as you can see below.

Antibiotic MICs of knockout mutants in three biological and three technical replicates were measured as described in the methods of the manuscript. Fold-changes are calculated relative to the WT. Mutants are color colored.

Action point: We did not add any of this data in the paper, as we think the QC of the *E. coli* chemical genetics has happened much more rigorously in the original paper, but also in many subsequent studies that have used this data as a starting point. But we provide the data asked here for the reviewer.

3. The operational definition employed by the authors to determine cross-resistance (XR) or collateral sensitivity (CS) is not clearly defined in the manuscript. It is essential to specify what is considered as cross-resistance or collateral sensitivity. Furthermore, the sensitivity of the analysis to different thresholds should be addressed to provide clarity on how changes in these thresholds may affect the classification of XR or CS. This detail is crucial for the reproducibility and robustness of the results.

This is very helpful feedback and a question that was raised one way or another by other reviewers too. We have improved our manuscript to clearly state first how OCDM measures XR or CS (lines 168-175), and second, how we define XR and CS in the validation experiment - experimental evolution (lines 202-207). Regarding the thresholds of OCDM to classify interactions, they were chosen based on the highest f1 score using known interactions. Considering the experimental evolution validation, we did two things differently than previous studies. First, we avoided adding a certain threshold on the fraction of lineages that need to show XR or CS to call the interaction - even if 1/12 exhibited an interaction, we defined the drug pair as XR or CS. We have several reasons to believe that the small numbers of lineages and under-sampling linked to experimental evolution experiments are behind the false negatives and the inconsistencies between studies. Second, we also decided to call XR any drug pair that contained both XR and CS interacting lineages. In heterogeneous populations,

mutants that are XR will dominate when the population is exposed to the second drug - even if some mutations lead to CS. Hence if there is a possibility that a drug pair leads to XR, it should be labelled as such, as this will dominate any CS interactions.

Finally, to improve data accessibility and the data behind the OCDM score, we have built a Shiny app, where users can see drug class-class interactions, select specific drug pairs or pairs based on drug classes, and select specific genes of interest that cause XR/CS.

Action point: We explained better how OCDM measures XR or CS (lines 168-175) and made all data available through a Shiny app. We also added more information on how we define XR and CS in the validation experimental evolution (lines 202-207).

4. The evolvability index formula presented in the manuscript appears to have an issue. It may be more appropriate to use fold changes, expressed as the logarithm base 10 of the ratio, instead of using the ratios directly. For example, in cases where there is no cross-resistance, the metric would be $0.5 * (\log_{10}(1) + \log_{10}(1)) = 0$. Calculating the sum of the ratios and then taking the logarithm base 2, as shown in Figure 6b, is likely to be problematic. Therefore, Figure 6b should be revised to reflect the use of log-transformed fold changes.

We used the established formula for the Evolvability Index (Munck et. al., 2014; PMID: 25391482):

$$\text{Evolvability Index} = \frac{1}{2} * \left(\frac{IC_{90}[\text{Drug 1}]_{\text{Drug1+Drug2}}}{IC_{90}[\text{Drug 1}]_{\text{Drug1}}} + \frac{IC_{90}[\text{Drug 2}]_{\text{Drug1+Drug2}}}{IC_{90}[\text{Drug 2}]_{\text{Drug2}}} \right)$$

According to this, when the index is (significantly) above 1, the drug combination promotes resistance, whereas when it is (significantly) below 1, the drug combination delays resistance, and neutrality is at 1. Also, the evolvability index is always >0. We calculated Log₂ of this index only to have a symmetry around 0 (neutral values).

As we understand, the reviewer is proposing to use the mean of the logged ratios instead of the logged mean of ratios (form below). For example, if one ratio is 2-fold down (drugA_IC90[evolved on A & B] / drugA_IC90[evolved on drug A] = 0.5), and the other ratio is 2-fold up (drugB_IC90[evolved on A & B] / drugB_IC90[evolved on drug B] = 2) then our initial formula would be $\text{Log}_2((0.5 + 2)/2) = 0.32$ (above 0; drug combination promotes resistance), while the reviewer's formula would be $(\text{Log}_2(0.5) + \log_2(2))/2 = 0$ (neutral). We were somewhat unclear of the rationale behind this (amounts to a geometric mean, but not back transformed to the original scale). Results remained very similar, with variance decreasing. However, since changing the published form would require more reasoning, we opted to stay at this point with the original published form. If the reviewer has a formal reason why is better to do this, we would be happy to change this plot in the final version.

The formula as proposed:

$$\text{Evolvability Index} = \frac{1}{2} * \left(\text{Log}_2 \left(\frac{IC_{90}[\text{Drug 1}]_{\text{Drug1+Drug2}}}{IC_{90}[\text{Drug 1}]_{\text{Drug1}}} \right) + \text{Log}_2 \left(\frac{IC_{90}[\text{Drug 2}]_{\text{Drug1+Drug2}}}{IC_{90}[\text{Drug 2}]_{\text{Drug2}}} \right) \right)$$

Plot with proposal formula:

5. My primary concern relates to the linear classifier model; the cross-validation process is unclear. A confusion matrix has not been provided. It is essential to know whether the authors tested a dataset that was not utilized for training. The Receiver Operating Characteristic (ROC) curve shown in Figure 2c, with an Area Under the Curve (AUC) of approximately 0.7, suggests that there is a discernible pattern the model is capable of learning, provided that this is not a result of overfitting. However, the sensitivity and specificity appear suboptimal. For instance, at a 20 percent false positive rate, the true positive rate is only around 30-35 percent. Additionally, the section describing the Optimal Class Distribution Model (OCDM) is confusing; it seems that it was used to enhance the signal when concordance is high and penalize discordance. I am uncertain whether this introduces bias. Nevertheless, the data from experimental validation seem more promising than the prediction model.

There is some misunderstanding here. OCDM does not stand for “Optimal Class Distribution Model”, but for Outlier Concordance Discordance Metric. We also want to clarify that we do not use machine learning models for the classification of XR/CS, but only the OCDM metric (see **Formula 1**). However, we have used a non-linear model (decision tree) to derive the metric using “a five-fold grid search cross-validation” (line 467 of first submission), fully avoiding the use of training data for the test set.

We also want to make the point that the training/ground truth dataset is far from optimal. For the few drug pairs the previously published experimental evolution studies have in common, they disagree in twice more cases than they agree. Almost all disagreements are false negatives in one study (neutral calls). Since most data in our ground truth dataset come from one study (78%), one is to expect that there are many more false negatives (i.e. wrongly deemed neutral interactions) looming in this ground truth dataset. This results in suboptimal ROC curves and “FPR” raising fast before TPR is high enough. Even like this, we get AUC is ~ 0.75, and with FPR ~0.2, we get TPR of ~0.7. We get much better sensitivity and specificity when using our validation set for two reasons: we test more lineages than most previous

studies and relax the criteria to call an interaction XR or CS (one lineage is enough). Both criteria are geared to reduce FN calls of interactions and to accommodate that typical experimental evolution experiments explore part of the solution space in a stochastic manner.

Action point: We have added a confusion matrix in **ED Fig. 1e**, and explained better how we classify XR/CS in OCDM and the reason for giving priority to XR in our OCDM metric - drug pairs are defined as XR if there is high concordance (lines 168-175)

Reviewer #3 (Remarks to the Author):

Sakenova and co-workers used published chemical genomics data and machine learning to make predictions about collateral sensitivity (CS) and cross-resistance (XR) between antibiotics. The rationale is that the more similar the chemical genomic footprints of two different antibiotics are, the more likely it is that evolving resistance to one of the drugs will also lead to resistance to the other (and vice versa for collateral sensitivity). The authors use a new formula to quantify the difference in chemical genomic footprints, predict many new XR and CS interactions, and perform experimental evolution to test these predictions. They claim that many of the newly predicted interactions are thus confirmed. The specific mutants that lead to the prediction of XR or CS from chemical genomics data can indicate the underlying mechanism. Using evolution experiments, the authors further confirm the general finding of previous studies (PMID: 24068739 and others) that CS can slow down resistance evolution for several new drug pairs. A key conclusion is that the approach based on chemical genomics data can systematically address XR and CS, while experimental evolution as used in previous studies is limited in this respect. The ability to predict XR and CS based on chemical genomics or other data on individual drugs would be of great interest. In principle, this would solve the combinatorial explosion problem and the technical challenge of running longer evolution experiments, making evolution predictable. Unfortunately, the present work overpromises on these points. Key parts of the study are overstated due to basic flaws in the analysis, and several key conclusions are misleading. The most promising aspect of this work is the mechanistic insight into the underlying mechanism of CS and XR for specific drug pairs.

We thank Reviewer #3 for their feedback. We think that the comments have helped us to improve our manuscript and make the main messages of our manuscript clearer. Also, to improve the ability of the reader to gain insights into the mechanistic information chemical genetics data provide, we have built a Shiny app, where users can see the genes behind specific or drug class-class interactions, and have a gene-centric view of interactions. The Shiny app is now hosted at https://shiny-portal.embl.de/shinyapps/app/21_xrcs.

Specific concerns:

1. The criteria for validating the XR and CS interactions predicted from chemical genomics data are not stringent enough. Figure 3 illustrates this point: Several cases where only one or a few of the 12 lineages showed CS are scored as CS according to Fig. 3c. The figure legend explicitly states that an interaction was considered to be validated if at least one (out of 12) lineages showed the predicted interaction (lines 609-611). Accordingly, in Fig. 3d, several cases where the majority of interactions showed CS but XR was predicted based on the chemical genomics data are shown as validated. The azithromycin-tetracycline pair highlights this issue: Here, 4, 6, and 2 lineages show XR, CS, and neutral interactions, respectively, but according to Fig. 3d, the XR prediction is considered validated. According to the criteria given in the legend, if the prediction had been CS (or neutral), it would also have been validated for this drug pair. There are many other similarly problematic examples. It is clear that these criteria are not useful: A minority of events or a single event consistent with a hypothesis in twelve replicates of an experiment is better described as an outlier, not a validation of the

hypothesis. Earlier papers likely found fewer such interactions because they used stringent criteria to validate them.

We appreciate these comments, as they have helped us to clarify and communicate better our main messages and decisions for the validation experiment.

First, we want to make the point that **Fig. 3** is the validation experiment for evaluating how well we predict interactions based on chemical genetics. It's not the main message of this study, just a confirmation that the chemical genetics-based metric works. Please note that we first assess our metric on the published data by others, and it works well (ROC on **Fig. 2c**).

Second, the reviewer is right, we have not explained well how we call XR and CS interactions in the validation experiment. This was brought up by reviewer 2 too. To set the stage, we should state that all previous studies (in *E. coli* or other microbes) use different metrics and thresholds to do this. But none of them ask for the majority of lineages to show the XR or CS interaction, or when they do this, they do it on much more lenient/sensitive ways of measuring fitness changes - i.e. do not expect changes in MIC. When comparing data from previous studies, we found that they disagreed more than they agreed (twice as much). In almost all cases they disagreed, one study called an interaction neutral, and the other XR or CS. Thus, we reasoned that the problem of these metrics is false negatives and another is under-sampling solution space. To address this, we decided to test more lineages in the evolution experiment (12 in each direction, 24 for drug pair), and to be more lenient in how we called interactions - even one lineage was enough. Please note that out of 65 XR or CS validated interactions, only 3 are based on one lineage (and 3 more in 2 lineages). It's also important to say that our measurements for changes in MIC are robust and based on multiple replicates. The second decision we made is that XR relationships dominate over CS, a logic we also followed for the OCDM metric. The reason for this was simple. In complex populations, where mutations that lead to both XR and CS to the second drug have been selected and co-exist when the second drug is applied, XR will dominate (even if it is a minority in the population). We have now explained this logic in the main text.

Finally, the layout of **Fig. 3b-d** may have been confusing with respect to which calls come from the OCDM metric and which from validation. We have now separated those to make this clearer. The tetracycline-azithromycin pair brought up as an example by the reviewer was predicted as XR from the OCDM score. In our experimental evolution is also called XR (and deemed validated), because in our logic XR interactions dominate over CS, even if the latter are slightly more. The same logic has hurt us in the validation of other examples - see for example pairs 24 and 25 of the updated **Fig. 3b**, where drug pairs are predicted CS, majority of lineages are CS, but because one lineage is XR, we call interaction XR in the validation experiment (so not validated). Overall, even if we had used a majority rule for non-monochromatic interactions, precision and recall based on the validation experiment would have been similar.

Action points: We have clarified the criteria we used to call interactions in the validation experiment lines 202-207- both that they are along the lines of what has been done before and the reasons for modifications: more lineages included, no threshold for number of lineages showing a strong XR/CS effect to call an interaction, and XR interactions dominate over CS. We have also improved the representation of **Fig. 3b-d** to make clearer what comes from chemical genetics (left of lineages) and what from validation experiment (right of lineages).

2. Closely related to the previous point, the threshold for considering a change in MIC to be meaningful seems to be too low. According to the methods, XR and CS were defined based on " \log_2 fold-change $> +1$ or -1 ", which probably means that a twofold difference in MIC in either direction was considered sufficient to classify an interaction as XR or CS. Since concentration gradients with twofold dilutions were used for MIC measurements, a twofold

change is the imprecision of this measurement, and much smaller actual changes in MIC than twofold could meet this threshold; i.e., tiny effects (which are irrelevant for drug resistance in practice) or experimental variability could end up being scored as XR or CS. This may be another reason why previous studies have found far fewer such interactions.

XR in the evolution experiment (validation) occurs when a resistant lineage to drug 1 is also more than 2-fold resistant than the parent strain to drug 2, whereas in CS cases lineage is more than 2-fold sensitive to drug 2. We do measure MICs in multiplicate replicates (and in drug gradients) to ensure our measurements are robust. Similar MIC thresholds have been used in most previous studies in the field e.g. PMID: 29307490, PMID: 35385351, PMID: 28553265, PMID: 37031190, and PMID: 28593940. Sometimes even smaller effects of fitness have been used (e.g. 10% growth reduction in a given concentration). So, by no means, we are less stringent on this front. Nevertheless, we do understand the reviewer's point and have now emphasized that the robustness of measurements of fitness changes can be one of the potential reasons for discrepancies in calling XR/CS relationships between different studies.

Action point: We have added that differences/robustness of methods used to assess changes in fitness are one of the possible reasons for discrepancies between different experimental evolution studies in calling XR/CS relationships (lines 112-113).

3. The selection of drug pairs used to validate XR and CS interactions in the evolution experiments seems biased. It is not clearly explained how the subset of 59 drug pairs was selected. There is a considerable risk of confirmation bias. The exclusion of drug pairs belonging to the same chemical class in this validation is particularly problematic: The authors argue that these are very likely to be XR. However, a common way of evolving beta-lactam resistance is via point mutations in genomic beta-lactamases, which are thus optimized for different beta-lactams, essentially leading to CS (see PMID: 16601193 and many subsequent papers), to give just one example. Such phenomena based on mutations other than loss-of-function are thus excluded from the validation process, resulting in a strong bias in favor of the chemical genomics approach (which by design can only provide information on the effects of loss-of-function mutations).

As we explain in the text, we pick a large number of drug pairs ($n = 70$), and test them in both directions, to benchmark our OCDM metric. To be as rigorous as possible, we pick mostly new interactions ($n = 38$) or interactions we classify differently from what was previously reported ($n = 21$) that cover a broad range of OCDM scores. Overall, we probe ~9% of all interactions we infer by the OCDM metric, and only the validation set itself is as big or bigger than previously published sets of studies focusing on measuring XR/CS with experimental evolution. We avoided antibiotics of the same class because we thought it would be the easy way out to show we are predicting well. That being said, we do agree with the reviewer that CS between antibiotics of the same class is possible, and we had brought this up in the original paper and cited 2 studies that use CS beta-lactam combinations in specific regimens or specific beta-lactamase genetic backgrounds (lines 380-383 of the first submission).

Please, note that the way we classify XR/CS using OCDM, an XR drug pair does not preclude that upon specific mutations (admittedly deletions in our case) can become CS - our call is based on the predominant patterns of concordance (what most mutants do), and we favor concordance over discordance. Indeed, there are cases like that visible in the library data. To allow readers to access such information for specific drug pairs, we have built a Shiny app, where users can see drug class-class interactions, select specific drug pairs or pairs based on drug classes, and select specific genes of interest that cause XR/CS.

To address more directly this comment, we included 3 beta-lactam pairs in our validation set (drug pairs 33-35, **Fig. 3d**), which we validated as XR with no CS observed for 24 lineages. However, to reiterate this does not mean CS between these drugs or same-class drugs is

impossible (especially if drug classes like beta-lactams are chemically diverse). For example, knockouts of *dsbB* and *dsbA*, which contribute to correct protein folding in the periplasm by disulfide bond formation, make *E. coli* cefaclor resistant but mecillinam sensitive (see the Shiny app; PMID: 35025730).

Action points: We have included three beta-lactam drug pairs in our validation dataset (**Fig. 3d**) - all validated as XR. We also specified that XR interactions in our data can have mutations that lead to CS (lines 173-175), and built a shiny app to make all this data easier accessible to reader.

4. The claim that the approach based on chemical genomics data is "superior" to laboratory evolution for mapping CS and XR interactions (which permeates the entire manuscript, e.g. lines 192-195) is misleading. First, it is paradoxical to conclude that any approach is superior to the technique used to validate it (validation paradox). Second, mutations other than loss-of-function mutations are (by design) completely neglected in chemical genomics. For example, gain-of-function mutations in resistance genes (e.g. beta-lactamases) or drug targets (e.g. DHFR) are known to play a key role in resistance evolution and cannot be captured by this approach. Third, higher-level resistance via spontaneous mutations generally requires multiple mutations. These generally exhibit epistatic interactions, which again are not captured by the chemical genomics approach. All of these phenomena are captured by evolution experiments. The chemical genomics approach is certainly promising in cases where loss-of-function mutations are the key to resistance; these do exist, but the limitations outlined above are severe. Therefore, the argument that chemical genomics "systematically explores the mutational space" and should better capture XR and CS interactions (lines 262-263) is misleading. Beyond these fundamental issues, the technical and other limitations of chemical genomics would also need to be considered and discussed.

We removed the statements about the superiority of our chemical genetic data in terms of accuracy and explained caveats of the metric (lines 409-416). To address the specific points raised:

a) We used experimental evolution in two ways. First, to optimize and benchmark our metric. When doing this, we realized the caveats of existing data as ground truth - data from different studies disagreed more than they agreed. Second, we used experimental evolution to validate new inferences and interactions we classified differently than now reported. In this validation experiment, we tried to deal with some of the caveats and minimize false negatives - increasing lineages tested and omitting strict thresholds on lineages that need to show interaction. Then experimental evolution and chemical genetics became more consistent. We don't see any validation paradox here.

b-c) We agree that epistatic interactions, point mutations, and (to some degree) gain-of-function mutations are not captured by chemical genetics with single-gene deletion libraries (re last part, please note that a deletion of a repressor, like *marR*, does lead to gain of function of the repressed genes, e.g. ones encoding for AcrAB-TolC). We had already acknowledged these as limitations of our metric - we tried to make it even more explicit now. That being said, we still believe chemical genetics are probing more systematically mutational space than experimental evolution - at least when the latter is performed in a few tens of lineages and with specific experimental conditions.

Action point: We have toned down statements about the superiority of this approach (in terms of accuracy) and focused more on the advantages it provides while being open about its limitations. The latter is clearly emphasized now (lines 409-416).

5. Closely related to the previous point, the conclusion that "evolution experiments are prone to false negative calls" (line 206) is problematic on several levels. First, it presumes that the

correct calls for XR and CS interactions are known (and given by the prediction based on the chemical genomics data), which they are not. This problem is prominent in the analysis in Figure 4, which appears to simply take the inferred XR and CS interactions (including the unvalidated ones) at face value. Second, this conclusion assumes that the results of the evolution experiments and the analysis in the present paper are correct, while the published results of several other groups are incorrect (e.g., lines 617-619). Given the questionable way in which interactions are called in the present study (points 1 and 2 above), it seems that previous studies simply found many fewer XR and CS interactions because they used rigorous criteria for calling them. In any case, the claim that the results of the authors' evolution experiments are valid while those of previous papers are not would require a detailed explanation of the arguments on which this claim is based.

We disagree with this point. The fact that experimental evolution experiments are prone to false negative calls is evident just by comparing the data that is available in the literature, without looking at our data. The majority of commonly tested drug pairs are called differently, often missed in one of the studies, and called neutral. Also, by definition having stringent criteria to call hits means you reduce false positives but increase false negatives. As also reviewer 1 points out, these discrepancies between studies are because of several reasons (we had mentioned in the text): selection biases in evolution experiments (e.g. different selection pressure, number of generations used), slight different criteria used in each study to define XR/CS, low power to call interactions (limited number of lineages tested), and population complexity.

In **Fig. 4**, we do indeed take our inferences at “face value” - this includes newly inferred interactions and reclassified ones. Prompted also by a question from reviewer 1, we validated more of the reclassified and new interactions in this manuscript version, a subset also in longer evolution experiments. In terms of reclassified interactions, we have now tested 21 drug pairs in experimental evolution, and validated 18. All of them were called before neutral, and all but 3 of the validated interactions were only probed in a single study before. From the newly inferred interactions, we had an even better precision: 35/38. Although we cannot say that all the rest new or reclassified interactions are also correct, having validated such a large fraction, warrants that we use them for large-scale analysis like in Fig 4.

Action points: We have expanded our validation dataset from 59 to 70 in total drug pairs, keeping precision to >90% (**Fig. 3b-d**). This is a significant fraction of all inferred new and reclassified interactions together (634 + 116 = 750). We have also performed a longer experimental evolution experiment for 14 drug pairs (ED **Fig. 2e**), illustrating that chemical genetics and OCDM also agree well with XR/CS measurements from longer evolution experiments. Overall, we feel confident we can use data for global analyses, but we are explicit to the reader that this does not mean that all new interactions inferred are correct.

6. Quantitative aspects are critical to the outcome of evolution experiments, but are not discussed. Factors such as population size, bottleneck at dilution, and selection pressure (in this case, drug concentration) can significantly affect the dynamics and outcome of evolution experiments, including which resistance mutations are selected and how variable the results are. For instance, in the clonal interference regime, deterministic outcomes are more likely. The main text should briefly explain the gist of the experimental protocol, including key factors such as population size, and provide an explanation of the evolutionary regime (clonal interference or other) and the rationale for conducting the experiments in this regime. Demonstrating the careful design of these experiments is particularly important because the authors claim (explicitly) that their chemical genomics-based approach is superior to evolution experiments, and (implicitly) that their evolution experiments (and subsequent analysis) are superior to those in previous work (see previous points).

We agree with the reviewer that quantitative aspects such as population size, bottleneck at dilution, etc. are critical to the outcome of the evolution experiment. These were described before in methods but are now added in the main text too. That being said, we would like to make the point again that experimental evolution is only a validation experiment for us. The main result of this paper is the chemical-genetics-based approach, which recapitulates well both the published experimental evolution data from others, and even better the validation experiment we performed here. The reason it recapitulates better the latter has to do with the design of the experiment (high number of lineages) and the metric we used being more similar to OCDM (XR outweighing CS in non-monochromatic interactions). We think the chemical genetics approach still has many advantages over experimental evolution - in terms of throughput, more systematic sampling of mutational space, and understanding of the underlying mechanism. But we have removed any unnecessary statements about superiority, and also clearly state the disadvantages it has.

Action points: We have clarified the parameters of experimental evolution in the main text and methods (lines 189-190; 565-571). We have explained why our experimental evolution experiment agrees even better with OCDM score than previous published studies (lines 202-207). We have removed claims of superiority of the OCDM score but highlighted its advantages and disadvantages over experimental evolution (lines 393-399; 409-416).

7. It should be noted that previous publications have identified evolutionary variability as a significant challenge in designing treatments based on CS. The disappearance of the initial CS over time in the evolution experiment (lines 273-275) reinforces these concerns. The conclusions drawn from these observations in lines 277-279 seem overly strong and would require experimental data to support them.

We agree with this reviewer. To clarify this, we added references to previous publications identifying the evolution variability as a challenge in designing a treatment based on CS (PMID: 30659188, 31058961). We have also toned down statements on specific resistance mechanisms. We also want to note that in response to a comment from reviewer 1, we did some longer evolution experiments for 13 more drug pairs, and we did notice some disappearance of CS interactions in lineages (lines 221-224).

Action points: We toned down this statement (lines 307-309), and have added 14 drug pairs that evolved for more generations and exhibit some decrease in CS across different lineages (**ED Fig. 2e**). We comment on this and cite relevant papers that discuss the challenge of designing treatments based on CS in discussion (lines 429-432).

Reviewer #4 (Remarks to the Author):

The manuscript of Sakenova et al., describes a comprehensive analysis of the classifications of pairs of drugs showing cross resistance (XR) versus collateral sensitivity (CS). While combination or sequential use of drugs with XR should be prevented, drugs pairs identified as CS may have a benefit in the clinic. The Typas lab is leading the high throughput characterization of drug interactions, and this work provide an invaluable resource for a broad range of applications involving drug combinations, as well as for designing new drugs. One of the most important results of this extensive analysis is in pointing out the difficulties in identifying pairs of drugs as resulting in XR. The high level of non monochromatic results, i.e. when the same pair may show XR, neutral or CS in different evolutionary lines, show that the whole concept of attributing XR to a pair of drugs is problematic. The work then follows up of some of these non-monochromatic pairs in evolutionary experiments. The ability to collect the comprehensive data on many classes of drugs now draws an extremely useful classification of drug classes according to their XR or CS propensity. The authors then demonstrate the usefulness of their analysis in showing the different propensity

to evolve resistance in various drug pairs. They identify bidirectional CS interactions as the most promising combinations for preventing the evolution of resistance. This finding, together with the plethora of new CS interactions identified provides an extremely useful playground for devising clever drug combinations. In summary, the work is extremely interesting and well carried out. It provides a comprehensive platform and methodology for characterizing drug interactions, as well as many avenues for further investigations. I enthusiastically support its publication in Nature Microbiology.

We thank the reviewer for the positive feedback and suggestions to improve our manuscript.

Minor comments:

1. I may have missed something but wouldn't we expect the level of discordance to correlate with non-monochromaticity? For example, if there are very resistant mutants to tet that happen to show collateral sensitivity and others with cross resistance to azi, then we could expect that in an evolution experiment, both XR and CS may be attained. This would mean that a drug pair should not only be classified as neutral, CS or XS but also as discordant.

We indeed observed that antibiotic pairs with non-monochromatic XR interactions exhibited stronger discordance scores in chemical genetics than the ones with monochromatic XR. We still classified those as XR (and not as discordant, as suggested) as we decided to call an interaction XR, even if some mechanisms would also result in CS. In our opinion, XR interactions outweigh CS ones, as in heterogeneous populations, mutants that are XR will dominate when the population is exposed to the second drug. Hence monochromatic CS drug pairs are the ones relevant for combination therapies.

Action points: We explained better how we define XR and CS based on chemical genetics (lines 168-175) - and that priority to XR is given in our OCDM. To improve our data accessibility, we have also built a Shiny app, where users can see drug class-class interactions, select specific drug pairs or pairs based on drug classes, and select specific genes of interest that cause XR/CS.

2. Figure 1: I understand that the figure is meant as an illustration but it would still be useful to mention the names of the drugs used in the diagrams of Fig. 1c, as it shows real data of mutants.

Action point: We agree with the reviewer and to avoid confusion, we have removed the gene names and replaced them with geneA, geneB, etc.

3. It would be fascinating to follow up of the classification of antibiotic classes as CS or XR into the results of clinical trials of antibiotic combinations or cycling. Can the authors relate their findings with previous clinical results?

This would indeed be interesting. Unfortunately, to our knowledge, no clinical study has systematically tested different antibiotic combinations and probed their impact on antibiotic resistance development. Thus, it is challenging for us at this point to compare clinical results and our data.

4. So far, the method works only for drug concentrations below the MIC, which may not reflect the clinical setting. It may be useful to mention this limitation and maybe suggest ways to expand the range to above MIC in the future.

Chemical genetics data is typically collected at sub-MIC concentrations to allow resolution for mutants with both increased and decreased resistance. Increasing the concentration at or above MIC when performing chemical genetics results in being able to quantify the growth of only a few mutants that make cells more resistant, selecting for secondary suppressor

mutations, and overall, complicating the quantification (both in arrayed colonies on plates as in the Nichols et al. 2011 study or pooled experiments). We believe that such sparse data will be less powerful in capturing XR and CS relationships. In contrast, the comprehensive nature of the sub-MIC chemical genetics data seems good enough to recapitulate well what happens in experimental evolution, in which bacterial populations evolve in concentrations above MIC.

We thank the reviewers for the time and effort they put into reviewing our manuscript

Reviewer #1:

Remarks to the Author:

The Authors have substantially strengthened their results with additional long-term evolution experiments and improved the manuscript by further expanding the limitations of their study. They also provided a shiny app for improve data availability/visualization of their main data (XR/CS interaction inferences based on chemical genetics) making it accessible for the scientific community. I believe that this study provides an important systematic framework to map XR/CS interactions and their mechanisms.

We thank the reviewer for the positive feedback.

Reviewer #2:

Remarks to the Author:

I appreciate the extensive work that has gone into improving the manuscript. All of my concerns have been addressed except for the evolvability formula. While I understand the desire to adhere to previous publications that were not criticized or assumed to be accepted, I believe this formula has inherent issues that need to be considered.

Additionally, I am unsure why the new figure was included in the rebuttal letter without a thorough discussion or comparison with the original analysis, beyond stating that the variance was less. However, I will respect the decision of the authors and the editor if they are satisfied with this representation.

We thank the reviewer for the positive feedback, and we have added the plot with revised version of the formula in the main text, and with the original in ED Fig, explaining the difference in the methods.

Reviewer #4:

Remarks to the Author:

The manuscript is clearer, and I have no further comments. I would have been glad to see directly the real data of Fig. 1c, but understand that the authors intend it only as illustration.

We thank the reviewer for the positive feedback.